# Dynamics-Aligned Latent Imagination in Contextual World Models for Zero-Shot Generalization

**Frank Röder**[*] **Jan Benad** **Manfred Eppe** **Pradeep Kr. Banerjee**[*]
Institute for Data Science Foundations, Blohmstraße 15, 21079 Hamburg, Germany

## Abstract

Real-world reinforcement learning demands adaptation to unseen environmental conditions without costly retraining. Contextual Markov Decision Processes (cMDP) model this challenge, but existing methods often require explicit context variables (e.g., friction, gravity), limiting their use when contexts are latent or hard to measure. We introduce Dynamics-Aligned Latent Imagination (DALI), a framework integrated within the Dreamer architecture that infers latent context representations from agent-environment interactions. By training a self-supervised encoder to predict forward dynamics, DALI generates actionable representations conditioning the world model and policy, bridging perception and control. We theoretically prove this encoder is essential for efficient context inference and robust generalization. DALI's latent space enables counterfactual consistency: Perturbing a gravity-encoding dimension alters imagined rollouts in physically plausible ways. On challenging cMDP benchmarks, DALI achieves significant gains over context-unaware baselines, often surpassing context-aware baselines in extrapolation tasks, enabling zero-shot generalization to unseen contextual variations.

## 1 Introduction

The ability to generalize across diverse environmental conditions is a fundamental challenge in reinforcement learning (RL). Contextual Markov Decision Processes (cMDPs) formalize this challenge by modeling task variations through latent parameters, such as friction, gravity, or object mass that govern dynamics [Hallak et al., 2015]. However, real-world agents rarely have direct access to these parameters. Consider a robotic control task where a legged agent must adapt to different surfaces, such as smooth tiles, rough gravel, or slippery ice. If the agent were explicitly provided with surface friction coefficients, adaptation would be straightforward. In practice, though, such ground-truth annotations are often unavailable or prohibitively expensive to obtain. Instead, the agent must infer these variations implicitly by observing how its actions influence the environment, using proprioceptive feedback and gait dynamics.

Existing approaches to contextual RL often rely on explicit context conditioning, where models are trained with domain-specific instrumentation or carefully crafted architectures [Seyed Ghasemipour et al., 2019, Ball et al., 2021, Eghbal-zadeh et al., 2021, Mu et al., 2022, Beukman et al., 2023, Benjamins et al., 2023, Prasanna et al., 2024]. While effective in controlled settings, such approaches scale poorly and break down in unstructured environments where contexts are latent or difficult to measure. The challenge is to develop RL agents that infer and adapt to hidden contexts in a self-supervised manner, enabling robust generalization without direct supervision.

To address this challenge, we propose *Dynamics-Aligned Latent Imagination* (DALI), a framework integrated within the DreamerV3 architecture [Hafner et al., 2025] that enables zero-shot generalization by inferring latent contexts directly from interaction histories. Unlike DreamerV3, which struggles

---

[*]Equal contribution

to generalize across diverse latent contexts due to its limited ability to retain critical environmental information [Prasanna et al., 2024], DALI overcomes these limitations through a self-supervised context encoder. This encoder learns to predict forward dynamics, capturing the relationship between actions, states, and their consequences. For example, by analyzing how applied forces influence motion trajectories, the encoder distills contextual factors (e.g., gravitational pull or surface friction) into compact, actionable embeddings. These embeddings condition the world model and policy, enabling the agent to reason about hidden dynamics (e.g., how increased inertia affects motion stability) and adapt its control strategies accordingly. Our theoretical analysis demonstrates that DALI's context encoder enables robust zero-shot generalization by efficiently capturing latent environmental variations, overcoming information bottlenecks and sample inefficiencies in learning diverse contexts [Prasanna et al., 2024, Hafner et al., 2025].

By inferring latent contexts from interactions, DALI enhances DreamerV3's learning framework, achieving robust zero-shot generalization in partially observable environments. Our work makes the following key contributions:

- **Theoretical foundations**: We establish that a dedicated context encoder is essential for robust generalization, proving that it infers latent contexts from short interaction windows with near-optimal sample complexity and mitigates information bottlenecks in partially observable settings.
- **Strong zero-shot generalization**: On challenging cMDP benchmarks, DALI achieves up to $+96.4\%$ gains over context-unaware baselines, often surpassing ground-truth context-aware baselines in extrapolation tasks.
- **Physically consistent counterfactuals**: We show that the learned latent space exhibits counterfactual consistency: perturbing a dimension encoding gravity, for instance, results in imagined rollouts where objects fall faster or slower, faithfully mirroring real-world physics.

## 2   Related Work

**Contextual RL for Zero-Shot Generalization.** Contextual RL has been studied in various forms, from cMDPs to domain randomization and meta-RL [Hallak et al., 2015, Modi et al., 2018]. A recent survey [Kirk et al., 2023] highlights its relevance for zero-shot generalization, emphasizing how clear context sets for training and evaluation enable systematic analysis. One research direction assumes context is explicitly observed as privileged information and integrates it into learning [Chen et al., 2018, Seyed Ghasemipour et al., 2019, Ball et al., 2021, Eghbal-zadeh et al., 2021, Sodhani et al., 2021, Mu et al., 2022, Benjamins et al., 2023, Prasanna et al., 2024]. In contrast, we follow recent work that assumes context is latent and must be inferred [Chen et al., 2018, Xu et al., 2019, Lee et al., 2020, Seo et al., 2020, Xian et al., 2021, Sodhani et al., 2022, Melo, 2022, Evans et al., 2022], focusing on self-supervised context inference through forward dynamics alignment.

**Model-Based RL.** Model-based RL improves sample efficiency by learning predictive environment models for planning and imagination. Approaches like Dreamer [Hafner et al., 2019, 2020, 2021, 2025] and TD-MPC [Hansen et al., 2024] achieve strong performance by learning compact latent representations of environment dynamics. Recent work has also explored general-purpose model-free RL that leverages model-based representations for broader applicability [Fujimoto et al., 2025]. While Dreamer has been explored for zero-shot generalization with explicit context conditioning [Prasanna et al., 2024], we build on it with a dynamics-aligned inference mechanism to enhance generalization without context supervision.

**Meta-RL.** Meta-RL trains agents to adapt rapidly to new tasks with minimal experience [Beck et al., 2023], typically by learning adaptive policies that infer task-specific information from past interactions. However, most meta-RL methods require fine-tuning on new tasks [Duan et al., 2016, Finn et al., 2017, Nagabandi et al., 2018, Rakelly et al., 2019, Zintgraf et al., 2019, Melo, 2022]. Our approach, in contrast, aims for zero-shot generalization by leveraging structured latent representations that transfer across environments.

## 3   Preliminaries

To investigate zero-shot generalization in partially observable environments, we adopt a contextual Markov Decision Process (cMDP) framework, following Hallak et al. [2015], Kirk et al. [2023].

A cMDP is defined as a tuple $\mathcal{M} = (\mathcal{S}, \mathcal{A}, \mathcal{O}, \mathcal{C}, \mathcal{R}, \mathcal{P}, \mathcal{E}, \mu, p_{\mathcal{C}})$, where $\mathcal{S}$, $\mathcal{A}$, and $\mathcal{O}$ are resp. the state, action, and observation spaces, $\mathcal{C} \subseteq \mathbb{R}^d$ is the context space with contexts $c \in \mathcal{C}$ drawn i.i.d. from a distribution $p_{\mathcal{C}}$, $\mathcal{R} : \mathcal{S} \times \mathcal{A} \times \mathcal{C} \rightarrow \mathbb{R}$ is the reward function, $\mathcal{P} : \mathcal{S} \times \mathcal{A} \times \mathcal{C} \rightarrow \Delta(\mathcal{S})$ is the stochastic transition function specifying the probability distribution over next states given the current state, action, and context, $\mathcal{E} : \mathcal{S} \times \mathcal{C} \rightarrow \Delta(\mathcal{O})$ is the observation function specifying the distribution over observations given the state and context, $\mu : \mathcal{C} \rightarrow \Delta(\mathcal{S})$ is the initial state distribution where $\mu(s_0|c)$ specifies the probability of initial state $s_0$ given context $c$. In this partially observable setting, the agent does not observe the state $s_t$ or context $c$ directly but receives observations $o_t \in \mathcal{O}$. The objective is to learn a policy $\pi(a_t|o_{1:t}, a_{1:t-1})$ that maximizes the expected return $\mathbb{E}\left[\sum_{t=0}^{T} r_t\right]$, where $T$ is the episode length.

In our framework, each context $c$ induces a distinct variation in the transition dynamics $\mathcal{P}$, such as changes in physical properties (e.g., gravity, friction), while the reward function $\mathcal{R}$ remains fixed across contexts. The context is latent and remains fixed within an episode but varies across episodes according to $p_{\mathcal{C}}$. The agent infers $c$ from interaction histories to adapt its policy. To formalize zero-shot generalization, we define two distributions over contexts: the training distribution $p_{\text{train}}(c)$, from which contexts are sampled during training, and the evaluation distribution $p_{\text{eval}}(c)$, representing unseen test contexts. The goal is to learn a policy using samples from $p_{\text{train}}(c)$ that maximizes the expected return for contexts drawn from $p_{\text{eval}}(c)$ without further adaptation or retraining.

## 4 Dynamics-Aligned Latent Imagination (DALI)

### 4.1 Background: DreamerV3

DreamerV3 [Hafner et al., 2025] is a model-based RL algorithm that enables agents to learn and plan in the latent space of a learned world model. It follows the general structure of the Dreamer family of algorithms [Hafner et al., 2019, 2020, 2021], incorporating three key components: (i) learning a generative world model to simulate environment dynamics, (ii) optimizing policies entirely within the latent space of the world model using imagined rollouts, and (iii) refining the world model and policy through interaction with the real environment.

**World model.** The agent interacts with the environment through observations $o_t$ (e.g., images or sensor data) and actions $a_t$. The world model, structured as a Recurrent State-Space Model (RSSM), encodes these observations into a compact latent state $s_t = \{h_t, z_t\}$, where $h_t = f_\theta(h_{t-1}, z_{t-1}, a_{t-1})$ is a deterministic recurrent state capturing temporal dependencies, and $z_t$ is a stochastic state encoding uncertainty about the current observation. The separation of deterministic ($h_t$) and stochastic ($z_t$) states enables long-horizon temporal reasoning while maintaining diversity in imagined rollouts. The RSSM operates in two distinct modes: (i) during training, the model conditions on the current observation $o_t$ to infer $z_t$, ensuring alignment with real-world dynamics (posterior inference); and (ii) during imagination, the model predicts future states without access to observations, sampling $\hat{z}_t$ to generate hypothetical trajectories (prior prediction). The world model reconstructs observations $\hat{o}$, predicts rewards $\hat{r}_t$, and estimates episode continuations $\hat{n}_t$ from the latent state $s_t$. These components are learned via the following structured models: posterior state representations (encoder) $z_t \sim q_\theta(z_t|h_t, o_t)$, prior state representations $\hat{z}_t \sim p_\theta(\hat{z}_t|h_t)$, reward predictor $\hat{r}_t \sim p_\theta(\hat{r}_t|h_t, z_t)$, continue predictor $\hat{n}_t \sim p_\theta(\hat{n}_t|h_t, z_t)$, and decoder $\hat{o}_t \sim p_\theta(\hat{o}_t|h_t, z_t)$.

**Learning in imagination.** Once the world model is trained, behavior is learned by optimizing a policy entirely within the latent space. An actor-critic architecture guides this process, with the critic estimating the cumulative future reward (value $v_\psi$) and the actor selecting actions $a_\tau \sim \pi_\phi$ to maximize this value. By decoupling policy learning from real-world interaction, DreamerV3 achieves strong sample efficiency, iteratively refining its world model and behaviors through cycles of imagination and execution.

### 4.2 Context Encoder for Dynamics-Aligned Representations

While DreamerV3 is effective in fixed environments, its reliance on static latent states limits adaptability to contextual variations, such as changes in gravity or friction. To address this, we introduce *Dynamics-Aligned Latent Imagination* (DALI), which extends DreamerV3 with a self-supervised context encoder that learns explicit context representations from interaction histories.

### 4.2.1 Forward Dynamics Alignment

The context encoder $g_\varphi$ maps a history of observations and actions $(o_{t-K:t}, a_{t-K:t-1})$ to a context representation $\mathfrak{z}_t = g_\varphi(o_{t-K:t}, a_{t-K:t-1})$, where $K$ defines the temporal window. To align $\mathfrak{z}_t$ with environmental transition dynamics, we optimize a forward dynamics loss:

$$L_{\text{FD}}(\varphi) = \mathbb{E} \left\| o_{t+1} - f_\varphi^w(o_t, a_t, \mathfrak{z}_t) \right\|_2^2, \tag{1}$$

where $f_\varphi^w(o_t, a_t, \mathfrak{z}_t)$ predicts the next observation $\hat{o}_{t+1}$ given the current observation $o_t$, action $a_t$, and context $\mathfrak{z}_t$. Minimizing $L_{\text{FD}}$ ensures that $\mathfrak{z}_t$ encodes contextual factors critical for accurate dynamics prediction, such as variations in physical parameters. The parameters $\varphi$ of $g_\varphi$ and $f_\varphi^w$ are trained jointly, enabling the encoder to capture essential dynamics from short interaction histories.

### 4.2.2 Cross-Modal Regularization

To enhance the context encoder's robustness, we introduce a cross-modal regularization that aligns the context representation $\mathfrak{z}_t = g_\varphi(o_{t-K:t}, a_{t-K:t-1})$ with the DreamerV3 world model's posterior state $z_t \sim q_\theta(z_t|h_t, o_t)$. The cross-modal loss enforces bidirectional alignment between $\mathfrak{z}_t$ and $z_t$:

$$L_{\text{cross}}(\varphi) = \mathbb{E} \left\| z_t - W_z \mathfrak{z}_t \right\|_2^2 + \mathbb{E} \left\| \mathfrak{z}_t - W_{\mathfrak{z}} z_t \right\|_2^2, \tag{2}$$

where $W_z$ and $W_{\mathfrak{z}}$ are linear maps between the context and state spaces. This bidirectional reconstruction leverages $z_t$'s observation-informed representation, which captures instantaneous dynamics relevant to the current context. By aligning with $z_t$ instead of the full latent state $s_t = \{h_t, z_t\}$, $\mathfrak{z}_t$ avoids encoding redundant trajectory-specific information from the deterministic $h_t$, which could impair generalization. The bidirectional constraints prevent degenerate solutions (e.g., $\mathfrak{z}_t$ collapsing to a constant) by enforcing invertibility, ensuring $\mathfrak{z}_t$ remains a rich, context-specific representation consistent with the $z_t$'s latent dynamics. The total loss combines both objectives:

$$L_{\text{total}}(\varphi) = L_{\text{FD}}(\varphi) + \lambda_{\text{cross}} L_{\text{cross}}(\varphi), \tag{3}$$

where $\lambda_{\text{cross}} \in \{0, 1\}$ balances dynamics prediction and cross-modal alignment.

### 4.3 Integrating Context into DreamerV3

To enable context-conditioned imagination and policy learning in DreamerV3 without access to ground-truth context variables, we propose two integration strategies for incorporating the context representation $\mathfrak{z}_t$.

**Shallow Integration.** This approach appends $\mathfrak{z}_t$ to the observation embedding in the world model's encoder, modifying it to $z_t \sim q_\theta(z_t|h_t, o_t, \mathfrak{z}_t)$. All other world model components, including the sequence model $h_t = f_\theta(h_{t-1}, z_{t-1}, a_{t-1})$, predictors $(\hat{z}_t, \hat{r}_t, \hat{n}_t, \hat{o}_t)$, and actor-critic networks $(\pi_\phi(a_\tau|s_\tau), v_\psi(s_t))$, remain unchanged. This lightweight modification enriches latent states with dynamics-aware context, enabling the world model to adapt implicitly to contextual variations.

**Deep Integration.** For deeper context awareness, $\mathfrak{z}_t$ is integrated into the world model and policy as follows: $h_t = f_\theta(h_{t-1}, z_{t-1}, a_{t-1}, \mathfrak{z}_t)$, $\hat{r}_t \sim p_\theta(\hat{r}_t|h_t, z_t, \mathfrak{z}_t)$, $\hat{n}_t \sim p_\theta(\hat{n}_t|h_t, z_t, \mathfrak{z}_t)$, $\hat{o}_t \sim p_\theta(\hat{o}_t|h_t, z_t, \mathfrak{z}_t)$. The posterior $z_t \sim q_\theta(z_t|h_t, o_t)$ and prior $\hat{z}_t \sim p_\theta(\hat{z}_t|h_t)$ access $\mathfrak{z}_t$ indirectly through $h_t$. The actor and critic explicitly condition on $\mathfrak{z}_t$ to optimize imagined trajectories over horizon $H$: $a_\tau \sim \pi_\phi(a_\tau|s_\tau, \mathfrak{z}_t)$, $v_\psi(s_t, \mathfrak{z}_t) \approx \mathbb{E}_{\pi(\cdot|s_\tau, \mathfrak{z}_t)} \sum_{\tau=t}^{t+H} \gamma^{\tau-t} r_\tau$. This ensures that policy optimization explicitly accounts for contextual variations, enabling context-conditioned imagination.

**Training.** Training the context encoder, the loss (3) aligns $\mathfrak{z}_t = g_\varphi(o_{t-K:t}, a_{t-K:t-1})$ with environmental dynamics and requires careful handling of the DreamerV3 world model's recurrence. Both integration strategies unroll the world model over a $K$-length window, initializing $h_{t-K} = \mathbf{0}$ (or a learned initial state), generating $\mathfrak{z}_\tau = g_\varphi(o_{\tau-K:\tau}, a_{\tau-K:\tau-1})$ for $\tau = t - K$ to $t$. In Shallow Integration, $z_\tau \sim q_\theta(z_\tau|h_\tau, o_\tau, \mathfrak{z}_\tau)$ and $h_{\tau+1} = f_\theta(h_\tau, z_\tau, a_\tau)$; in Deep Integration, $z_\tau \sim q_\theta(z_\tau|h_\tau, o_\tau)$ and $h_{\tau+1} = f_\theta(h_\tau, z_\tau, a_\tau, \mathfrak{z}_\tau)$. For both, gradients through $h_\tau$ and $z_\tau$ are stopped in the recurrent dynamics, and through $h_\tau$ in the world model's encoder, preventing updates to $\theta$. In Shallow Integration, $\mathfrak{z}_\tau$ gradients are preserved in the world model's encoder and losses $L_{\text{FD}}$ and $L_{\text{cross}}$, allowing updates to $\varphi$, $W_z$, and $W_{\mathfrak{z}}$. In Deep Integration, $\mathfrak{z}_\tau$ gradients are stopped in the recurrent dynamics and preserved only in $L_{\text{FD}}$ and $L_{\text{cross}}$. This decouples context learning from the world model's recurrence, ensuring $\mathfrak{z}_t$ captures relevant dynamics for both strategies. For detailed pseudocode, see Algorithms 1 and 2 in Appendix B for Shallow Integration, and Algorithms 3 and 4 for Deep Integration.

# 5 Theoretical Insights into DALI's Context Encoder

In this section, we provide an exposition of our key theoretical results, elucidating why DALI's context encoder is essential for robust generalization. We emphasize conceptual insights, with formal statements and proofs deferred to Appendix A.

Our analysis is grounded in a cMDP framework with continuous contexts $c$ (e.g., physical parameters such as gravity or actuator strength), sampled i.i.d. per episode from a distribution $p_{\mathcal{C}}$. Observations are noisy (e.g., $o_t = s_t + \eta_t$, $\eta_t \sim \mathcal{N}(0, \sigma^2 I)$), and dynamics are assumed to be Lipschitz continuous, so that small changes in context (e.g., slight shifts in gravity) lead to smoothly varying behaviors (e.g., comparable object trajectories or locomotion patterns). The observation-action process is $\beta$-mixing [Rio, 2017], meaning distant observations become nearly independent over time (e.g., periodic motions or gait cycles decorrelate under exploratory actions). These structural assumptions are characteristic of many continuous control environments such as DMC [Tassa et al., 2018], CARL [Benjamins et al., 2023], and MetaWorld [Yu et al., 2020], and they underpin DALI's efficient context inference. $\beta$-mixing captures realistic decorrelation in RL tasks, enabling short interaction windows to capture $c$. An exploratory policy further ensures that distinct contexts yield distinguishable trajectories, a common feature in RL training.

Domain randomization trains DreamerV3 across diverse contexts, with $c \sim p_{\mathcal{C}}$, using $h_t^{\text{RSSM}} = f_\theta(h_{t-1}, z_{t-1}, a_{t-1})$ and $z_t \sim q_\theta(z_t | h_t, o_t)$. The recurrent state $h_t^{\text{RSSM}}$ implicitly accumulates information about $c$ over time through its interaction history. For instance, in environments with pendulum-like dynamics, increased gravity may result in faster oscillatory behavior, a pattern that $h_t^{\text{RSSM}}$ can gradually encode after observing enough transitions. However, this strategy has limitations. The recurrent state $h_t^{\text{RSSM}}$ compresses all episode information, including context, states, and noise into a fixed-size GRU, introducing an information bottleneck. As dynamic state information (e.g., object motion) and transient noise compete for limited capacity, essential cues about the underlying context (e.g., gravity) may be lost or delayed. Identifying $c$ often requires accumulating evidence across many transitions, potentially spanning an entire episode of length $T$, since early actions may not sufficiently excite the dynamics to reveal contextual differences. This delay hinders rapid adaptation to novel contexts, as $h_t^{\text{RSSM}}$ needs a long temporal window to reliably disambiguate $c$.

DALI's context encoder decouples context inference from dynamics modeling. It functions as a specialized module that learns a representation $\mathfrak{z}_t$ capturing context information from short, local histories, allowing its recurrent state $h_t^{\text{DALI}}$ to focus on dynamics and reducing the burden on the GRU. The $\beta$-mixing property ensures that $K$ transitions suffice to encode $c$, as the dynamics' dependence on past states fades rapidly. This enables faster adaptation compared to DreamerV3's $h_t^{\text{RSSM}}$. We formalize this intuition for the case of Deep Integration, where the context encoder enhances the RSSM's context awareness. The functions $h(\cdot)$ and $\mathcal{I}(\cdot; \cdot)$ denote the differential entropy and mutual information, respectively [Thomas and Cover, 1999].

**Theorem 1.** *In a cMDP with $\beta$-mixing and Lipschitz dynamics, DALI's context encoder $\mathfrak{z}_t$ captures near-optimal context information, $\mathcal{I}(c; \mathfrak{z}_t) \geq (1-\delta)h(c)$ for $\delta \in (0, 1)$, using $N = \mathcal{O}(1/\delta^2)$ windows of $K = \Omega(\log(1/\delta)/\lambda)$ transitions, where $\lambda$ is the mixing rate. Moreover, DALI's RSSM retains more context information than DreamerV3's, $\mathcal{I}(c; h_t^{DALI}) \geq \mathcal{I}(c; h_t^{RSSM}) - \epsilon(K)$, with $\epsilon(K) = \mathcal{O}(e^{-\lambda K/2})$. Compared to DreamerV3's RSSM processing full episodes of $T \gg K$ transitions, DALI achieves a sample complexity gain of $\mathcal{O}(T/K)$.*

For the formal statement and proof of Theorem 1, see Appendix A. The $\beta$-mixing property ensures that a short window of $K = \Omega(\log(1/\delta)/\lambda)$ transitions is sufficient to achieve high context fidelity. For instance, when $\delta = 0.01$ and the typical mixing rate in DMC tasks is $\lambda \approx 0.1$, a window of $K \approx 64$ steps results in a conditional entropy error bounded by $\delta' \approx 0.1$. DALI requires $N = \mathcal{O}(1/\delta^2)$ such windows, totaling $\mathcal{O}(K/\delta^2)$ transitions. In contrast, DreamerV3's $h_t^{\text{RSSM}}$, constrained by the GRU's finite capacity and observation noise, loses context information and requires $\mathcal{O}(T/\delta^2)$ transitions per episode (e.g., $T = 1000$), where $T \gg K$. The sample complexity gain of $\mathcal{O}(T/K)$ reflects DALI's efficiency in exploiting local histories. Furthermore, $\mathfrak{z}_t$ achieves superior fidelity, with $\mathcal{I}(c; \mathfrak{z}_t)$ approaching $h(c)$ within $\delta$, while DreamerV3's RSSM incurs a larger information loss, $\epsilon(K) = \mathcal{O}(e^{-\lambda K/2})$, due to its bottleneck. While the theorem assumes Deep Integration, this sample complexity gain remains consistent across Deep and Shallow Integration, as $\mathfrak{z}_t$'s efficiency derives from its training on $K$-length windows, which is identical in both configurations. These results highlight the necessity and efficiency of DALI's context encoder in enabling robust generalization.

# 6 Experiments and Analysis

In Sections 6.1 and 6.2, we evaluate DALI's ability to generalize in a zero-shot manner across unseen context variations. In Section 6.3, we show that the dynamics-aligned context encoder learns a structured latent representation, where perturbations to individual dimensions produce physically plausible counterfactuals (e.g., shorter ball swings for higher imagined gravity). Our code is available at `https://github.com/frankroeder/DALI`.

## 6.1 Zero-Shot Generalization

We evaluate DALI's zero-shot generalization on contextualized DMC Ball-in-Cup and Walker Walk tasks from the CARL benchmark [Benjamins et al., 2023].

**Methods.** Our *context-unaware* methods, denoted DALI-S/D-$\chi$/blank, employ either Shallow (S) or Deep (D) integration with the inferred context $\mathfrak{z}_t$ (see Section 4.3). These models are trained using either the forward dynamics loss alone (1) (denoted as "blank", e.g., DALI-D), or the cross-modal regularized objective (2) (denoted as $\chi$, e.g., DALI-S-$\chi$). We compare our models to the context-unaware baseline Dreamer-DR, which corresponds to DreamerV3 with domain randomization [Tobin et al., 2017]. We also evaluate against the *context-aware* baselines cRSSM-S and cRSSM-D, which use the same world model and actor-critic architecture but directly incorporate the ground-truth context $c$ [Prasanna et al., 2024]. For further details, see Appendix A.1.

**Setup.** We train our methods using the *small* variant of DreamerV3 [Hafner et al., 2025], with a transformer-based context encoder [Vaswani et al., 2017], for 200K timesteps (Ball-in-Cup) or 500K timesteps (Walker) across 10 random seeds, following the setup of Prasanna et al. [2024]. Hyperparameters and architectural details are provided in Appendix C.

We conduct evaluation on two observation modalities: *Featurized*, which provides structured state inputs with low partial observability, and *Pixel*, which requires latent state estimation from raw images. We assess performance under three generalization regimes [Kirk et al., 2023]: *Interpolation* (contexts within the training range), *Extrapolation* (OOD contexts beyond the training range), and *Mixed* (one context OOD, one within training range). The Extrapolation regime tests generalization beyond seen contexts, requiring agents to capture underlying physical principles. The Mixed regime probes the ability to interpolate and recombine learned representations. Together, these settings reveal whether the agent learns meaningful abstractions or simply memorizes specific contexts.

**Context ranges.** For Ball-in-Cup, the context parameters are gravity (training: $[4.9, 14.7]$, evaluation: $[0.98, 4.9] \cup (14.7, 19.6]$, default: 9.81) and string length (training: $[0.15, 0.45]$, evaluation: $[0.03, 0.15) \cup (0.45, 0.6]$, default: 0.3). For Walker, the parameters are gravity (same ranges as Ball-in-Cup) and actuator strength (training: $[0.5, 1.5]$, evaluation: $[0.1, 0.5) \cup (1.5, 2.0]$, default: 1.0). These OOD ranges, particularly in Ball-in-Cup, pose major generalization challenges, while Walker's actuator strength remains closer to training. Full training settings, including single and dual context variations, are in Appendix C.

**Evaluation Metrics.** To evaluate zero-shot generalization, we use the Interquartile Mean (IQM) and Probability of Improvement (PoI) metrics from the rliable framework [Agarwal et al., 2021]. IQM averages the central 50% of performance scores, reducing the impact of outliers prevalent in RL due to high variance in training dynamics, particularly in OOD cMDP contexts (e.g., extreme gravity in Ball-in-Cup). We ensure statistical reliability by computing stratified bootstrap 95% confidence intervals over seeds and aggregated contexts. PoI quantifies the probability that a randomly selected run of one algorithm outperforms a randomly selected run of another, offering a robust comparative metric across methods and modalities.

**Results.** We report IQM and PoI across 10 seeds in Figure 1, comparing DALI-S, DALI-S-$\chi$, Dreamer-DR, cRSSM-S, and cRSSM-D for Featurized and Pixel observations on the Ball-in-Cup and Walker Walk tasks.

In Ball-in-Cup's Interpolation, all methods achieve high IQM (0.92–0.95), with DALI-S-$\chi$ competitive (0.9490 Featurized, 0.9440 Pixel). In Extrapolation, DALI-S-$\chi$ excels, outperforming Dreamer-DR by $87.9\%$ (Featurized, IQM 0.3720) to $96.4\%$ (Pixel, IQM 0.2730), surpassing cRSSM-S by $63.9\%$ (Featurized) and $45.9\%$ (Pixel), and cRSSM-D by $33.8\%$ (Featurized) and $12.8\%$ (Pixel). This suggests ground-truth context may overfit, limiting OOD adaptability. Low absolute IQM scores

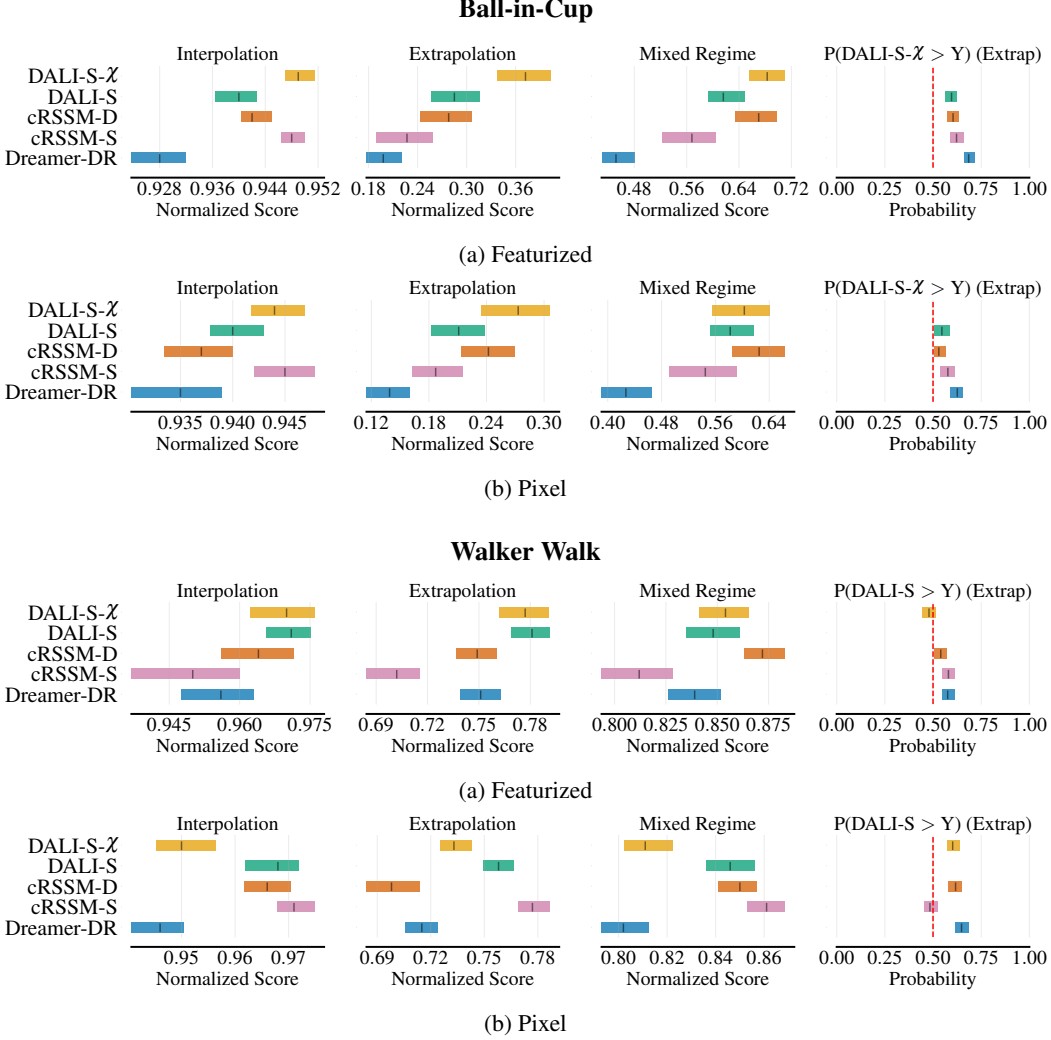

Figure 1: Interquartile Mean (IQM) scores and Probability of Improvement (PoI) for the DMC Ball-in-Cup and Walker Walk tasks under contextual variations: gravity and string length for Ball-in-Cup, and gravity and actuator strength for Walker Walk. Results are shown for Featurized and Pixel observations in each environment. Scores aggregate across single and combined contexts. Shaded intervals represent 95% stratified bootstrap confidence intervals over seeds and aggregated contexts. The rightmost panel in each plot displays PoI in the Extrapolation regime for DALI-S-$\mathcal{X}$ (Ball-in-Cup) and DALI-S (Walker Walk), relative to baseline methods.

(0.14–0.37) reflect the extreme OOD context ranges, such as gravity values of 0.98 or 19.6, far from the training range of $[4.9, 14.7]$. In the Mixed regime, DALI-S-$\mathcal{X}$ leads in Featurized modality (51.1% over Dreamer-DR, 20.3% over cRSSM-S, 1.9% over cRSSM-D, IQM 0.6830), while cRSSM-D excels in Pixel (IQM 0.6250), leveraging ground-truth context for visual dynamics. DALI-S-$\mathcal{X}$ shows moderate PoI ($> 0.6$) against Dreamer-DR in both modalities for Extrapolation (see Figure 1).

In Walker's Interpolation, methods score high (0.94–0.97), with DALI-S leading (0.9710 Featurized). In Extrapolation, DALI-S achieves 4.0% over Dreamer-DR (Featurized, IQM 0.7810), 11.3% over cRSSM-S, and 4.3% over cRSSM-D. Higher IQM scores (0.7–0.78) across methods reflect less extreme OOD actuator strength ($[0.1, 2.0]$ vs. $[0.5, 1.5]$), reducing generalization demands. In Pixel modality, context-aware cRSSM-S leads (IQM 0.7770), leveraging ground-truth context for robust visual feature extraction. DALI-S attains an IQM of 0.7580, compared to DALI-S-$\mathcal{X}$ at 0.7330. In the Mixed regime, context-aware cRSSM-D leads (Featurized, IQM 0.8720), with DALI methods gaining

1.1%–1.8% over Dreamer-DR; in Pixel, cRSSM-S leads (0.8610), with DALI gains 1.1%–5.5%. In Pixel Extrapolation, DALI-S shows moderate PoI ($\sim 0.6$) against Dreamer-DR and cRSSM-D, while cRSSM-S leads, leveraging ground-truth context for visual dynamics (see Figure 1).

## 6.2 Generalization Trends Across Environments and Modalities

We provide additional interpretations of the IQM results in Figure 1, focusing on the generalization patterns of DALI-S and DALI-S-$\mathcal{X}$. We emphasize modality-specific effects, environmental nuances, integration strategies, and comparisons with context-aware baselines (cRSSM-S, cRSSM-D), highlighting the role of cross-modal regularization.

**Cross-Modal Regularization Effects.** DALI-S-$\mathcal{X}$ consistently outperforms DALI-S in Ball-in-Cup across all regimes and modalities, indicating that cross-modal regularization ($L_{\text{cross}}$, see 2) enhances the context encoder's ability to capture pendulum-like dynamics. Aligning $\mathfrak{z}_t$ with the world model's posterior $z_t$ enhances context inference for nonlinear dynamics, benefiting from low partial observability in Featurized inputs and stabilizing inference in Pixel modality's noisy visual inputs. In contrast, DALI-S outperforms DALI-S-$\mathcal{X}$ in Walker across most regimes and modalities, indicating that the forward dynamics loss ($L_{\text{FD}}$, see 1) alone suffices for contexts where actuator strength linearly scales joint torques. By simply optimizing for next-step predictions, DALI-S effectively handles predictable torque scaling, particularly in the Pixel modality where visual noise may amplify $L_{\text{cross}}$'s complexity in DALI-S-$\mathcal{X}$. This underscores the need for task-specific regularization. Ball-in-Cup exhibits larger performance drops from Featurized to Pixel modalities (e.g., DALI-S-$\mathcal{X}$: 0.3720 to 0.2730 in Extrapolation) compared to Walker, reflecting higher partial observability in visual settings with nonlinear dynamics.

**Comparison with Context-Aware Baselines.** DALI-S-$\mathcal{X}$ outperforms cRSSM-S and cRSSM-D in Ball-in-Cup Featurized Extrapolation (0.3720 vs. 0.2270, 0.2780) and Pixel Extrapolation (0.2730 vs. 0.1870, 0.2420), indicating that $\mathfrak{z}_t$'s inferred context generalizes better than ground-truth $c$, which tends to overfit to training contexts. In Walker Featurized Extrapolation, both DALI-S and DALI-S-$\mathcal{X}$ surpass cRSSM-S and cRSSM-D (0.7810, 0.7770 vs. 0.7020, 0.7490), benefiting from $\mathfrak{z}_t$'s inference of actuator torque scaling. However, in Ball-in-Cup Pixel Mixed, cRSSM-D (0.6250) outperforms DALI-S-$\mathcal{X}$ (0.6030), indicating that Deep Integration of ground-truth $c$ better supports modeling of complex visual dynamics. Similarly, in Walker Pixel Extrapolation, cRSSM-S (0.7770) surpasses DALI-S (0.7580), leveraging ground-truth context for precise visual modeling.

**Shallow Context Propagation as Regularization.** In our experiments, Shallow Integration, as implemented in DALI-S and DALI-S-$\mathcal{X}$, consistently performs well, leading to its exclusive use in the results reported here. Shallow Integration incorporates the inferred context $\mathfrak{z}_t$ solely in the world model's encoder, $z_t \sim q_\theta(z_t|h_t, o_t, \mathfrak{z}_t)$, allowing context information to propagate indirectly to the recurrent state $h_t = f_\theta(h_{t-1}, z_{t-1}, a_{t-1})$ through recurrence. This design regularizes the world model, potentially mitigating overfitting to noisy $\mathfrak{z}_t$ estimates, which can be particularly beneficial in OOD settings. The empirical effectiveness of Shallow Integration in our experiments suggests that implicit context propagation can effectively leverage the $\beta$-mixing property in practice.

## 6.3 Validating Physically Consistent Counterfactuals

We assess whether structured perturbations to the latent space $\mathfrak{z}$ induce counterfactual trajectories that adhere to Newtonian physics in the Ball-in-Cup task, validating that $\mathfrak{z}$ encodes mechanistic factors (e.g., gravity, string length) critical for generalization. For this task, the agent must swing a ball into a cup under variable gravity and string length. In our setup, the learned context representation $\mathfrak{z}$ is an 8-dimensional vector, i.e., $\mathfrak{z} \in \mathbb{R}^8$, produced by the dynamics-aligned encoder. Our goal is to identify which latent dimensions in $\mathfrak{z}$ dominantly encode these mechanistic factors.

For each latent dimension $\mathfrak{z}_j$, $j = 1, \ldots, 8$, we sample a fixed observation $o_t$ from a test episode and use the learned context encoder $g_\varphi$ to infer the representation $\mathfrak{z}_t = g_\varphi(o_{t-K:t}, a_{t-K:t-1})$ with $K = 50$. Encoding $o_t$ into the latent state $s_t = \{h_t, z_t\}$, we then roll out actions $a_{t:t+H}$ under both the original $\mathfrak{z}$ and a perturbed $\mathfrak{z}' = \mathfrak{z} + \Delta \cdot \mathbf{e}_j$, where $\mathbf{e}_j$ is a one-hot vector, and $\Delta$ is the standard deviation of $\mathfrak{z}_j$ across the dataset. Using the frozen world model and policy, we decode the predicted observation sequences, yielding two diverging trajectories: baseline trajectory under the original $\mathfrak{z}$, $\mathcal{T}^{(0)} = \{\hat{o}_t, \hat{o}_{t+1}, ..., \hat{o}_{t+H}\}$, and counterfactual trajectory under the perturbed $\mathfrak{z}'$, $\mathcal{T}'^{(j)} = \{\hat{o}'_t, \hat{o}'_{t+1}, ..., \hat{o}'_{t+H}\}$ with $H = 50$. We repeat this process $N = 2500$ times to generate 2500

baseline trajectories and 2500 perturbed trajectories for each $\mathfrak{z}_j$, yielding a total of 5000 trajectory samples: $\mathcal{D}_{cf}^{(j)} = \{\mathcal{T}_1^{(0)}, \mathcal{T}_1'^{(j)}, \ldots, \mathcal{T}_{2500}^{(0)}, \mathcal{T}_{2500}'^{(j)}\}$ for $\mathfrak{z}_j$. We train a binary classifier $c_\nu^{(j)}$ to distinguish between perturbed trajectories $\mathcal{T}'^{(j)}$ (label 1) and baseline trajectories $\mathcal{T}^{(0)}$ (label 0). The classifier outputs the probability $p_n = P(\text{label} = 1|\mathcal{T}_n)$ that a given trajectory $\mathcal{T}_n \in \mathcal{D}_{cf}^{(j)}$ is perturbed. High classifier accuracy for a given dimension suggests it captures mechanistic relationships rather than superficial correlations, with perturbations inducing systematic and physically consistent changes.

We compute 95% bootstrap confidence intervals (CIs) for the classifier AUC for each latent dimension $\mathfrak{z}_j$ and rank the 8 dimensions based on the AUCs, and select the top dimension of $\mathfrak{z}$ for validating physically consistent counterfactuals. See Appendix D for details on the classifier implementation and additional experiments on physically consistent counterfactuals.

### 6.3.1 Mechanistic Alignment in Latent Imagination: Gravity-String Dynamics

We demonstrate DALI's capacity for physically consistent latent imagination by analyzing counterfactual trajectories generated through perturbations to the top-ranked latent dimension ($\mathfrak{z}_6$) in DALI-S-$\chi$ (Shallow Integration with cross-modal regularization). We show how $\mathfrak{z}_6$ encodes coupled gravity-string dynamics in the Ball-in-Cup task.

Our analysis reveals consistent physical behavior across both Featurized and Pixel-based modalities. We demonstrate systematic alignment between the original imagined trajectories and their counterfactual counterparts generated by perturbing the latent context representation in a trained world model. For the Featurized experiments, we tracked predefined state variables (e.g., ball position and velocity), while in the Pixel experiments, we captured rendered environment frames at 5-step intervals over a 64-step imagination horizon (excluding the final frame for visual clarity).

Leveraging our AUC-based ranking analysis, we identify $\mathfrak{z}_6$ as the dominant latent dimension across modalities. To isolate the influence of this dimension on passive dynamics, we generate counterfactual trajectories by perturbing $\mathfrak{z}_6$ and enforce zero-action rollouts. This suppresses policy-driven behavior, exposing the system's intrinsic swinging dynamics.

**Pixel Modality.** Figure 2a contrasts the original (top) and perturbed (middle) imagined trajectories. The perturbed rollout (initialized from the same observation) reveals two key effects: (1) Shorter string length: The blue ball (counterfactual) hovers higher than the original (yellow) in frame 40, indicating a shorter string, and (2) Higher acceleration: The counterfactual ball overtakes the original in frame 15 and 45, demonstrating faster swing cycles due to increased gravitational pull. These observations align with Newtonian mechanics: shorter strings reduce string length, increasing oscillation frequency, while higher gravity amplifies acceleration.

**Featurized Modality.** Figures 2b and 2c quantify these effects. The counterfactual trajectory (orange) in Figure 2b exhibits reduced amplitude (lower peak Z-position) compared to the original (blue), consistent with the pixel-based evidence of a shorter string. This alignment confirms that perturbing shortens the string length, physically constraining the ball's swing. In Figure 2c, the counterfactual's velocity peaks earlier and higher, confirming faster acceleration under perturbed $\mathfrak{z}_6$. Both modalities reveal that $\mathfrak{z}_6$ encodes a coupled relationship between gravity and string length, as a perturbation simultaneously alters both parameters in a physically plausible manner.

## 7 Discussion and Outlook

We introduced Dynamics-Aligned Latent Imagination (DALI), a framework built on DreamerV3 that enables zero-shot generalization by inferring latent contexts from interaction histories. In the Extrapolation regime, our methods DALI-S and DALI-S-$\chi$ achieve substantial performance gains over the context-unaware Dreamer-DR baseline, ranging from $+4.0\%$ to $+96.4\%$ across Featurized and Pixel modalities in the DMC Ball-in-Cup and Walker Walk environments. Notably, DALI-S-$\chi$ surpasses the ground-truth context-aware baselines cRSSM-S by $+45.9\%$ to $+63.9\%$ and cRSSM-D by $+12.8\%$ to $+33.8\%$ in Ball-in-Cup across both modalities. Additionally, DALI-S outperforms cRSSM-S by $+11.3\%$ and cRSSM-D by $+4.3\%$ in Walker's Featurized modality, demonstrating effective zero-shot generalization to unseen contexts.

**Limitations for DALI's Context Inference.** Theorem 2 demonstrates that DALI's context encoder achieves near-optimal context inference, $\mathcal{I}(c; \mathfrak{z}_t) \geq (1 - \delta)h(c)$ using short windows of length

$K = \Omega(\log(1/\delta)/\lambda)$, yielding a sample complexity gain of $\mathcal{O}(T/K)$ over DreamerV3, assuming $\beta$-mixing and Lipschitz dynamics. As an information-theoretic result, it does not address the downstream impact of $\mathfrak{z}_t$ on policy performance, which depends on joint optimization of the context encoder, world model, and actor-critic components. Reliance on an exploratory policy may falter in sparse-reward or high-dimensional settings, producing noisy estimates. Furthermore, the $\beta$-mixing assumption may not hold in environments with slow-mixing dynamics, such as highly correlated trajectories or restricted exploration, potentially limiting generalization. Practical challenges, including training instability, sensitivity to hyperparameters, and the risk of overfitting in high-dimensional observation spaces, are also beyond its scope. Future theoretical work should model how context representations influence policy learning and robustness in complex RL environments.

**Integration Strategies and Practical Implications.** Empirically, DALI's Shallow Integration strategy, where $\mathfrak{z}_t$ is appended only to the encoder input, consistently delivers strong performance in the tasks we evaluated, particularly under OOD conditions. Its simplicity acts as a regularizer, implicitly propagating $\mathfrak{z}_t$ through recurrence and mitigating overfitting, yielding gains of up to $+87.9\%$ (Featurized) and $+96.4\%$ (Pixel) over Dreamer-DR. Future work could investigate the regularization effects of Shallow Integration, explore hybrid strategies interpolating between Shallow and Deep, and develop theoretical models connecting context inference to policy performance. These directions would deepen our understanding of DALI's integration mechanisms and their practical impact on generalization.

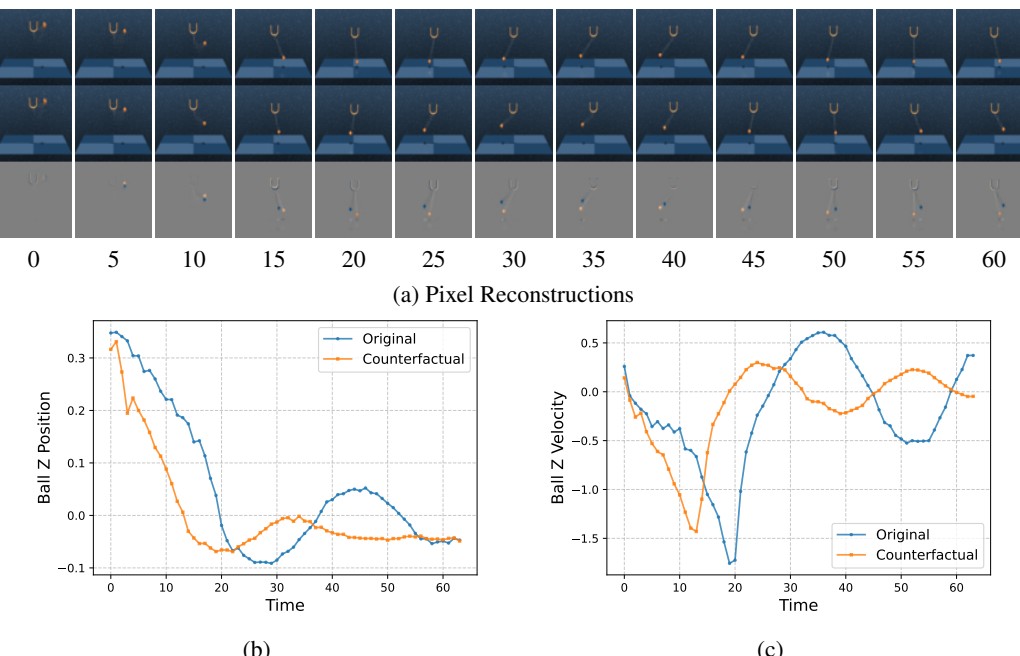

(a) Pixel Reconstructions

(b)

(c)

Figure 2: (a) **(Pixel Modality) Counterfactual Trajectories in Pixel Space**: Top: Original imagined trajectory of the Ball-in-Cup under default gravity and string length. Middle: Perturbed trajectory after adding noise $\Delta$ to the top-ranked latent dimension $\mathfrak{z}_6$. Bottom: Pixel-wise differences ($\delta = |\hat{o}_t - \hat{o}'_t|$). The perturbed trajectory (blue) exhibits a shorter string (frame 40) and faster acceleration (overtaking the original trajectory in frames 15 and 45), aligning with increased gravitational effects. Rollouts use zero actions to isolate passive dynamics. (b) **(Featurized Modality) Ball Z-Position Under Latent Perturbation**: Comparison of original (blue) and counterfactual (orange) ball height trajectories. The perturbed $\mathfrak{z}_6$ reduces oscillation amplitude (lower peak Z-position) and accelerates descent, consistent with shorter string length and higher gravity. (c) **(Featurized Modality) Ball Z-Velocity Under Latent Perturbation**: Velocity profiles for original (blue) and counterfactual (orange) trajectories. The perturbed $\mathfrak{z}_6$ induces earlier and higher velocity peaks, confirming faster swing dynamics. This mirrors the pixel-based evidence of increased gravitational acceleration.

## Acknowledgments and Disclosure of Funding

The authors thank the anonymous reviewers for their valuable feedback and the NeurIPS community for fostering open and rigorous scientific exchange. JB and ME gratefully acknowledge funding by the German Research Foundation DFG through the MoReSpace (402776968) project.

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

## A  Formal Results and Proofs

Theorems 2 and 5 provide formal underpinnings for Theorem 1 in the main text, establishing the necessity and efficiency of DALI's context encoder with Deep Integration (see Section 4.3) compared to DreamerV3 with domain randomization in cMDPs, focusing on context identifiability, sample complexity, and information bottleneck reduction.

Theorem 2 compares DALI with Deep Integration, using context encoder $\mathfrak{z}_t$, against DreamerV3 with domain randomization, using recurrent state $h_t$, in a cMDP. The theorem establishes: (1) DALI's $\mathfrak{z}_t$ captures at least as much mutual information with context $c$ as DreamerV3's $h_t$, up to an error $\epsilon(K) = \mathcal{O}(e^{-\lambda K/2})h(c)$ that decays exponentially with window size $K$; (2) DALI achieves near-optimal $\mathcal{I}(c; \mathfrak{z}_t) \geq (1 - \delta)h(c)$ with $N = \mathcal{O}(1/\delta^2)$ samples of $K = \Omega(\log(1/\delta)/\lambda)$ transitions, leveraging shorter windows compared to DreamerV3's full episodes of length $T$. This demonstrates the necessity and efficiency of DALI's context encoder for context identifiability and learning.

**Theorem 2** (Necessity and efficiency of Context encoder). *Consider a cMDP $\mathcal{M} = (\mathcal{S}, \mathcal{A}, \mathcal{O}, \mathcal{C}, \mathcal{R}, \mathcal{P}, \mathcal{E}, \mu, p_\mathcal{C})$ with:*

- *Continuous context $c \in \mathcal{C} \subseteq \mathbb{R}^d$, static within episodes, drawn i.i.d. as $c \sim p_\mathcal{C}$, with $h(c) < \infty$.*

- *Lipschitz dynamics: $\exists L > 0$ such that $\forall s, s' \in \mathcal{S}, a \in \mathcal{A}, c, c' \in \mathcal{C}$,*

$$\|\mathcal{P}_c(s'|s,a) - \mathcal{P}_{c'}(s'|s,a)\|_1 \leq L\|c - c'\|_2.$$

- *Observation noise: The observation function $\mathcal{E}$ satisfies $o_t \sim \mathcal{N}(s_t, \sigma^2 I)$, with fixed variance $\sigma^2 > 0$.*

- *$\beta$-mixing: The process $\tau_t = (o_t, a_t)$ is $\beta$-mixing with*

$$\beta(K) = \sup_t \sup_{A \in \sigma(\tau_s, s \leq t), B \in \sigma(\tau_s, s \geq t+K)} |P(A \cap B) - P(A)P(B)| \leq Ce^{-\lambda K},$$

*where $\sigma(\tau_s, s \leq t)$ denotes the sigma-algebra generated by $\{\tau_s : s \leq t\}$, and similarly for $\sigma(\tau_s, s \geq t + K)$, for constants $C, \lambda > 0$.*

- *Sufficient exploration: The policy ensures that for any $c \neq c' \in \mathcal{C}$, there exists a constant $\alpha > 0$ such that $\mathrm{KL}(p(o_{t-K:t}, a_{t-K:t-1}|c)\|p(o_{t-K:t}, a_{t-K:t-1}|c')) \geq \alpha K$.*

- *Bounded log-likelihood ratios: $\left|\log \frac{p(o_i, a_i|c', \tau_{t-K:i-1})}{p(o_i, a_i|c, \tau_{t-K:i-1})}\right| \leq M$, ensured by restricting observations and actions to compact sets or imposing suitable policy constraints.*

- *Universal approximator: DALI's $g_\varphi$ (neural network) approximates any continuous function on compact domains.*

- *Training data: Both models are trained on trajectories from $\mathcal{M}$, with contexts $c \sim p_\mathcal{C}$ i.i.d. per episode, using an exploratory policy. DreamerV3 with domain randomization trains its RSSM on episodes $(o_{1:T}, a_{1:T-1})$ of length $T$. DALI trains its context encoder $g_\varphi$ on $K$-length windows $(o_{t-K:t}, a_{t-K:t-1})$ with loss $L_{FD} = \mathbb{E}[\|o_{t+1} - f_\varphi^w(o_t, a_t, \mathfrak{z}_t)\|_2^2]$, and its RSSM on full episodes, conditioning on $\mathfrak{z}_t$.*

- *Both models use a GRU $f_\theta$ with finite parameters for their recurrent state updates.*

*Let $h_t = f_\theta(h_{t-1}, z_{t-1}, a_{t-1})$ be DreamerV3's recurrent state, with $z_t \sim q_\theta(z_t|h_t, o_t)$, and $\mathfrak{z}_t = g_\varphi(o_{t-K:t}, a_{t-K:t-1})$ DALI's context encoding, with $h_t = f_\theta(h_{t-1}, z_{t-1}, a_{t-1}, \mathfrak{z}_t)$. Then:*

1. **Context identifiability**: *For any $K$, there exists $g_\varphi$ such that, within an episode:*

$$\mathcal{I}(c; \mathfrak{z}_t) \geq \mathcal{I}(c; h_t) - \epsilon(K), \quad \epsilon(K) = \mathcal{O}(e^{-\lambda K/2})h(c).$$

2. **Sample complexity**: *For $K = \Omega\left(\frac{\log(1/\delta)}{\lambda}\right)$ and $\delta \in (0,1)$, DALI achieves*

$$\mathcal{I}(c; \mathfrak{z}_t) \geq (1-\delta)h(c)$$

*with $N = \mathcal{O}\left(\frac{1}{\delta^2}\right)$ samples of $K$ transitions, vs. a hypothetical DreamerV3 context estimator $g_\psi(o_{1:T}, a_{1:T-1})$ requiring $N = \mathcal{O}\left(\frac{1}{\delta^2}\right)$ samples of $T$ transitions.*

**Remark 3** (Non-trivial sample complexity gain of DALI's Context encoder). *The sample complexity gain of $\mathcal{O}(T/K)$ for DALI over DreamerV3 with domain randomization, as shown in Theorem 2, arises from DALI's use of $K$-length windows versus DreamerV3's full episodes of length $T$. While this gain appears to scale with input lengths, it is non-trivial: Both models achieve $N = \mathcal{O}(1/\delta^2)$ samples, but DALI's window size $K = \Omega(\log(1/\delta)/\lambda)$ leverages the $\beta$-mixing property (Lemma 6) to ensure $h(c|\tau_{t-K:t}) \leq C'e^{-\lambda K}h(c)$, enabling efficient context identification. In contrast, DreamerV3 relies on longer trajectories ($T \gg K$) to achieve comparable context identification. The gain reflects DALI's context encoder exploiting short, local histories determined by the mixing rate $\lambda$ and desired accuracy $\delta$.*

**Remark 4** (Consistency of DALI's sample complexity across integration strategies). *The sample complexity advantage of DALI established in Theorem 2 for Deep Integration extends to Shallow Integration (Section 4.3), since the context encoder $\mathfrak{z}_t$ is trained identically on $K$-length windows in both configurations. Incorporating the cross-modal loss $L_{cross}$ in (2) aligns $\mathfrak{z}_t$ with $z_t$, which may allow the context representation $\mathfrak{z}_t$ to leverage structured priors from the world model, potentially enhancing its robustness. This addition does not affect sample complexity, as $L_{cross}$ operates on the same $K$-length windows (Algorithm 2). Recurrent unrolls are limited to $K$ steps, and $\varphi$ updates depend solely on the current window, with detached states blocking backpropagation beyond it. The world model parameters ($\theta$) are frozen during context learning (inputs to $f_\theta$ and $q_\theta$ are detached), so gradient stopping isolates $\varphi$ and preserves the $\mathcal{O}(T/K)$ efficiency.*

Theorem 5 proves that DALI's sequence model, incorporating a context encoder $\mathfrak{z}_t$, reduces the information bottleneck in cMDPs compared to DreamerV3's RSSM. Specifically, it shows that DALI's recurrent state retains more context information, satisfying $\mathcal{I}(c; h_t^{\text{DALI}}) \geq \mathcal{I}(c; h_t^{\text{RSSM}}) - \epsilon(K)$, where $\epsilon(K) = \mathcal{O}(e^{-\lambda K/2})h(c)$, leveraging the encoder's ability to efficiently capture context over short windows.

**Theorem 5** (Context encoder reduces information bottleneck). *Consider a cMDP $\mathcal{M} = (\mathcal{S}, \mathcal{A}, \mathcal{O}, \mathcal{C}, \mathcal{R}, \mathcal{P}, \mathcal{E}, \mu, p_\mathcal{C})$ with:*

- *Continuous context $c \in \mathcal{C} \subseteq \mathbb{R}^d$, static within episodes, drawn i.i.d. as $c \sim p_\mathcal{C}$, with $h(c) < \infty$.*

- *Observations $o_t = s_t + \eta_t$, $\eta_t \sim \mathcal{N}(0, \sigma^2 I)$, $\sigma^2 > 0$.*

- *Lipschitz dynamics: $\exists L > 0$ such that $\forall s, s' \in \mathcal{S}, a \in \mathcal{A}, c, c' \in \mathcal{C}$,*

$$||\mathcal{P}_c(s'|s,a) - \mathcal{P}_{c'}(s'|s,a)||_1 \leq L||c - c'||_2.$$

- *$\beta$-mixing with $\beta(K) \leq Ce^{-\lambda K}$, exploratory policy with $\text{KL}(p(\tau_{t-K:t}|c)\|p(\tau_{t-K:t}|c')) \geq \alpha K$, $\alpha > 0$.*

- *Context encoder $g_\varphi$ is a universal approximator, trained with forward dynamics loss $L_{FD} = \mathbb{E}[||o_{t+1} - f_\varphi^w(o_t, a_t, \mathfrak{z}_t)||_2^2]$, satisfying $h(c|\mathfrak{z}_t) \leq C'e^{-\lambda K}h(c) + \delta'$, $\delta' = \mathcal{O}(e^{-\lambda K/2})$ (Lemma 7).*

- *Both models use a GRU $f_\theta$ with finite parameters.*

*Let $h_t^{RSSM} = f_\theta(h_{t-1}, z_{t-1}, a_{t-1})$ be DreamerV3's recurrent state, with $z_t \sim q_\theta(z_t|h_t, o_t)$, and $h_t^{DALI} = f_\theta(h_{t-1}, z_{t-1}, a_{t-1}, \mathfrak{z}_t)$ be DALI's recurrent state. Then, for any window size $K$, DALI's sequence model satisfies:*

$$\mathcal{I}(c; h_t^{DALI}) \geq \mathcal{I}(c; h_t^{RSSM}) - \epsilon(K),$$

*where $\epsilon(K) = \mathcal{O}(e^{-\lambda K/2})h(c)$, implying that the context encoder $\mathfrak{z}_t$ reduces the information bottleneck compared to DreamerV3's RSSM.*

For $\beta$-mixing sequences with $\beta(K) \leq Ce^{-\lambda K}$, blocks of $K$-separated transitions are approximately independent. Lemma 6 establishes that, in a $\beta$-mixing cMDP, the conditional entropy of the context $c$ given a history $\tau_{t-K:t}$ decays exponentially as $h(c|\tau_{t-K:t}) \leq C'e^{-\lambda K}h(c)$. This result is crucial for Theorems 2 and 5, as it quantifies how historical trajectories reduce uncertainty about the context, enabling DALI's context encoder to achieve near-optimal information capture with short windows, supporting both the identifiability and sample complexity claims.

**Lemma 6** (Entropy decay from mixing). *Consider a cMDP with context $c \sim p_\mathcal{C}$, observation noise $o_t = s_t + \eta_t$, $\eta_t \sim \mathcal{N}(0, \sigma^2 I)$ and history $\tau_{t-K:t} = (o_{t-K:t}, a_{t-K:t-1})$, where the process $\tau_t = (o_t, a_t)$ is $\beta$-mixing with $\beta(K) \leq Ce^{-\lambda K}$ for constants $C, \lambda > 0$, bounding dependence errors in tail probabilities by $\beta(K)$, log-likelihood ratios are bounded as $\left|\log \frac{p(o_i, a_i|c', \tau_{t-K:i-1})}{p(o_i, a_i|c, \tau_{t-K:i-1})}\right| \leq M$, the policy ensures exploration such that for any $c \neq c' \in \mathcal{C}$, there exists a constant $\alpha \geq M\sqrt{2\lambda}$ satisfying $\mathrm{KL}(p(\tau_{t-K:t}|c)|p(\tau_{t-K:t}|c')) \geq \alpha K$, and $c$ has bounded density on $\mathcal{C} \subseteq \mathbb{R}^d$ with differential entropy $h(c) < \infty$. Then, there exists a constant $C' > 0$ such that:*

$$h(c|\tau_{t-K:t}) \leq C'e^{-\lambda K}h(c).$$

*Proof of Lemma 6.* We show that uncertainty about $c$ decreases exponentially with $K$, enabling DALI's $\mathfrak{z}_t$ to infer $c$.

By assumption, the process $\tau_t = (o_t, a_t)$ is $\beta$-mixing, with $\beta(K) \leq Ce^{-\lambda K}$. The condition $\mathrm{KL}(p(\tau_{t-K:t}|c)\|p(\tau_{t-K:t}|c')) \geq \alpha K$, with $\alpha \geq M\sqrt{2\lambda}$, ensures distinct histories. We need $h(c|\tau_{t-K:t}) \leq C'e^{-\lambda K}h(c)$, so we show:

$$h(c|\tau_{t-K:t}) \leq C''\beta(K)h(c).$$

Discretize $\mathcal{C}$ into $N$ points, $\log N \approx h(c)$. Let $\hat{c} = \arg\max_{c'} p(\tau_{t-K:t}|c')$ and let $P_e = P(\hat{c} \neq c|\tau_{t-K:t})$ be the probability of incorrectly estimating $c$. Fano's inequality gives [Thomas and Cover, 1999]:

$$h(c|\tau_{t-K:t}) \leq h(P_e) + P_e h(c). \tag{4}$$

Error $P_e = P(\hat{c} \neq c|c)$ occurs if there exists any $c' \neq c$ such that $p(\tau_{t-K:t}|c') > p(\tau_{t-K:t}|c)$. By the union bound:

$$P_e = P\left(\bigcup_{c' \neq c}\{p(\tau_{t-K:t}|c') > p(\tau_{t-K:t}|c)\}\Big|c\right) \leq \sum_{c' \neq c} P(p(\tau_{t-K:t}|c') > p(\tau_{t-K:t}|c)|c).$$

For $c' \neq c$, define: $Z = \log \frac{p(\tau_{t-K:t}|c')}{p(\tau_{t-K:t}|c)}$. Then $Z > 0$ corresponds to the maximum likelihood estimator $\hat{c}$ incorrectly favoring $c'$ over $c$. The probability of this error under context $c$ is $P(Z > 0|c)$. By the exploration assumption:

$$\mathbb{E}_c[Z] = -\mathrm{KL}(p(\tau_{t-K:t}|c) \| p(\tau_{t-K:t}|c')) \leq -\alpha K.$$

By the chain rule in the cMDP, $p(\tau_{t-K:t}|c) = \prod_{i=t-K}^{t-1} p(o_i, a_i|\tau_{t-K:i-1}, c) \cdot p(o_t|\tau_{t-K:t-1}, c)$, where $\tau_{t-K:i-1} = (o_{t-K:i-1}, a_{t-K:i-1})$.

Thus, $Z = \sum_{i=t-K}^{t-1} \log \frac{p(o_i, a_i|\tau_{t-K:i-1}, c')}{p(o_i, a_i|\tau_{t-K:i-1}, c)} + \log \frac{p(o_t|\tau_{t-K:t-1}, c')}{p(o_t|\tau_{t-K:t-1}, c)}$.

Define $Z_i = \log \frac{p(o_i, a_i|\tau_{t-K:i-1}, c')}{p(o_i, a_i|\tau_{t-K:i-1}, c)}$ for $i = t - K$ to $t - 1$, and $Z_t = \log \frac{p(o_t|\tau_{t-K:t-1}, c')}{p(o_t|\tau_{t-K:t-1}, c)}$. So $Z = \sum_{i=t-K}^{t} Z_i$. $Z_i$ measures how much more (or less) likely $(o_i, a_i)$ is under context $c'$ compared to $c$, given past observations and actions. By the bounded log-likelihood assumption, $|Z_i| \leq M$. For independent $Z_i$, Hoeffding's inequality [Boucheron et al., 2013] bounds the tail probability for bounded random variables:

$$\mathbb{P}(Z - \mathbb{E}_c[Z] \geq \alpha K) \leq \exp\left(-\frac{2(\alpha K)^2}{K(2M)^2}\right) = \exp\left(-\frac{\alpha^2 K}{2M^2}\right).$$

Since $\tau_t$ is $\beta$-mixing with $\beta(K) \leq Ce^{-\lambda K}$, the dependence adds an error $\beta(K)$, as $\beta(K)$ bounds the deviation from independence between events separated by $K$ time steps:

$$P(Z - \mathbb{E}_c[Z] \geq \alpha K) \leq \exp\left(-\frac{\alpha^2 K}{2M^2}\right) + \beta(K).$$

Set $\kappa = \frac{\alpha^2}{2M^2}$. Since $\alpha \geq M\sqrt{2\lambda}$, we have $\kappa \geq \lambda$, so $\exp(-\kappa K) \leq e^{-\lambda K}$. Thus:

$$P(Z > 0|c) \leq \exp(-\kappa K) + \beta(K) \leq \exp(-\lambda K) + Ce^{-\lambda K} = (1+C)e^{-\lambda K}.$$

Sum over $N-1$ contexts:

$$P_e \leq (N-1)(1+C)e^{-\lambda K} = C_1\beta(K), \quad C_1 = \frac{(N-1)(1+C)}{C}.$$

Apply to (4):
$$h(c|\tau_{t-K:t}) \leq h(C_1\beta(K)) + C_1\beta(K)h(c).$$

For small $\beta(K)$, $h(C_1\beta(K)) \approx 0$, so:

$$h(c|\tau_{t-K:t}) \leq C_1\beta(K)h(c).$$

With $\beta(K) \leq Ce^{-\lambda K}$,

$$h(c|\tau_{t-K:t}) \leq C_1 \cdot Ce^{-\lambda K}h(c) = C'e^{-\lambda K}h(c).$$

$\square$

Lemma 7 provides a generalization bound for DALI's context encoder, showing that the conditional entropy $h(c|\mathfrak{z}_t)$ is bounded by the entropy given the history plus a small error, $h(c|\mathfrak{z}_t) \leq h(c|\tau_{t-K:t}) + \delta'$, with $\delta' = \mathcal{O}(\sqrt{\delta})$ and $\delta = Ce^{-\lambda K}$. This lemma is pivotal for Theorems 2 and 5, as it ensures that the encoder $\mathfrak{z}_t$ effectively captures context information with a sample complexity of $N = \mathcal{O}(1/\delta^2)$, facilitating DALI's efficiency and reduced bottleneck.

**Lemma 7** (DALI's generalization bound for context inference). *Consider a cMDP satisfying Theorem 2's assumptions: context $c \sim p_{\mathcal{C}}$ with bounded density on $\mathcal{C} \subseteq \mathbb{R}^d$ and finite entropy $h(c) < \infty$, observation noise $o_t = s_t + \eta_t$, $\eta_t \sim \mathcal{N}(0, \sigma^2 I)$, $\beta$-mixing with $\beta(K) \leq Ce^{-\lambda K}$ for constants $C, \lambda > 0$, exploratory policy ensuring $\mathrm{KL}(p(\tau_{t-K:t}|c)\|p(\tau_{t-K:t}|c')) \geq \alpha K$ for some $\alpha > 0$, Lipschitz dynamics with $\|\mathcal{P}_c(s'|s,a) - \mathcal{P}_{c'}(s'|s,a)\|_1 \leq L\|c - c'\|_2$, and universal approximator $g_\varphi$. DALI's context encoding $\mathfrak{z}_t = g_\varphi(o_{t-K:t}, a_{t-K:t-1})$, trained with the forward dynamics loss $L_{FD} = \mathbb{E}[\|o_{t+1} - f_\varphi^w(o_t, a_t, \mathfrak{z}_t)\|_2^2]$, satisfies:*

$$h(c|\mathfrak{z}_t) \leq h(c|\tau_{t-K:t}) + \delta', \quad \delta' = \mathcal{O}(\sqrt{\delta}),$$

*with $N = \mathcal{O}\left(\frac{1}{\delta^2}\right)$ samples, where $\delta = Ce^{-\lambda K}$.*

*Proof of Lemma 7.* The context encoder $g_\varphi$ minimizes the forward dynamics loss:

$$L_{\mathrm{FD}} = \mathbb{E}[\|o_{t+1} - f_\varphi^w(o_t, a_t, \mathfrak{z}_t)\|_2^2], \quad \mathfrak{z}_t = g_\varphi(o_{t-K:t}, a_{t-K:t-1}),$$

where $\tau_{t-K:t} = (o_{t-K:t}, a_{t-K:t-1})$, $o_t = s_t + \eta_t$, and $\eta_t \sim \mathcal{N}(0, \sigma^2 I)$. The empirical loss over $N$ samples is:

$$\hat{L}_{\mathrm{FD}} = \frac{1}{N}\sum_{i=1}^{N} \|o_{i+1} - f_\varphi^w(o_i, a_i, g_\varphi(o_{i-K:i}, a_{i-K:i-1}))\|_2^2.$$

To benchmark $L_{\mathrm{FD}}$, consider an ideal predictor with access to the true context $c$. Since $o_{t+1} = s_{t+1} + \eta_{t+1}$, $\eta_{t+1} \sim \mathcal{N}(0, \sigma^2 I)$, and $o_t = s_t + \eta_t$, the predictor $f(o_t, a_t, c)$ uses noisy observations. The loss is:

$$\mathbb{E}[\|o_{t+1} - f(o_t, a_t, c)\|_2^2] = \mathbb{E}[\|s_{t+1} - f(o_t, a_t, c)\|_2^2] + \mathbb{E}[\|\eta_{t+1}\|_2^2].$$

For $d_s$-dimensional $\eta_{t+1}$, $\mathbb{E}[\|\eta_{t+1}\|_2^2] = d_s\sigma^2$. The ideal predictor $f^*(o_t, a_t, c) = \mathbb{E}[s_{t+1}|o_t, a_t, c]$, the expected next state given $o_t, a_t, c$, minimizes $L_{\mathrm{FD}}$ by setting $f^*$ to the best estimate of $s_{t+1}$, yielding $\mathbb{E}[\|o_{t+1} - f^*(o_t, a_t, c)\|_2^2] = \mathbb{E}[\|s_{t+1} - \mathbb{E}[s_{t+1}|o_t, a_t, c]\|_2^2] + d_s\sigma^2$, where the first term is the variance of $s_{t+1}$ and the second is the noise variance. By the universal approximation property, $f_\varphi^w$ can approximate $f^*$ if $\mathfrak{z}_t$ encodes $c$.

Consider the loss function for fixed $\varphi, w$:

$$l(\varphi, w; o_{i-K:i+1}, a_{i-K:i}) = \|o_{i+1} - f_\varphi^w(o_i, a_i, g_\varphi(o_{i-K:i}, a_{i-K:i-1}))\|_2^2,$$

with $|l| \leq M$ (by bounded $\mathcal{S}, \mathcal{A}$) and $L_l$-Lipschitz in $\varphi, w$. The function class $\mathcal{F} = \{(o_i, a_i, o_{i-K:i}, a_{i-K:i-1}) \mapsto f_\varphi^w(o_i, a_i, g_\varphi(o_{i-K:i}, a_{i-K:i-1}))\}$ has Rademacher complexity $\mathcal{R}_N(\mathcal{F}) \leq \frac{C_\mathcal{F}}{\sqrt{N}}$, so the loss class $\mathcal{L}$ has:

$$\mathcal{R}_N(\mathcal{L}) \leq \frac{C_\mathcal{F} L_l}{\sqrt{N}}.$$

For i.i.d. samples and a fixed $\varphi, w$, the generalization error is bounded by:

$$|\mathbb{E}[\hat{L}_{\text{FD}}] - \hat{L}_{\text{FD}}| \leq 2\mathcal{R}_N(\mathcal{L}) = \frac{2C_\mathcal{F} L_l}{\sqrt{N}}.$$

By McDiarmid's inequality, since changing one sample affects $\hat{L}_{\text{FD}}$ by at most $\frac{2M}{N}$, with probability at least $1 - \delta$ [Mohri et al., 2018]:

$$\mathbb{E}[\hat{L}_{\text{FD}}] \leq \hat{L}_{\text{FD}} + \frac{2C_\mathcal{F} L_l}{\sqrt{N}} + M\sqrt{\frac{2\log(1/\delta)}{N}}.$$

For $\beta$-mixing data with $\beta(K) \leq Ce^{-\lambda K}$, the deviation from i.i.d. adds an error bounded by $Ce^{-\lambda K}$ to the expected loss for sequences spaced by $K$ steps:

$$\mathbb{E}[\hat{L}_{\text{FD}}] \leq \hat{L}_{\text{FD}} + \frac{2C_\mathcal{F} L_l}{\sqrt{N}} + M\sqrt{\frac{2\log(1/\delta)}{N}} + Ce^{-\lambda K}.$$

Set the generalization error to $\epsilon^2 = \mathcal{O}(\delta)$, where $\delta = Ce^{-\lambda K}$, to match the $\beta$-mixing decay. Solve:

$$\frac{2C_\mathcal{F} L_l}{\sqrt{N}} + M\sqrt{\frac{2\log(1/\delta)}{N}} + Ce^{-\lambda K} \leq C''\delta.$$

The first term requires: $\frac{2C_\mathcal{F} L_l}{\sqrt{N}} \leq C''\delta \implies N \geq \left(\frac{2C_\mathcal{F} L_l}{C''\delta}\right)^2 = \mathcal{O}\left(\frac{1}{\delta^2}\right)$.

The second term requires: $M\sqrt{\frac{2\log(1/\delta)}{N}} \leq C''\delta \implies N \geq \frac{2M^2 \log(1/\delta)}{(C''\delta)^2} = \mathcal{O}\left(\frac{\log(1/\delta)}{\delta^2}\right)$.

The third term $Ce^{-\lambda K} = \delta \leq C''\delta$ holds for $C'' \geq 1$.

The first term dominates, so $N = \mathcal{O}\left(\frac{1}{\delta^2}\right)$ suffices.

If $\mathbb{E}[\hat{L}_{\text{FD}}] \leq \mathbb{E}[\|o_{t+1} - f^*(o_t, a_t, c)\|_2^2] + \epsilon^2$, then:

$$\|f_\varphi^w(o_t, a_t, \mathfrak{z}_t) - f^*(o_t, a_t, c)\|_2 \leq \epsilon, \quad \epsilon = \mathcal{O}(\sqrt{\delta}).$$

Given the Markov chain $c \to \tau_{t-K:t} \to \mathfrak{z}_t$, the entropy difference is:

$$h(c|\mathfrak{z}_t) - h(c|\tau_{t-K:t}) \leq \mathbb{E}[\text{KL}(p(c|\tau_{t-K:t})\|p(c|\mathfrak{z}_t))].$$

The cMDP's Lipschitz dynamics, $\|\mathcal{P}_c(s'|s, a) - \mathcal{P}_{c'}(s'|s, a)\|_1 \leq L\|c - c'\|_2$, and $\beta$-mixing ensure $p(c|\tau_{t-K:t})$ is $L_p$-Lipschitz in Wasserstein distance, as small changes in $\tau_{t-K:t}$ reflect proportional changes in $c$. Assume there exists a mapping $\psi : \mathcal{Z} \to \mathcal{C}$ such that the context encoder satisfies:

$$\|\psi(g_\varphi(\tau_{t-K:t})) - c\|_2 \leq L_c \epsilon = \mathcal{O}(\sqrt{\delta}),$$

where $\mathcal{Z}$ is the encoder's output space. Then:

$$\text{KL}(p(c|\tau_{t-K:t})\|p(c|\mathfrak{z}_t)) \leq L_p L_c \epsilon = \mathcal{O}(\sqrt{\delta}).$$

Thus:

$$h(c|\mathfrak{z}_t) \leq h(c|\tau_{t-K:t}) + \mathcal{O}(\sqrt{\delta}).$$

From Lemma 6, $h(c|\tau_{t-K:t}) \leq Ce^{-\lambda K}h(c) = \mathcal{O}(\delta)h(c)$. Since $h(c) < \infty$ and $\sqrt{\delta} \geq \delta$ for small $\delta$, the bound becomes:

$$h(c|\mathfrak{z}_t) \leq \mathcal{O}(\sqrt{\delta}),$$

yielding $\delta' = \mathcal{O}(\sqrt{\delta})$ with $N = \mathcal{O}\left(\frac{1}{\delta^2}\right)$ samples.

$\square$

Lemma 8 demonstrates that DreamerV3's RSSM suffers from a persistent information bottleneck in cMDPs, with the conditional entropy $h(c|h_t) \geq \kappa h(c)$ for a constant $\kappa \in (0,1)$. This result is essential for Theorems 2 and 5, as it establishes a baseline limitation in DreamerV3's ability to retain context information, against which DALI's superior performance in context identifiability and bottleneck reduction is compared.

**Lemma 8** (Information bottleneck in DreamerV3's RSSM). *Consider DreamerV3's RSSM in a cMDP with context $c \sim p_\mathcal{C}$ with bounded density on $\mathcal{C} \subseteq \mathbb{R}^d$ and finite entropy $h(c) < \infty$, observation noise $o_t = s_t + \eta_t$, $\eta_t \sim \mathcal{N}(0, \sigma^2 I)$, Lipschitz dynamics with $\|\mathcal{P}_c(s'|s,a) - \mathcal{P}_{c'}(s'|s,a)\|_1 \leq L\|c - c'\|_2$, and recurrent state $h_t = f_\theta(h_{t-1}, z_{t-1}, a_{t-1})$, where $z_t \sim q_\theta(z_t|h_t, o_t)$, and $f_\theta$ is a GRU with finite parameters. Then, there exists $\kappa \in (0,1)$ such that:*

$$h(c|h_t) \geq \kappa h(c).$$

*Proof of Lemma 8.* The context $c$ determines the cMDP's dynamics $\mathcal{P}_c(s'|s,a)$. Since $h_t$ depends on the trajectory $\tau_{1:t} = (o_{1:t}, a_{1:t-1})$ and latent variables $z_{1:t-1}$, we have a Markov chain $c \to \tau_{1:t} \to h_t$, so $p(h_t|\tau_{1:t}, c) = p(h_t|\tau_{1:t})$. By the data processing inequality (DPI) [Thomas and Cover, 1999]:

$$\mathcal{I}(c; h_t) \leq \mathcal{I}(c; \tau_{1:t}) \leq h(c).$$

Since $o_{1:t} = s_{1:t} + \eta_{1:t}$, and $a_{1:t-1}$ depend on $o_{1:t-1}$, we have:

$$h(c|\tau_{1:t}) = h(c|o_{1:t}, a_{1:t-1}) = h(c|s_{1:t}, \eta_{1:t}) = h(c|s_{1:t}), \quad \text{as } \eta_{1:t} \perp c.$$

The GRU $f_\theta$, with finite parameters and Lipschitz continuity (due to bounded weights and standard GRU operations), produces $h_t \in \mathbb{R}^d$, where $d$ is a fixed dimension. The GRU is trained to maximize the ELBO:

$$\sum_{t=1}^{T} \mathbb{E}_{q_\theta(z_t|h_t, o_t)} \left[ \log p_\theta(o_t, r_t, n_t | h_t, z_t) \right] - \mathrm{KL}(q_\theta(z_t|h_t, o_t) \| p_\theta(z_t|h_t)),$$

which optimizes $h_t$ to predict observations $o_t$, rewards $r_t$, and termination signals $n_t$. The input $\tau_{1:t}$ includes:

- Context information: $c$ affects $s_{1:t}$ through dynamics.

- Noise information: $\eta_{1:t} \sim \mathcal{N}(0, \sigma^2 I_{t \cdot d_s})$, with $d_s$ being the dimension of the state space $\mathcal{S}$. $\eta_{1:t}$ contributes significant entropy, $h(\eta_{1:t}) = t \cdot \frac{d_s}{2} \ln(2\pi e \sigma^2)$, which grows linearly with $t$.

- State transients: $s_{1:t}$ includes dynamic behaviors not directly tied to $c$.

The GRU's finite parameters and fixed dimension $d$ limit its capacity, forcing $h_t$ to compress $\tau_{1:t}$, potentially discarding context information.

Assume $h_t$ is Gaussian with covariance $\Sigma_h$, where $\mathrm{Var}(h_t^i) \leq V$ for $i = 1, \ldots, d$. The variance bound $V$ follows from compact $\mathcal{C}, \mathcal{S}, \mathcal{A}$, Lipschitz dynamics, and the Lipschitz GRU, ensuring bounded outputs. For a $d$-dimensional Gaussian channel with output $h_t \sim \mathcal{N}(\mu_c, \Sigma_h)$, where $\mu_c$ depends on $c$, and noise variance at most $\sigma^2$, the mutual information is bounded by [Thomas and Cover, 1999]:

$$\mathcal{I}(c; h_t) \leq \frac{d}{2} \ln \left( 1 + \frac{\mathrm{tr}(\Sigma_h)}{\sigma^2} \right) \leq \frac{d}{2} \ln \left( 1 + \frac{dV}{\sigma^2} \right).$$

The GRU's compression limits $\mathcal{I}(c; h_t)$ relative to $h(c)$ for all $h(c) \geq 0$, as follows. The ELBO's KL-term $\mathrm{KL}(q_\theta(z_t|h_t, o_t) \| p_\theta(z_t|h_t))$ regularizes $q_\theta(z_t|h_t, o_t)$, reducing the information about $o_t$ (and thus $c$) in $z_t$. Since $h_t$ depends on $z_{1:t-1}$, this limits context information in $h_t$. The GRU's finite parameters and dimension $d$ further constrain capacity. Let the channel capacity be reduced by a factor $\alpha \in (0,1)$, reflecting this compression:

$$\mathcal{I}(c; h_t) \leq \alpha \cdot \frac{d}{2} \ln \left( 1 + \frac{dV}{\sigma^2} \right),$$

where $\alpha < 1$ (e.g., $\alpha = 1 - \epsilon$, with $\epsilon > 0$ depending on the GRU's architecture).

To satisfy the lemma, we seek a constant $\kappa \in (0, 1)$ such that $h(c|h_t) \geq \kappa h(c)$ for all $p_{\mathcal{C}}$. Since $h(c|h_t) = h(c) - \mathcal{I}(c; h_t)$ and $\mathcal{I}(c; h_t) \leq \alpha \cdot \frac{d}{2} \ln \left(1 + \frac{dV}{\sigma^2}\right)$, define:

$$B = \frac{d}{2} \ln \left(1 + \frac{dV}{\sigma^2}\right), \quad \mathcal{I}(c; h_t) \leq \alpha B.$$

For $h(c) > 0$, we want $h(c) - \mathcal{I}(c; h_t) \geq \kappa h(c)$. Using $\mathcal{I}(c; h_t) \leq \alpha B$, this holds if $h(c) - \alpha B \geq \kappa h(c)$, or $\kappa \leq 1 - \frac{\alpha B}{h(c)}$. Thus, define:

$$\kappa = \begin{cases} 1 - \frac{\alpha B}{h(c)} & \text{if } h(c) > 0, \\ \frac{1}{2} & \text{if } h(c) = 0. \end{cases}$$

If $h(c) > 0$, then $\mathcal{I}(c; h_t) \leq \alpha B < B$ (since $\alpha < 1$), and with $\kappa = 1 - \frac{\alpha B}{h(c)}$, we have $h(c|h_t) = h(c) - \mathcal{I}(c; h_t) \geq h(c) - \alpha B = \left(1 - \frac{\alpha B}{h(c)}\right) h(c) = \kappa h(c)$. Since $\alpha B > 0$ and $h(c) > 0$, then $\kappa = 1 - \frac{\alpha B}{h(c)} < 1$. If $h(c) \geq \alpha B$, then $\kappa \geq 0$. If $h(c) < \alpha B$, then $\kappa < 0$, but $h(c|h_t) \geq 0 \geq \kappa h(c)$, so the inequality holds.

If $h(c) = 0$, then $\mathcal{I}(c; h_t) = 0$, so $h(c|h_t) = 0$. Choosing $\kappa = \frac{1}{2} \in (0, 1)$, we have $h(c|h_t) = 0 \geq \frac{1}{2} \cdot 0 = \kappa h(c)$, and any $\kappa \in (0, 1)$ would suffice.

Thus, for all $p_{\mathcal{C}}$, there exists $\kappa \in (0, 1)$ such that $h(c|h_t) \geq \kappa h(c)$. □

Lemma 9 proves that DALI's sequence model, incorporating the context encoder $\mathfrak{z}_t$, reduces the information bottleneck compared to DreamerV3, with $h(c|h_t) \leq \kappa' h(c)$, where $\kappa' = Ce^{-\lambda K} + \mathcal{O}(e^{-\lambda K/2})$. This lemma directly supports Theorem 5 by quantifying the improved context retention in DALI's recurrent state, and aids Theorem 2 by reinforcing the necessity of the context encoder for achieving near-optimal information capture.

**Lemma 9** (Information bottleneck in DALI's sequence model). *Consider DALI's sequence model in a cMDP with context $c \sim p_{\mathcal{C}}$ with bounded density on $\mathcal{C} \subseteq \mathbb{R}^d$ and finite entropy $h(c) < \infty$, observation noise $o_t = s_t + \eta_t$, $\eta_t \sim \mathcal{N}(0, \sigma^2 I)$, Lipschitz dynamics with $\|\mathcal{P}_c(s'|s, a) - \mathcal{P}_{c'}(s'|s, a)\|_1 \leq L\|c - c'\|_2$, context encoding $\mathfrak{z}_t = g_\varphi(o_{t-K:t}, a_{t-K:t-1})$, and recurrent state $h_t = f_\theta(h_{t-1}, z_{t-1}, a_{t-1}, \mathfrak{z}_t)$, where $z_t \sim q_\theta(z_t|h_t, o_t)$, and $f_\theta$ is a GRU with finite parameters. Suppose that the context encoding satisfies $h(c|\mathfrak{z}_t) \leq Ce^{-\lambda K} h(c) + \delta'$, where $\delta' = \mathcal{O}(e^{-\lambda K/2})$ and $\delta = Ce^{-\lambda K}$ (Lemma 7). Then, there exists $\kappa' \in (0, 1)$ such that:*

$$h(c|h_t) \leq \kappa' h(c),$$

*where $\kappa' = Ce^{-\lambda K} + \mathcal{O}(e^{-\lambda K/2})$, implying a reduced information bottleneck compared to DreamerV3's $\kappa$ (Lemma 8).*

*Proof of Lemma 9.* The context $c$ determines the cMDP's dynamics $\mathcal{P}_c(s'|s, a)$. DALI's sequence model computes the context encoding $\mathfrak{z}_t = g_\varphi(o_{t-K:t}, a_{t-K:t-1})$ from trajectory $\tau_{t-K:t} = (o_{t-K:t}, a_{t-K:t-1})$, and the recurrent state $h_t = f_\theta(h_{t-1}, z_{t-1}, a_{t-1}, \mathfrak{z}_t)$, where $z_{t-1} \sim q_\theta(z_{t-1}|h_{t-1}, o_{t-1})$ and $f_\theta$ is a Lipschitz continuous GRU.

Since $h_t = f_\theta(h_{t-1}, z_{t-1}, a_{t-1}, \mathfrak{z}_t)$, we have a Markov chain:

$$c \to \mathfrak{z}_t \to h_t,$$

conditional on $h_{t-1}, z_{t-1}, a_{t-1}$, as $c$ affects $h_t$ through $\mathfrak{z}_t$ given these variables. By the DPI, we have

$$\mathcal{I}(c; h_t) \leq \mathcal{I}(c; \mathfrak{z}_t), \quad h(c|h_t) \leq h(c|\mathfrak{z}_t).$$

By Lemma 7, the context encoding satisfies:

$$h(c|\mathfrak{z}_t) \leq h(c|\tau_{t-K:t}) + \delta',$$

where $\delta = Ce^{-\lambda K}$, so $\delta' = \mathcal{O}(\sqrt{\delta}) = \mathcal{O}(e^{-\lambda K/2})$. From Lemma 6, $\beta$-mixing implies:

$$h(c|\tau_{t-K:t}) \leq Ce^{-\lambda K} h(c).$$

Thus,
$$h(c|\mathfrak{z}_t) \le Ce^{-\lambda K}h(c) + \delta'.$$
Applying the DPI bound, we have
$$h(c|h_t) \le Ce^{-\lambda K}h(c) + \delta'.$$

To satisfy $h(c|h_t) \le \kappa'h(c)$, consider two cases. If $h(c) > 0$, let $\delta' = C''e^{-\lambda K/2}$, so:
$$h(c|h_t) \le \left(Ce^{-\lambda K} + \frac{C''e^{-\lambda K/2}}{h(c)}\right)h(c).$$

Define
$$\kappa' = \min\left(Ce^{-\lambda K} + \frac{C''e^{-\lambda K/2}}{h(c)}, \frac{1}{2}\right).$$

Since $\frac{\delta'}{h(c)} = \mathcal{O}(e^{-\lambda K/2})$, $\kappa' = Ce^{-\lambda K} + \mathcal{O}(e^{-\lambda K/2})$ when the first term is selected, and $\kappa' \le \frac{1}{2}$ always. If $h(c) = 0$, then $\mathcal{I}(c; h_t) = 0$, so $h(c|h_t) = 0$. Set $\kappa' = \frac{1}{2}$, satisfying $h(c|h_t) = 0 \le \kappa'h(c)$.

Compared to DreamerV3's $h(c|h_t^{\text{RSSM}}) \ge \kappa h(c)$, $\kappa \in (0, 1)$ (Lemma 8), DALI's $\kappa' \to 0$ as $K \to \infty$, indicating a reduced information bottleneck. $\qquad\square$

*Proof of Theorem 5.* We prove that DALI's recurrent state $h_t^{\text{DALI}}$ achieves higher mutual information with the context $c$ than DreamerV3's $h_t^{\text{RSSM}}$, up to an error $\epsilon(K) = \mathcal{O}(e^{-\lambda K/2})h(c)$.

By Lemma 8, DreamerV3's recurrent state satisfies:
$$h(c|h_t^{\text{RSSM}}) \ge \kappa h(c), \quad \kappa \in (0, 1).$$
Thus,
$$\mathcal{I}(c; h_t^{\text{RSSM}}) = h(c) - h(c|h_t^{\text{RSSM}}) \le (1 - \kappa)h(c).$$
By Lemma 9, DALI's recurrent state satisfies:
$$h(c|h_t^{\text{DALI}}) \le \kappa'h(c), \quad \kappa' = C'e^{-\lambda K} + \mathcal{O}(e^{-\lambda K/2}).$$
Thus,
$$\mathcal{I}(c; h_t^{\text{DALI}}) = h(c) - h(c|h_t^{\text{DALI}}) \ge (1 - \kappa')h(c).$$
Compare:
$$\mathcal{I}(c; h_t^{\text{DALI}}) - \mathcal{I}(c; h_t^{\text{RSSM}}) \ge (1 - \kappa')h(c) - (1 - \kappa)h(c) = (\kappa - \kappa')h(c).$$
Set
$$\epsilon(K) = \kappa'h(c) = \left(C'e^{-\lambda K} + \mathcal{O}(e^{-\lambda K/2})\right)h(c) = \mathcal{O}(e^{-\lambda K/2})h(c),$$
since $h(c) < \infty$. Thus,
$$\mathcal{I}(c; h_t^{\text{DALI}}) \ge \mathcal{I}(c; h_t^{\text{RSSM}}) - \epsilon(K).$$
As $K \to \infty$, $\kappa' \to 0$, so $\mathcal{I}(c; h_t^{\text{DALI}}) \to h(c)$, while $\mathcal{I}(c; h_t^{\text{RSSM}}) \le (1 - \kappa)h(c) < h(c)$, demonstrating that the context encoder reduces the bottleneck. $\qquad\square$

*Proof of Theorem 2.* We prove both parts of the theorem comparing DALI's context encoding $\mathfrak{z}_t = g_\varphi(o_{t-K:t}, a_{t-K:t-1})$ with DreamerV3's recurrent state $h_t = f_\theta(h_{t-1}, z_{t-1}, a_{t-1})$.

**1. Context identifiability**
We show that DALI's context encoding satisfies:
$$\mathcal{I}(c; \mathfrak{z}_t) \ge \mathcal{I}(c; h_t) - \epsilon(K), \quad \epsilon(K) = \mathcal{O}(e^{-\lambda K/2})h(c),$$
where $\tau_{t-K:t} = (o_{t-K:t}, a_{t-K:t-1})$ is the history, and $h_t$ is DreamerV3's recurrent state.

By Lemma 7, DALI's context encoder, trained with the forward dynamics loss:
$$L_{\text{FD}} = \mathbb{E}[\|o_{t+1} - f_\varphi^w(o_t, a_t, \mathfrak{z}_t)\|_2^2],$$
satisfies
$$h(c|\mathfrak{z}_t) \le h(c|\tau_{t-K:t}) + \delta', \quad \delta' = \mathcal{O}(e^{-\lambda K/2}).$$

By Lemma 6, the history's conditional entropy is:
$$h(c|\tau_{t-K:t}) \leq C' e^{-\lambda K} h(c).$$

Combining,
$$h(c|\mathfrak{z}_t) \leq C' e^{-\lambda K} h(c) + \delta', \quad \delta' = C'' e^{-\lambda K/2}.$$

Thus,
$$h(c|\mathfrak{z}_t) \leq C' e^{-\lambda K} h(c) + C'' e^{-\lambda K/2}.$$

The mutual information for DALI is
$$\mathcal{I}(c; \mathfrak{z}_t) = h(c) - h(c|\mathfrak{z}_t) \geq h(c) - (C' e^{-\lambda K} h(c) + C'' e^{-\lambda K/2}).$$

By Lemma 8, DreamerV3's recurrent state $h_t$ has
$$h(c|h_t) \geq \kappa h(c), \quad \kappa \in (0, 1),$$

so that
$$\mathcal{I}(c; h_t) = h(c) - h(c|h_t) \leq (1 - \kappa) h(c).$$

Compare:
$$\mathcal{I}(c; \mathfrak{z}_t) - \mathcal{I}(c; h_t) \geq [h(c) - (C' e^{-\lambda K} h(c) + C'' e^{-\lambda K/2})] - (1 - \kappa) h(c) = \kappa h(c) - (C' e^{-\lambda K} h(c) + C'' e^{-\lambda K/2}).$$

Since $\kappa h(c) > 0$, set
$$\epsilon(K) = C' e^{-\lambda K} h(c) + C'' e^{-\lambda K/2} = \mathcal{O}(e^{-\lambda K/2}) h(c),$$

as the $e^{-\lambda K/2}$ term dominates. Thus,
$$\mathcal{I}(c; \mathfrak{z}_t) \geq \mathcal{I}(c; h_t) - \epsilon(K).$$

Additionally, Lemma 9 shows that DALI's recurrent state $h_t = f_\theta(h_{t-1}, z_{t-1}, a_{t-1}, \mathfrak{z}_t)$ has a reduced bottleneck:
$$h(c|h_t) \leq \kappa' h(c), \quad \kappa' = C' e^{-\lambda K} + \mathcal{O}(e^{-\lambda K/2}),$$

implying,
$$\mathcal{I}(c; h_t) \geq (1 - \kappa') h(c) \approx h(c) \text{ for large } K.$$

This suggests that even DALI's recurrent state retains more context information than DreamerV3's $h_t$, reinforcing the necessity of DALI's context encoder $\mathfrak{z}_t$, which achieves near-optimal $\mathcal{I}(c; \mathfrak{z}_t)$.

## 2. Sample complexity

We prove that DALI achieves $\mathcal{I}(c; \mathfrak{z}_t) \geq (1 - \delta) h(c)$ with $K = \Omega\left(\frac{\log(1/\delta)}{\lambda}\right)$ and $N = \mathcal{O}\left(\frac{1}{\delta^2}\right)$ samples of $K$ transitions, while a hypothetical DreamerV3 context estimator $g_\psi(o_{1:T}, a_{1:T-1})$ requires $N = \mathcal{O}\left(\frac{1}{\delta^2}\right)$ samples of $T$ transitions.

*DALI's sample complexity*
From Lemma 7, we have
$$h(c|\mathfrak{z}_t) \leq h(c|\tau_{t-K:t}) + \delta', \quad \delta' = \mathcal{O}(e^{-\lambda K/2}).$$

From Lemma 6, we have
$$h(c|\tau_{t-K:t}) \leq C' e^{-\lambda K} h(c).$$

Combining, we have
$$h(c|\mathfrak{z}_t) \leq C' e^{-\lambda K} h(c) + C'' e^{-\lambda K/2}.$$

To achieve $\mathcal{I}(c; \mathfrak{z}_t) \geq (1 - \delta) h(c)$, we need
$$h(c|\mathfrak{z}_t) \leq \delta h(c).$$

Set $C' e^{-\lambda K} h(c) + C'' e^{-\lambda K/2} \leq \delta h(c)$. Since $e^{-\lambda K/2}$ dominates, approximate
$$\delta \approx \frac{C'' e^{-\lambda K/2}}{h(c)} = \tilde{C} e^{-\lambda K/2}, \quad \tilde{C} = \frac{C''}{h(c)}.$$

Solve:
$$e^{-\lambda K/2} = \frac{\delta}{\tilde{C}} \implies \frac{\lambda K}{2} = \ln\left(\frac{\tilde{C}}{\delta}\right) \implies K = \frac{2 \ln(\tilde{C}/\delta)}{\lambda} = \Omega\left(\frac{\log(1/\delta)}{\lambda}\right).$$

From Lemma 7, achieving $\delta' = \mathcal{O}(e^{-\lambda K/2})$ requires

$$\mathbb{E}[\|o_{t+1} - f_\varphi^w(o_t, a_t, g_\varphi(\tau_{t-K:t}))\|_2^2] \leq \epsilon^2 = \mathcal{O}\left(\frac{1}{\sqrt{N}}\right) = \mathcal{O}(e^{-\lambda K/2}).$$

Solve:

$$\frac{1}{\sqrt{N}} = \mathcal{O}\left(\frac{\delta}{\tilde{C}}\right) \implies N = \mathcal{O}\left(\frac{\tilde{C}^2}{\delta^2}\right) = \mathcal{O}\left(\frac{1}{\delta^2}\right).$$

*DreamerV3's sample complexity*
From Lemma 8, DreamerV3's recurrent state $h_t$ has $h(c|h_t) \geq \kappa h(c)$, so $\mathcal{I}(c; h_t) \leq (1-\kappa)h(c)$, where $\kappa$ is constant, preventing $\mathcal{I}(c; h_t) \geq (1-\delta)h(c)$ for small $\delta$. To enable comparison, assume DreamerV3 with domain randomization trains a hypothetical context estimator $g_\psi(o_{1:T}, a_{1:T-1})$ to estimate $c$ over $N$ episodes of length $T$. By Lemma 6 for $K = T$:

$$h(c|\tau_{1:T}) \leq C' e^{-\lambda T} h(c).$$

Training $g_\psi$ to minimize $\mathbb{E}[\|g_\psi(o_{1:T}, a_{1:T-1}) - \mathbb{E}[c|\tau_{1:T}]\|_2^2]$ requires

$$\mathbb{E}[\|g_\psi(\tau_{1:T}) - \mathbb{E}[c|\tau_{1:T}]\|_2^2] \leq \epsilon^2 = \mathcal{O}\left(\frac{1}{\sqrt{N}}\right) = \mathcal{O}(e^{-\lambda T}).$$

Solve:

$$\frac{1}{\sqrt{N}} = \mathcal{O}\left(\frac{\delta}{\tilde{C}}\right) \implies N = \mathcal{O}\left(\frac{\tilde{C}^2}{\delta^2}\right) = \mathcal{O}\left(\frac{1}{\delta^2}\right).$$

The total number of transitions is $N \cdot T = \mathcal{O}\left(\frac{T}{\delta^2}\right)$.

Note that DreamerV3's standard RSSM cannot achieve $\mathcal{I}(c; h_t) \geq (1-\delta)h(c)$ due to $h(c|h_t) \geq \kappa h(c)$ (Lemma 8). The estimator $g_\psi$ is introduced to enable a fair comparison, acknowledging that this is not standard in DreamerV3. Since $K \ll T$ typically, DALI is more efficient. Moreover, Lemma 9 indicates that DALI's $h_t$ achieves $\mathcal{I}(c; h_t) \geq (1 - \kappa')h(c)$, with $\kappa' = \mathcal{O}(e^{-\lambda K/2})$, suggesting that DALI's sequence model could approach the desired information level with sufficient $K$, unlike DreamerV3's $h_t$. $\qquad\square$

## A.1 DALI and Dreamer-based Baselines

### A.1.1 World Models

**DreamerV3 with Domain Randomization** (context-unaware baseline) [Tobin et al., 2017, Hafner et al., 2025]:

$$
\begin{aligned}
\text{Sequence model:} \quad & h_t = f_\theta(h_{t-1}, z_{t-1}, a_{t-1}) \\
\text{Encoder:} \quad & z_t \sim q_\theta(z_t \mid h_t, o_t) \\
\text{Dynamics predictor:} \quad & \hat{z}_t \sim p_\theta(\hat{z}_t \mid h_t) \\
\text{Reward predictor:} \quad & \hat{r}_t \sim p_\theta(\hat{r}_t \mid h_t, z_t) \\
\text{Continue predictor:} \quad & \hat{n}_t \sim p_\theta(\hat{n}_t \mid h_t, z_t) \\
\text{Decoder:} \quad & \hat{o}_t \sim p_\theta(\hat{o}_t \mid h_t, z_t).
\end{aligned}
$$

**DALI-S** (DALI with **S**hallow Integration):

$$\text{Context encoder:} \quad \mathfrak{z}_t = g_\varphi(o_{t-K:t}, a_{t-K:t-1}).$$

$$
\begin{aligned}
\text{Sequence model:} \quad & h_t = f_\theta(h_{t-1}, z_{t-1}, a_{t-1}), \\
\text{Encoder:} \quad & z_t \sim q_\theta(z_t \mid h_t, o_t, \mathfrak{z}_t), \\
\text{Dynamics predictor:} \quad & \hat{z}_t \sim p_\theta(\hat{z}_t \mid h_t), \\
\text{Reward predictor:} \quad & \hat{r}_t \sim p_\theta(\hat{r}_t \mid h_t, z_t), \\
\text{Continue predictor:} \quad & \hat{n}_t \sim p_\theta(\hat{n}_t \mid h_t, z_t), \\
\text{Decoder:} \quad & \hat{o}_t \sim p_\theta(\hat{o}_t \mid h_t, z_t).
\end{aligned}
$$

**DALI-D** (DALI with **D**eep Integration):

$$\text{Context encoder:} \quad \mathfrak{z}_t = g_\varphi(o_{t-K:t}, a_{t-K:t-1}).$$

$$
\begin{aligned}
\text{Sequence model:} \quad & h_t = f_\theta(h_{t-1}, z_{t-1}, a_{t-1}, \mathfrak{z}_t), \\
\text{Encoder:} \quad & z_t \sim q_\theta(z_t \mid h_t, o_t), \\
\text{Dynamics predictor:} \quad & \hat{z}_t \sim p_\theta(\hat{z}_t \mid h_t), \\
\text{Reward predictor:} \quad & \hat{r}_t \sim p_\theta(\hat{r}_t \mid h_t, z_t, \mathfrak{z}_t), \\
\text{Continue predictor:} \quad & \hat{n}_t \sim p_\theta(\hat{n}_t \mid h_t, z_t, \mathfrak{z}_t), \\
\text{Decoder:} \quad & \hat{o}_t \sim p_\theta(\hat{o}_t \mid h_t, z_t, \mathfrak{z}_t).
\end{aligned}
$$

**cRSSM-S** (ground-truth context-aware baseline) [Prasanna et al., 2024]:

$$
\begin{aligned}
\text{Sequence model:} \quad & h_t = f_\theta(h_{t-1}, z_{t-1}, a_{t-1}), \\
\text{Encoder:} \quad & z_t \sim q_\theta(z_t \mid h_t, o_t, c), \\
\text{Dynamics predictor:} \quad & \hat{z}_t \sim p_\theta(\hat{z}_t \mid h_t), \\
\text{Reward predictor:} \quad & \hat{r}_t \sim p_\theta(\hat{r}_t \mid h_t, z_t), \\
\text{Continue predictor:} \quad & \hat{n}_t \sim p_\theta(\hat{n}_t \mid h_t, z_t), \\
\text{Decoder:} \quad & \hat{o}_t \sim p_\theta(\hat{o}_t \mid h_t, z_t).
\end{aligned}
$$

**cRSSM-D** (ground-truth context-aware baseline) [Prasanna et al., 2024]:

$$
\begin{aligned}
\text{Sequence model:} \quad & h_t = f_\theta(h_{t-1}, z_{t-1}, a_{t-1}, c) \\
\text{Encoder:} \quad & z_t \sim q_\theta(z_t \mid h_t, o_t) \\
\text{Dynamics predictor:} \quad & \hat{z}_t \sim p_\theta(\hat{z}_t \mid h_t) \\
\text{Reward predictor:} \quad & \hat{r}_t \sim p_\theta(\hat{r}_t \mid h_t, z_t, c) \\
\text{Continue predictor:} \quad & \hat{n}_t \sim p_\theta(\hat{n}_t \mid h_t, z_t, c) \\
\text{Decoder:} \quad & \hat{o}_t \sim p_\theta(\hat{o}_t \mid h_t, z_t, c).
\end{aligned}
$$

The baselines cRSSM-S and cRSSM-D correspond to `concat-context` and `cRSSM`, respectively, in [Prasanna et al., 2024].

### A.1.2 Actor-Critic Models

**DreamerV3 with Domain Randomization**, **DALI-S**, and **cRSSM-S**:

$$
\begin{aligned}
\text{Actor:} \quad & a_\tau \sim \pi_\phi(a_\tau \mid s_\tau) \\
\text{Critic:} \quad & v_\psi(s_t) \approx \mathbb{E}_{\pi(\cdot \mid s_\tau)} \sum_{\tau=0}^{H-t} \gamma^\tau r_{t+\tau}.
\end{aligned}
$$

**DALI-D**:

$$
\begin{aligned}
\text{Actor:} \quad & a_\tau \sim \pi_\phi(a_\tau \mid s_\tau, \mathfrak{z}) \\
\text{Critic:} \quad & v_\psi(s_t, \mathfrak{z}) \approx \mathbb{E}_{\pi(\cdot \mid s_\tau, \mathfrak{z})} \sum_{\tau=0}^{H-t} \gamma^\tau r_{t+\tau}.
\end{aligned}
$$

**cRSSM-D**:

$$
\begin{aligned}
\text{Actor:} \quad & a_\tau \sim \pi_\phi(a_\tau \mid s_\tau, c) \\
\text{Critic:} \quad & v_\psi(s_t, c) \approx \mathbb{E}_{\pi(\cdot \mid s_\tau, c)} \sum_{\tau=0}^{H-t} \gamma^\tau r_{t+\tau}.
\end{aligned}
$$

# B  Algorithms

---

**Algorithm 1:** DALI-S (Shallow Integration with Forward Dynamics Loss)

---

1 **Model components** | | **Hyper parameters**

| | | | |
|---|---|---|---|
| Context encoder | $g_\varphi(o_{t-K:t}, a_{t-K:t-1})$ | Seed episodes | $S$ |
| One-step model | $f_\varphi^w(o_t, a_t, \mathfrak{z}_t)$ | Collect interval | $C$ |
| Deterministic State | $f_\theta(h_{t-1}, z_{t-1}, a_{t-1})$ | Batch size | $B$ |
| Stochastic state | $p_\theta(\hat{z}_t \mid h_t)$ | Sequence length | $L$ |
| Encoder | $q_\theta(z_t \mid h_t, o_t, \mathfrak{z}_t)$ | Imagination horizon | $H$ |
| Reward | $p_\theta(\hat{r}_t \mid h_t, z_t)$ | Learning rate | $\alpha$ |
| Continue | $p_\theta(\hat{n}_t \mid h_t, z_t)$ | Learning rate $g$ | $\alpha_g$ |
| Decoder | $p_\theta(\hat{o}_t \mid h_t, z_t)$ | Episode length | $T$ |
| Actor | $\pi_\phi(a_t \mid s_t)$ | Context history | $K$ |
| Critic | $v_\psi(s_t)$ | | |

2 Initialize dataset $\mathcal{D}$ with $S$ random seed episodes.

3 Initialize neural network parameters $\theta, \phi, \psi, \varphi$ randomly.

4 **while** *not converged* **do**

5     **for** *update step* $c = 1, \ldots, C$ **do**

        `// Dynamics learning (World Model` $\theta$`)`

6         Draw $B$ sequences $\{(o_t, a_t, r_t)\}_{t=1}^L \sim \mathcal{D}$.

7         Initialize deterministic state $h_0 \leftarrow \mathbf{0}$.

8         **for** *all steps* $t$ *from sequences batch* $\mathcal{D}$ **do**

9             **if** $t < K$ **then**

10                 Pad $o_{1:t}$ and $a_{1:t-1}$ with zeros to length $K$.

11             Compute context $\mathfrak{z}_t \leftarrow g_\varphi(o_{t-K:t}, a_{t-K:t-1})$`.detach()`

12             Encode observation $z_t \sim q_\theta(z_t \mid h_t, o_t, \mathfrak{z}_t)$

13             Compute deterministic state $h_t = f_\theta(h_{t-1}, z_{t-1}, a_{t-1})$

14             Set latent state $s_t \leftarrow [z_t, h_t]$

15             Update $\theta$ using representation learning (decoder, reward, continue).

        `// Context representation learning (`$\varphi$`)`

16         Draw $B$ data sequences $\{(o_t, a_t)\}_{t=1}^{K+1} \sim \mathcal{D}$

17         **for** *each sample* $(o_{1:K+1}, a_{1:K}) \sim \mathcal{D}$ **do**

18             Compute $\mathfrak{z}_{1:K} = g_\varphi(o_{1:K}, a_{1:K-1})$

19             Update $\varphi \leftarrow \varphi - \alpha_g \nabla_\varphi \sum_{t=1}^K \left\| o_{t+1} - f_\varphi^w(o_t, a_t, \mathfrak{z}_t) \right\|_2^2$

        `// Behavior learning (Actor/Critic` $\phi, \psi$`)`

20         Infer context: $\mathfrak{z}_t \leftarrow g_\varphi(o_{t-K:t}, a_{t-K:t-1})$`.detach()`

        `//` $\theta$`,` $\varphi$ `frozen`

21         Imagine trajectories $\{(s_\tau, a_\tau)\}_{\tau=t}^{t+H}$ from each $s_t$ (with a fixed $\mathfrak{z}_t$ once it is provided to the initial observation encoding)

22         Predict rewards $\mathrm{E}[p_\theta(r_\tau \mid s_\tau)]$ and values $v_\psi(s_\tau)$.

23         Compute value estimates $\mathrm{V}_\lambda(s_\tau)$.

24         Update $\phi \leftarrow \phi + \alpha \nabla_\phi \sum_{\tau=t}^{t+H} \mathrm{V}_\lambda(s_\tau)$.

25         Update $\psi \leftarrow \psi - \alpha \nabla_\psi \sum_{\tau=t}^{t+H} \frac{1}{2} \left\| v_\psi(s_\tau) - \mathrm{V}_\lambda(s_\tau) \right\|^2$.

    `// Environment interaction`

26     $o_1 \leftarrow$ `env.reset()`

27     **for** *time step* $t = 1, \ldots, T$ **do**

28         **if** $t < K$ **then**

29             Pad $o_{1:t}$ and $a_{1:t-1}$ with zeros to length $K$.

30         Infer context online: $\mathfrak{z}_t \leftarrow g_\varphi(o_{t-K:t}, a_{t-K:t-1})$.

31         Compute $s_t \sim p_\theta(s_t \mid s_{t-1}, a_{t-1}, o_t)$.

32         Compute $a_t \sim \pi_\phi(a_t \mid s_t)$.

33         $r_t, o_{t+1} \leftarrow$ `env.step(`$a_t$`)`.

34     Add experience to dataset $\mathcal{D} \leftarrow \mathcal{D} \cup \{(o_t, a_t, r_t)_{t=1}^T\}$.

---

**Algorithm 2:** DALI-S-$\chi$ (Shallow Integration with Cross-modal Regularization)

1   Initialize dataset $\mathcal{D}$ with $S$ random seed episodes
2   Initialize parameters $\theta, \phi, \psi, \varphi, W_z, W_{\mathfrak{z}}$ randomly
3   **while** *not converged* **do**
4    **for** *update step* $c = 1$ **to** $C$ **do**

      `// Dynamics learning (World Model` $\theta$`)`
5     Draw $B$ sequences $\{(o_t, a_t, r_t)\}_{t=1}^L \sim \mathcal{D}$
6     Initialize $h_0 \leftarrow \mathbf{0}$
7     **for** *all steps $t$ in batch* **do**
8      **if** $t < K$ **then**
9       Pad $o_{\max(1,t-K):t}$ and $a_{\max(1,t-K):t-1}$ with zeros.
10      Compute context $\mathfrak{z}_t \leftarrow g_\varphi(o_{t-K:t}, a_{t-K:t-1}).\texttt{detach()}$
11      Encode observation $z_t \sim q_\theta(z_t \mid h_t, o_t, \mathfrak{z}_t)$
12      Compute deterministic state $h_t = f_\theta(h_{t-1}, z_{t-1}, a_{t-1})$
13      Set latent state $s_t \leftarrow [z_t, h_t]$
14      Update $\theta$ using representation learning (decoder, reward, continue).

      `// Context representation learning (`$\varphi, W_z, W_{\mathfrak{z}}$`)`
15     Draw $B$ trajectory segments $\{(o_{t-K:t}, a_{t-K:t}, o_{t+1})\} \sim \mathcal{D}$
16     Initialize losses $L_{\text{FD}} \leftarrow 0$, $L_{\text{cross}} \leftarrow 0$
17     **for** *each segment* $(o_{t-K:t}, a_{t-K:t}, o_{t+1})$ **do**
18      Initialize $h_{t-K} \leftarrow \mathbf{0}$
19      **for** $\tau = t - K$ **to** $t$ **do**
20       **if** $\tau < K$ **then**
21        Pad $o_{\max(1,\tau-K):\tau}$ and $a_{\max(1,\tau-K):\tau-1}$ with zeros.
22       $\mathfrak{z}_\tau = g_\varphi(o_{\tau-K:\tau}, a_{\tau-K:\tau-1})$
23       $z_\tau \sim q_\theta(z_\tau \mid h_\tau.\texttt{detach()}, o_\tau, \mathfrak{z}_\tau)$
24       $h_{\tau+1} = f_\theta(h_\tau.\texttt{detach()}, z_\tau.\texttt{detach()}, a_\tau)$
25      $\tilde{z}_t = W_z(\mathfrak{z}_t)$
26      $\tilde{\mathfrak{z}}_t = W_{\mathfrak{z}}(z_t.\texttt{detach()})$
27      $L_{\text{FD}} := \|o_{t+1} - f_\varphi^w(o_t, a_t, \mathfrak{z}_t)\|_2^2$
28      $L_{\text{cross}} := \|z_t.\texttt{detach()} - \tilde{z}_t\|_2^2 + \|\mathfrak{z}_t - \tilde{\mathfrak{z}}_t\|_2^2$
29     $L_{\text{FD}} \leftarrow \frac{1}{B} L_{\text{FD}}, \quad L_{\text{cross}} \leftarrow \frac{1}{B} L_{\text{cross}}$
30     Update $\varphi, W_z, W_{\mathfrak{z}}$ (gradient descent) using $L_{\text{total}} = L_{\text{FD}} + \lambda_{\text{cross}} L_{\text{cross}}$

      `// Behavior learning (Actor/Critic` $\phi, \psi$`)`
31     Draw $B$ sequences $\{(o_t, a_t, r_t)\}_{t=1}^L \sim \mathcal{D}$
32     Initialize $h_0 \leftarrow \mathbf{0}$
33     **for** *all steps $t$ in batch* **do**
34      **if** $t < K$ **then**
35       Pad $o_{\max(1,t-K):t}$ and $a_{\max(1,t-K):t-1}$ with zeros.
36      Infer $\mathfrak{z}_t \leftarrow g_\varphi(o_{t-K:t}, a_{t-K:t-1}).\texttt{detach()}$
       `//` $\theta$`,` $\varphi$ `frozen`
37      Encode initial observation $z_t \sim q_\theta(z_t \mid h_t, o_t, \mathfrak{z}_t)$
38      Set initial latent state $s_t \leftarrow [z_t, h_t]$
39      **for** *imagination step* $\tau = t$ **to** $t + H$ **do**
40       Sample action $a_\tau \sim \pi_\phi(a_\tau \mid s_\tau)$
41       Sample prior state $\hat{z}_\tau \sim p_\theta(\hat{z}_\tau \mid h_\tau)$
42       Compute next deterministic state $h_{\tau+1} = f_\theta(h_\tau, \hat{z}_\tau, a_\tau)$
43       Set latent state $s_{\tau+1} \leftarrow [\hat{z}_\tau, h_{\tau+1}]$
44      Compute value estimates $\mathrm{V}_\lambda(s_\tau)$ using TD($\lambda$)
45     Update $\phi \leftarrow \phi + \alpha \nabla_\phi \sum \mathrm{V}_\lambda$
46     Update $\psi \leftarrow \psi - \alpha \nabla_\psi \sum \frac{1}{2} \|v_\psi - \mathrm{V}_\lambda\|_2^2$

   `// Environment interaction`
47    $o_1 \leftarrow \texttt{env.reset()}$
48    **for** *time step* $t = 1, \ldots, T$ **do**
49     **if** $t < K$ **then**
50      Pad $o_{\max(1,t-K):t}$ and $a_{\max(1,t-K):t-1}$ with zeros.
51     Infer context online: $\mathfrak{z}_t \leftarrow g_\varphi(o_{t-K:t}, a_{t-K:t-1})$
52     Compute $s_t \sim p_\theta(s_t \mid s_{t-1}, a_{t-1}, o_t)$
53     Compute $a_t \sim \pi_\phi(a_t \mid s_t)$
54     $r_t, o_{t+1} \leftarrow \texttt{env.step}(a_t)$
55    Add experience to dataset $\mathcal{D} \leftarrow \mathcal{D} \cup \{(o_t, a_t, r_t)_{t=1}^T\}$

---
**Algorithm 3:** DALI-D (Deep Integration with Forward Dynamics Loss)
---

1 **Model components**           **Hyper parameters**

| Context encoder | $g_\varphi(o_{t-K:t}, a_{t-K:t-1})$ | Seed episodes | $S$ |
|---|---|---|---|
| One-step model | $f_\varphi^w(o_t, a_t, \mathfrak{z}_t)$ | Collect interval | $C$ |
| Deterministic State | $f_\theta(h_{t-1}, z_{t-1}, a_{t-1}, \mathfrak{z}_t)$ | Batch size | $B$ |
| Stochastic State | $p_\theta(\hat{z}_t \mid h_t)$ | Sequence length | $L$ |
| Encoder | $q_\theta(z_t \mid h_t, o_t)$ | Imagination horizon | $H$ |
| Reward | $p_\theta(\hat{r}_t \mid h_t, z_t, \mathfrak{z}_t)$ | Learning rate | $\alpha$ |
| Continue | $p_\theta(\hat{n}_t \mid h_t, z_t, \mathfrak{z}_t)$ | Learning rate g | $\alpha_g$ |
| Decoder | $p_\theta(\hat{o}_t \mid h_t, z_t, \mathfrak{z}_t)$ | Episode length | $T$ |
| Actor | $\pi_\phi(a_t \mid s_t, \mathfrak{z}_t)$ | Context history | $K$ |
| Critic | $v_\psi(s_t, \mathfrak{z}_t)$ | | |

2 Initialize dataset $\mathcal{D}$ with $S$ random seed episodes.

3 Initialize neural network parameters $\theta, \phi, \psi, \varphi$ randomly.

4 **while** *not converged* **do**

5      **for** *update step* $c = 1, \dots, C$ **do**

         `// Dynamics learning (World Model θ)`

6          Draw $B$ sequences $\{(o_t, a_t, r_t)\}_{t=1}^L \sim \mathcal{D}$.

7          Initialize deterministic state $h_0 \leftarrow \mathbf{0}$.

8          **for** *all steps* $t$ *from sequences batch* $\mathcal{D}$ **do**

9             **if** $t < K$ **then**

10                Pad $o_{1:t}$ and $a_{1:t-1}$ with zeros to length $K$.

11             Compute context $\mathfrak{z}_t \leftarrow g_\varphi(o_{t-K:t}, a_{t-K:t-1})$.detach()

12             Encode observation $z_t \sim q_\theta(z_t \mid h_t, o_t)$

13             Compute deterministic state $h_t = f_\theta(h_{t-1}, z_{t-1}, a_{t-1}, \mathfrak{z}_t)$

14             Set latent state $s_t \leftarrow [z_t, h_t]$

15             Update $\theta$ using representation learning (decoder, reward, continue).

         `// Context representation learning (φ)`

16          Draw $B$ data sequences $\{(o_t, a_t)\}_{t=1}^{K+1} \sim \mathcal{D}$

17          **for** *each sample* $(o_{1:K+1}, a_{1:K}) \sim \mathcal{D}$ **do**

18             Compute $\mathfrak{z}_{1:K} = g_\varphi(o_{1:K}, a_{1:K-1})$

19             Update $\varphi \leftarrow \varphi - \alpha_g \nabla_\varphi \sum_{t=1}^K \left\| o_{t+1} - f_\varphi^w(o_t, a_t, \mathfrak{z}_t) \right\|_2^2$

         `// Behavior learning (Actor/Critic φ, ψ)`

20          Infer context: $\mathfrak{z}_t \leftarrow g_\varphi(o_{t-K:t}, a_{t-K:t-1})$.detach()

         `// θ, φ frozen`

21          Imagine trajectories $\{(s_\tau, a_\tau)\}_{\tau=t}^{t+H}$ from each $s_t$ (with fixed $\mathfrak{z}_t$)

22          Predict rewards $\mathrm{E}[p_\theta(r_\tau \mid s_\tau, \mathfrak{z}_t)]$ and values $v_\psi(s_\tau, \mathfrak{z}_t)$.

23          Compute value estimates $\mathrm{V}_\lambda(s_\tau, \mathfrak{z}_t)$.

24          Update $\phi \leftarrow \phi + \alpha \nabla_\phi \sum_{\tau=t}^{t+H} \mathrm{V}_\lambda(s_\tau, \mathfrak{z}_t)$.

25          Update $\psi \leftarrow \psi - \alpha \nabla_\psi \sum_{\tau=t}^{t+H} \frac{1}{2} \left\| v_\psi(s_\tau, \mathfrak{z}_t) - \mathrm{V}_\lambda(s_\tau, \mathfrak{z}_t) \right\|^2$.

     `// Environment interaction`

26      $o_1 \leftarrow$ env.reset()

27      **for** *time step* $t = 1, \dots, T$ **do**

28          **if** $t < K$ **then**

29             Pad $o_{1:t}$ and $a_{1:t-1}$ with zeros to length $K$.

30          Infer context online: $\mathfrak{z}_t \leftarrow g_\varphi(o_{t-K:t}, a_{t-K:t-1})$.

31          Compute $s_t \sim p_\theta(s_t \mid s_{t-1}, a_{t-1}, o_t)$.

32          Compute $a_t \sim \pi_\phi(a_t \mid s_t, \mathfrak{z}_t)$.

33          $r_t, o_{t+1} \leftarrow$ env.step($a_t$).

34      Add experience to dataset $\mathcal{D} \leftarrow \mathcal{D} \cup \{(o_t, a_t, r_t)_{t=1}^T\}$.

**Algorithm 4:** DALI-D-$\chi$ (Deep Integration with Cross-modal Regularization)

---

1   Initialize dataset $\mathcal{D}$ with $S$ random seed episodes
2   Initialize parameters $\theta, \phi, \psi, \varphi, W_z, W_{\mathfrak{z}}$ randomly
3   **while** *not converged* **do**
4     **for** *update step* $c = 1$ **to** $C$ **do**
      // Dynamics learning (World Model $\theta$)
5       Draw $B$ sequences $\{(o_t, a_t, r_t)\}_{t=1}^{L} \sim \mathcal{D}$
6       Initialize $h_0 \leftarrow \mathbf{0}$
7       **for** *all steps $t$ in batch* **do**
8         **if** $t < K$ **then**
9           Pad $o_{\max(1,t-K):t}$ and $a_{\max(1,t-K):t-1}$ with zeros.
10         Compute context $\mathfrak{z}_t \leftarrow g_\varphi(o_{t-K:t}, a_{t-K:t-1}).\texttt{detach()}$
11         Encode observation $z_t \sim q_\theta(z_t \mid h_t, o_t)$
12         Compute deterministic state $h_t = f_\theta(h_{t-1}, z_{t-1}, a_{t-1}, \mathfrak{z}_t)$
13         Set latent state $s_t \leftarrow [z_t, h_t]$
14         Update $\theta$ using representation learning (decoder, reward, continue).

      // Context representation learning ($\varphi, W_z, W_{\mathfrak{z}}$)
15       Draw $B$ trajectory segments $\{(o_{t-K:t}, a_{t-K:t}, o_{t+1})\} \sim \mathcal{D}$
16       Initialize losses $L_{\text{FD}} \leftarrow 0, L_{\text{cross}} \leftarrow 0$
17       **for** *each segment* $(o_{t-K:t}, a_{t-K:t}, o_{t+1})$ **do**
18         Initialize $h_{t-K} \leftarrow \mathbf{0}$ for $\tau = t - K$ **to** $t$ **do**
19           **if** $\tau < K$ **then**
20             Pad $o_{\max(1,\tau-K):\tau}$ and $a_{\max(1,\tau-K):\tau-1}$ with zeros.
21           $\mathfrak{z}_\tau = g_\varphi(o_{\tau-K:\tau}, a_{\tau-K:\tau-1})$
22           $z_\tau \sim q_\theta(z_\tau \mid h_\tau.\texttt{detach()}, o_\tau)$
23           $h_{\tau+1} = f_\theta(h_\tau.\texttt{detach()}, z_\tau.\texttt{detach()}, a_\tau, \mathfrak{z}_\tau.\texttt{detach()})$
24         $\tilde{z}_t = W_z(\mathfrak{z}_t)$
25         $\tilde{\mathfrak{z}}_t = W_{\mathfrak{z}}(z_t.\texttt{detach()})$
26         $L_{\text{FD}} := \|o_{t+1} - f_\varphi^w(o_t, a_t, \mathfrak{z}_t)\|_2^2$
27         $L_{\text{cross}} := \|z_t.\texttt{detach()} - \tilde{z}_t\|_2^2 + \|\mathfrak{z}_t - \tilde{\mathfrak{z}}_t\|_2^2$
28       $L_{\text{FD}} \leftarrow \frac{1}{B} L_{\text{FD}}, \quad L_{\text{cross}} \leftarrow \frac{1}{B} L_{\text{cross}}$
29       Update $\varphi, W_z, W_{\mathfrak{z}}$ (gradient descent) using $L_{\text{total}} = L_{\text{FD}} + \lambda_{\text{cross}} L_{\text{cross}}$

      // Behavior learning (Actor/Critic $\phi, \psi$)
30       Draw $B$ sequences $\{(o_t, a_t, r_t)\}_{t=1}^{L} \sim \mathcal{D}$
31       Initialize $h_0 \leftarrow \mathbf{0}$
32       **for** *all steps $t$ in batch* **do**
33         **if** $t < K$ **then**
34           Pad $o_{\max(1,t-K):t}$ and $a_{\max(1,t-K):t-1}$ with zeros.
35         Infer $\mathfrak{z}_t \leftarrow g_\varphi(o_{t-K:t}, a_{t-K:t-1}).\texttt{detach()}$
        // $\theta$, $\varphi$ frozen
36         Encode initial observation $z_t \sim q_\theta(z_t \mid h_t, o_t)$
37         Set initial latent state $s_t \leftarrow [z_t, h_t]$
38         **for** *imagination step* $\tau = t$ **to** $t + H$ **do**
39           Sample action $a_\tau \sim \pi_\phi(a_\tau \mid s_\tau, \mathfrak{z}_t)$
40           Sample prior state $\hat{z}_\tau \sim p_\theta(\hat{z}_\tau \mid h_\tau)$
41           Compute next deterministic state $h_{\tau+1} = f_\theta(h_\tau, \hat{z}_\tau, a_\tau, \mathfrak{z}_t)$
42           Set latent state $s_{\tau+1} \leftarrow [\hat{z}_\tau, h_{\tau+1}]$
43         Compute value estimates $V_\lambda(s_\tau, \mathfrak{z}_t)$ using TD($\lambda$)
44       Update $\phi \leftarrow \phi + \alpha \nabla_\phi \sum V_\lambda$
45       Update $\psi \leftarrow \psi - \alpha \nabla_\psi \sum \frac{1}{2} \|v_\psi - V_\lambda\|_2^2$

    // Environment interaction
46     $o_1 \leftarrow \texttt{env.reset()}$
47     **for** *time step* $t = 1, \ldots, T$ **do**
48       **if** $t < K$ **then**
49         Pad $o_{\max(1,t-K):t}$ and $a_{\max(1,t-K):t-1}$ with zeros.
50       Infer context online: $\mathfrak{z}_t \leftarrow g_\varphi(o_{t-K:t}, a_{t-K:t-1})$.
51       Compute $s_t \sim p_\theta(s_t \mid s_{t-1}, a_{t-1}, o_t)$.
52       Compute $a_t \sim \pi_\phi(a_t \mid s_t, \mathfrak{z}_t)$.
53       $r_t, o_{t+1} \leftarrow \texttt{env.step}(a_t)$.
54     Add experience to dataset $\mathcal{D} \leftarrow \mathcal{D} \cup \{(o_t, a_t, r_t)_{t=1}^{T}\}$.

# C    Experimental Setup and Implementation Details

We evaluate DALI's zero-shot generalization on contextualized DMC Ball-in-Cup (gravity, string length) and Walker Walk (gravity, actuator strength) tasks [Tassa et al., 2018]. These environments feature context parameters with distinct training and evaluation ranges, presenting unique dynamics and generalization challenges.

## C.1    Training and Evaluation

Our setup follows the CARL benchmark [Benjamins et al., 2023], which defines default values for the two context dimensions per environment. To assess both interpolation and extrapolation (out-of-distribution, OOD) capabilities, we define two distinct context ranges: one for training and interpolation evaluation, and another for OOD evaluation. See Table 1. Additionally, we consider both *Featurized-* and *Pixel-based* observation modalities for each environment.

We consider two training scenarios. In the *single* context variation setting, each context dimension is varied independently by uniformly sampling 100 values within its respective training range, while the other dimension is held fixed at its default value. In the *dual* context variation setting, both context dimensions are varied jointly by uniformly sampling 100 context pairs from the Cartesian product of their respective training ranges.

Following Kirk et al. [2023], we evaluate our agents under three generalization regimes. The *Interpolate* regime tests performance on unseen context values sampled within the training range, while the *Extrapolate* regime assesses generalization to OOD contexts beyond the training range. The *Mixed* regime evaluates generalization when both context dimensions vary: one is sampled from the extrapolation range and the other from the interpolation range. Results in the Interpolate and Extrapolate regimes are aggregated over cases where one or both context dimensions vary.

## C.2    Environments

**DMC Ball-in-Cup.** This task involves an agent controlling a cup attached to a ball by a string, aiming to swing the ball into the cup, with context parameters gravity and string length. The ball's pendulum-like motion exhibits complex dynamics due to gravity's influence on oscillation frequency and string length's effect on swing amplitude. Extreme gravity values, such as $0.98$ or $19.6$, significantly alter the ball's trajectory, while short or long strings (e.g., $0.03$ or $0.6$) change the swing's frequency and reach, creating complex interactions between these parameters. The interaction amplifies nonlinear effects, and poses significant generalization challenges, as the context encoder must adapt to substantial shifts in motion patterns, particularly in the OOD regime.

**DMC Walker Walk.** This task involves a planar bipedal robot walking forward, with context parameters gravity and actuator strength. Actuator strength scales torque applied to joint motors (e.g., hips, knees), modulating walking speed and stability, while gravity affects balance and ground reaction forces. Despite simple torque scaling, locomotion dynamics are intricate, driven by multi-joint coordination and contact interactions. Generalization is less challenging than in Ball-in-Cup, as the actuator strength evaluation range ($0.1$ to $0.5$ and $1.5$ to $2.0$) is closer to training ($0.5$ to $1.5$), and its torque effects are predictable. However, extreme gravity values test the context encoder's ability to maintain stable, coordinated walking in the OOD regime.

Table 1: Context ranges for considered environments.

| Environment | Context | Training/ Interpolation | Extrapolation |
|---|---|---|---|
| DMC ball_in_cup-catch-v0 | gravity | $[4.9, 14.7]$ | $[0.98, 4.9) \cup (14.7, 19.6]$ |
| | string length | $[0.15, 0.45]$ | $[0.03, 0.15) \cup (0.45, 0.6]$ |
| DMC walker-walk-v0 | gravity | $[4.9, 14.7]$ | $[0.98, 4.9) \cup (14.7, 19.6]$ |
| | actuator strength | $[0.5, 1.5]$ | $[0.1, 0.5) \cup (1.5, 2.0]$ |

## C.3   Architectural details

Figure 3 shows a high-level illustration of the DALI architecture.

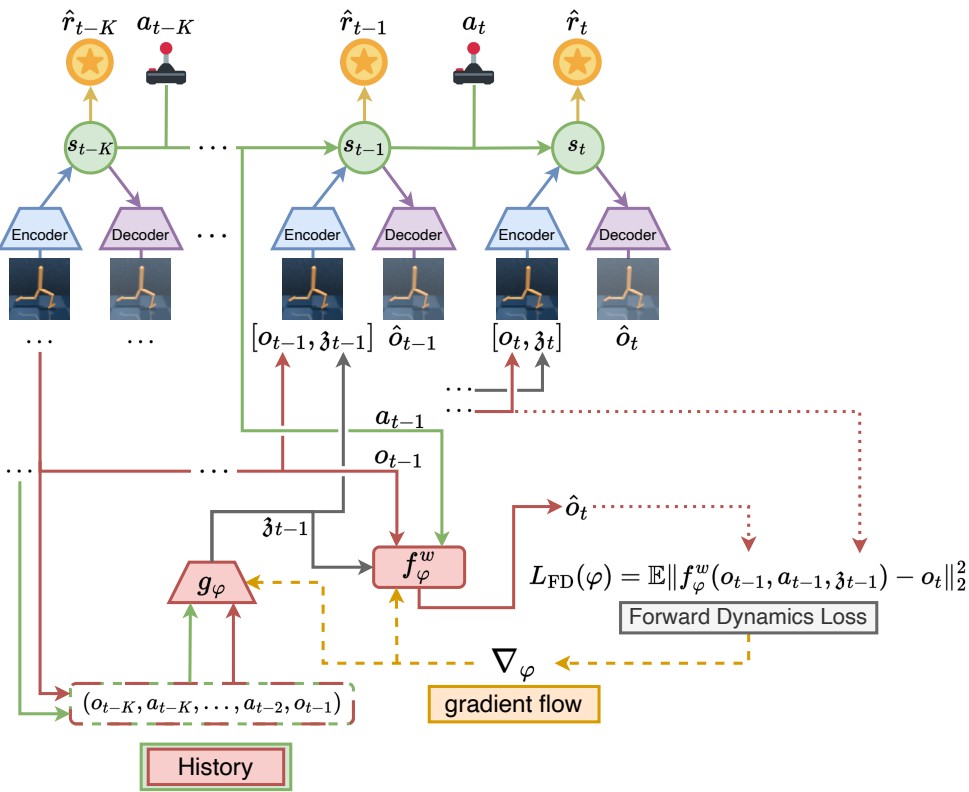

Figure 3: DALI architecture overview.

**Context Encoder.** The context encoder $g_\phi$ employs a standard transformer encoder block [Vaswani et al., 2017] to process a sequence of $K$ transitions, $(o_{t-K:t}, a_{t-K:t-1})$, and produce the context representation $\mathfrak{z}_t \in \mathbb{R}^8$. The input sequence is fed directly into a dense layer with 256 units, bypassing a traditional embedding layer. This is followed by layer normalization and a single-head self-attention block with a skip connection from the dense layer's output. The attention output is combined with the skip connection via a residual connection, followed by a second layer normalization. A two-layer MLP, each layer with 256 units, processes this output, with another residual connection combining the MLP output with the second layer norm's output. A final dense layer projects the 256-dimensional intermediate representation to $\mathfrak{z}_t \in \mathbb{R}^8$. All hidden layers, including the MLP and attention head, use 256 units and SiLU activation functions, consistent with DreamerV3 [Hafner et al., 2025]. No masking is applied in the self-attention mechanism, as the input sequence is complete, enabling full contextual processing to generate $\mathfrak{z}_t$.

**Forward Model.** The forward model $f_\varphi^w$ in (1) for dynamics alignment is a two-layer MLP with 128 units and SiLU activations.

$W_z \in \mathbb{R}^{32 \times 8}$ and $W_{\mathfrak{z}_t} \in \mathbb{R}^{8 \times 32}$ in (2) are learnable parameter matrices that map between the context and state spaces.

**DreamerV3.** We adopt the *small* DreamerV3 variant with hyperparameters from Hafner et al. [2025], following the setup of Prasanna et al. [2024] to ensure fair and reproducible comparison with their cRSSM-S/D baselines.

DALI adds only about 4% parameter overhead (e.g., Dreamer-DR: 15.73M vs. DALI-S: 16.45M) while consistently improving performance with minimal additional complexity.

## C.4 Computational Setup and Resource Requirements

Training and evaluation of the baselines and our DALI approaches were conducted on NVIDIA A100 GPUs with 80GB of VRAM and Intel Xeon Platinum 8352V CPUs. Typically, our setup provides access to 2 GPUs on average, with up to 4 GPUs available in the best-case scenario. Parallelization differs between modalities: featurized experiments can run 10 seeds in parallel, while pixel-based experiments are limited to 4 seeds due to higher memory requirements.

Our complete experimental setup includes 10 seeds across 5 algorithm variants, 2 environments, and 2 modalities, evaluated over 3 context settings (`single_0`, `single_1`, `double`), yielding a total of 600 individual training runs. Table 2 summarizes the computational requirements, where reported GPU hours account for 10 parallel processes in the featurized case and 4 in the pixel-based case. Note that training times per trial are similar for both variants, as the bottleneck lies in environment interaction rather than network size.

In a purely sequential execution, the total computational cost would reach approximately **24,000 GPU hours**. Leveraging parallel execution significantly reduces wall-clock time: in the **best-case scenario** with 4 GPUs, total wall-clock time is approximately **1,051 GPU hours** ($\approx$44 days). In the more typical **worst-case scenario** with 2 GPUs, wall-clock time increases to approximately **2,101 GPU hours** ($\approx$88 days).

| Task | Modality | GPU Hrs per Trial | Runs Total | Wall-Clock (GPU Hrs) 2 GPUs | 4 GPUs |
|---|---|---|---|---|---|
| Ball in Cup | Featurized | 20–26 | 150 | 195 | 98 |
| | Pixel-based | 20–26 | 150 | 488 | 244 |
| Walker | Featurized | 46–54 | 150 | 405 | 203 |
| | Pixel-based | 46–54 | 150 | 1,013 | 506 |
| **Total Featurized** | | – | **300** | **600** | **301** |
| **Total Pixel-based** | | – | **300** | **1,501** | **750** |
| **Grand Total** | | – | **600** | **2,101** | **1,051** |

Table 2: Computational requirements for the full experimental evaluation. Each task is run across 5 algorithm variants with 10 seeds each in both featurized and pixel-based modalities. Wall-clock times assume optimal GPU utilization given the stated parallel capacities.

## C.5 Additional Experiments

Figure 4 shows the learning curves for DMC Ball-in-Cup and Walker Walk tasks under Featurized and Pixel-based observation modalities.

DALI's context encoder adds only about $4\%$ more parameters to DreamerV3-small while scaling effectively to high-dimensional tasks. We evaluate it on **DMC Quadruped Walk** (56D observations, 12D actions), a larger environment than Walker Walk (24D, 6D). The 56D observation space emphasizes the challenge of torque-sensitive locomotion, where increased coordination and a higher number of actuators (12 vs. 6) demand more robust context inference. Trained for 600K timesteps and contextualized with gravity and actuator strength (same ranges as Walker Walk), DALI attains Featurized Extrapolation IQMs of $0.326 \pm 0.043$ (DALI-S), and $0.389 \pm 0.027$ (DALI-D-$\chi$), yielding up to $76.8\%$ improvement over the context-unaware baseline $0.220 \pm 0.023$ (Dreamer-DR), and up to $50.8\%$ improvement over the context-aware baselines ($0.258 \pm 0.028$, cRSSM-D; $0.317 \pm 0.031$, cRSSM-S).

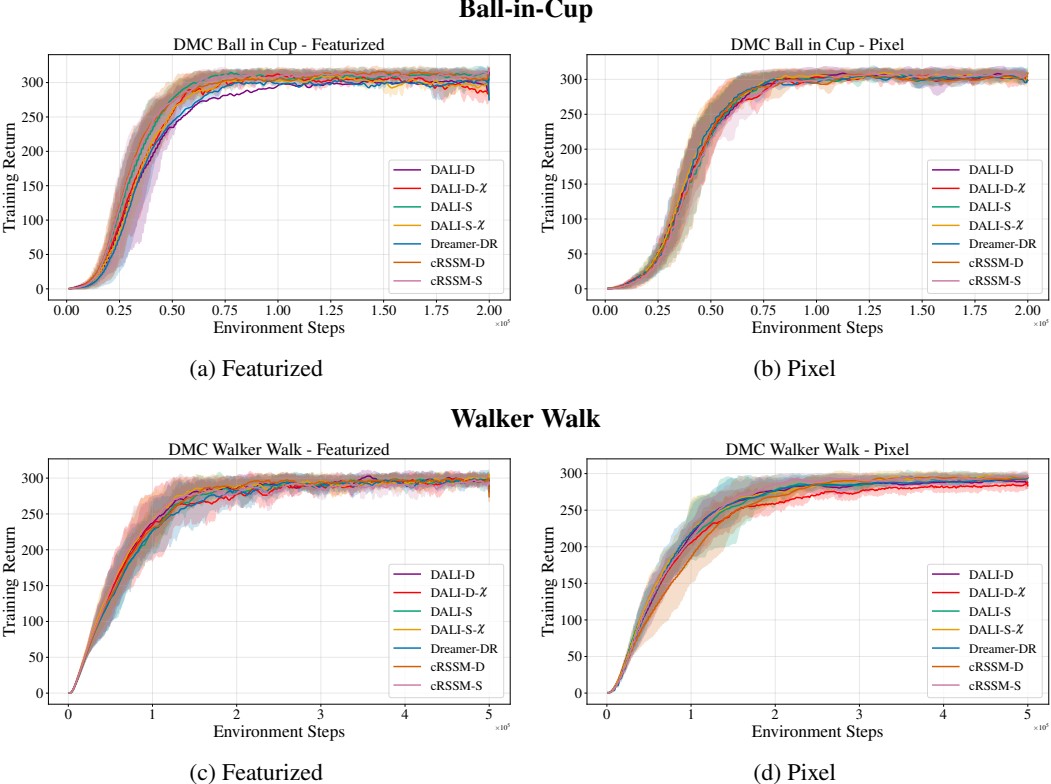

**Figure 4:** Learning curves for DMC Ball-in-Cup and Walker Walk tasks under Featurized and Pixel-based observation modalities. Results show mean episode returns with 25th–75th percentile confidence intervals.

# D Supplementary Experiments on Physically Consistent Counterfactuals

We provide additional details supporting Section 6.3. First, we validate the statistical significance of the influence of latent dimensions on counterfactuals using AUC-based rankings (Section D.1). Next, we analyze actuator-driven recovery dynamics in the DMC Walker Walk task, analogous to the counterfactual analysis conducted for the Ball-in-Cup task in Section 6.3.1 (Section D.2).

## D.1 AUC Rankings and Significance of Counterfactual Perturbations

To identify latent dimensions that systematically alter imagined dynamics, we train a binary classifier to distinguish trajectories generated under perturbed ($\mathcal{T}'^{(j)}$) and original ($\mathcal{T}^{(0)}$) context representations. For each dimension $\mathfrak{z}_j$, we generate $N = 2500$ trajectory pairs, perturbing $\mathfrak{z}_j$ by $\Delta = \sigma(\mathfrak{z}_j)$ to induce counterfactuals (see Section 6.3).

We implement an ensemble classifier comprising a support vector machine, an MLP, and an AdaBoost classifier, trained via stratified 5-fold cross-validation. Inputs are trajectory sequences $\mathcal{T}^{(0)}$ (label 0) and $\mathcal{T}'^{(j)}$ (label 1). Predictions are aggregated across folds to compute AUC. To ensure robustness, we compute 95% bootstrap confidence intervals for each $\mathfrak{z}_j$'s AUC using 500 resamples. Permutation tests (1000 iterations) validate statistical significance between top-ranked dimensions by comparing observed AUC gaps to a shuffled null distribution.

For the Ball-in-Cup task, results are shown in Figure 5, where higher AUCs indicate dimensions whose perturbations induce physically distinguishable dynamics (e.g., gravity or string length changes). Confidence intervals highlight stability across bootstrap samples.

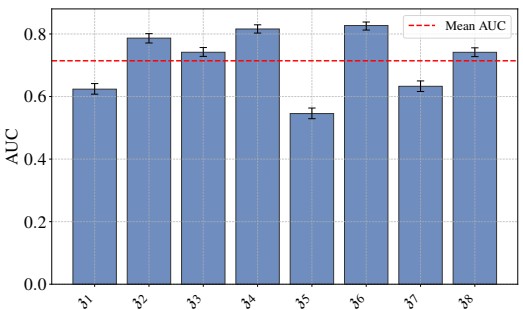

Figure 5: AUC $\pm$ 95% CI per context dimension $\mathfrak{z}_j$ for the Ball-in-Cup task.

## D.2 Actuator-Driven Recovery and Locomotion in Latent Imagination

Analogous to the Ball-in-Cup analysis (Section 6.3.1), we evaluate DALI's ability to generate physically consistent counterfactuals in the DMC Walker Walk task contextualized with gravity and actuator strength. We show that perturbations to the top-ranked latent dimension $\mathfrak{z}_3$ in DALI-S, identified via AUC-based rankings (see Section D.1), influence the Walker's capacity to recover from destabilizing falls and sustain locomotion, indicating that $\mathfrak{z}_3$ is mechanistically aligned with actuator strength. Imagined rollouts are initialized with a real observation and then rolled out in latent imagination. Unlike in the Ball-in-Cup analysis, we retain the original policy actions during imagination to examine how actuator strength modulates the Walker's recovery and locomotion behavior. For the Featurized modality, we tracked low-level state variables (e.g., torso height) at each timestep, while in the Pixel modality, we captured rendered environment frames at 5-step intervals over a 64-step imagination horizon.

**Pixel Modality.** Figure 6a contrasts the original (top) and counterfactual (middle) trajectories. Both agents initially fall (frames 0-20), but the perturbed $\mathfrak{z}_3$ trajectory (counterfactual) exhibits enhanced actuator strength, enabling the Walker to stand from a challenging pose (left leg extended, right leg bent in frames 30-45) and locomote forward (frames 50–64). The original trajectory fails to recover, collapsing after the fall. Pixel-wise differences (bottom) highlight kinematic deviations, particularly in leg articulation and torso alignment.

**Featurized Modality.** Figure 6b shows torso height dynamics. While both trajectories share similar initial descent (time steps 0-20), the counterfactual (orange) achieves sustained elevation (after time step 20), reflecting successful recovery and locomotion. This aligns with the pixel-based evidence of increased actuator strength, confirming $\mathfrak{z}_3$'s role in encoding torque dynamics critical for stability.

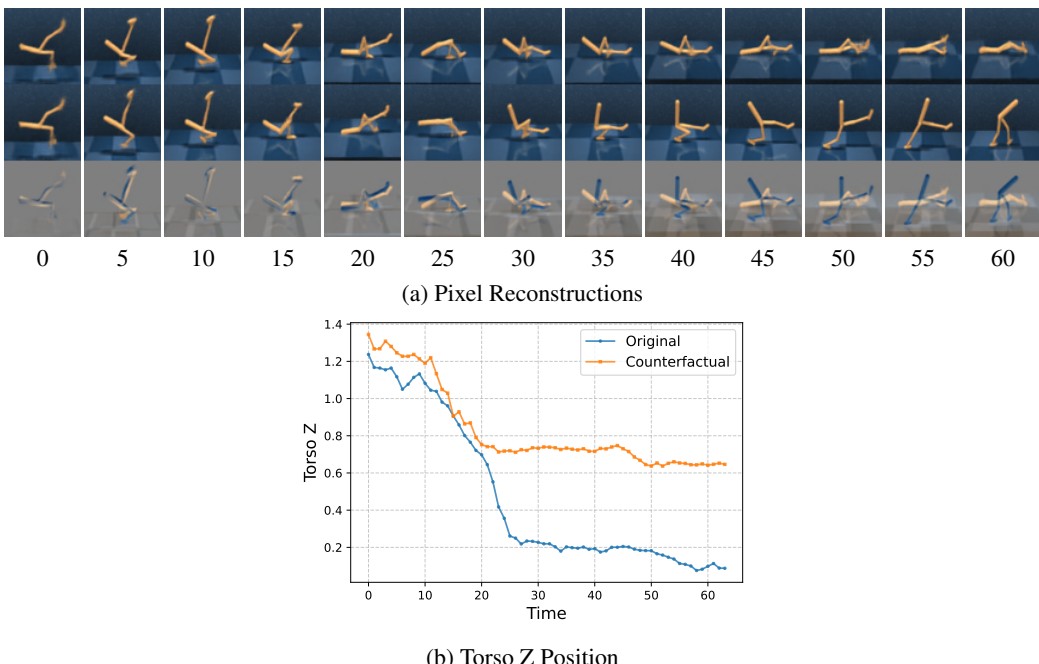

(a) Pixel Reconstructions

(b) Torso Z Position

Figure 6: (a) **(Pixel Modality) Counterfactual Trajectories in Pixel Space**: Top: Original imagined trajectory of the Walker Walk under default gravity and actuator strength. Middle: Perturbed trajectory after adding noise $\Delta$ to the top-ranked latent dimension $\mathfrak{z}_3$. Original trajectory shows the Walker failing to recover after a fall. The perturbed trajectory with enhanced actuator strength enables recovery and forward locomotion. Bottom: Pixel-wise differences ($\delta = |\hat{o}_t - \hat{o}'_t|$). The counterfactual agent refines leg kinematics (frames 30-45) and stabilizes upright (frames 50-64). (b) **(Featurized Modality) Torso Z-Position Under Latent Perturbation**: Comparison of original (blue) and counterfactual (orange) torso heights. The perturbed $\mathfrak{z}_3$ trajectory maintains higher elevation after time step 20, reflecting successful stabilization and locomotion.

