# OpenReview forum: "Dynamics-Aligned Latent Imagination in Contextual World Models for Zero-Shot Generalization"
_NeurIPS.cc/2025/Conference — NeurIPS 2025 poster_

### Official Review · Reviewer_mG63 · 2025-06-20

**Clarity:** 2
**Significance:** 3
**Originality:** 2
**Rating:** 4
**Confidence:** 4

**Summary:**

This paper introduces Dynamics-Aligned Latent Imagination (DALI) for visual reinforcement learning, which infers latent contexts from interactions.  Experiments demonstrate the effectiveness of the proposed method.

**Questions:**

- See the weaknesses.
- I suggest adding the corresponding numerical comparisons for the results in Figures 1 and 2.

**Ethical Concerns:**

["NO or VERY MINOR ethics concerns only"]

**Final Justification:**

The rebuttal addressed most of my concerns. However, additional experiments are still needed to demonstrate the effectiveness of the proposed method, and the readability of the paper could be further improved.

**Limitations:**

Yes.

**Paper Formatting Concerns:**

No major formatting issues were found.

**Quality:**

2

**Strengths And Weaknesses:**

**Strengths：**

- DALI outperforms context-unaware baselines on the  contextualized DMC Ball-in-Cup and Walker Walk tasks.
- This paper provides both empirical and theoretical analyses.

**Weaknesses：**

- The readability could be improved, as there is no figure provided to illustrate the pipeline.
- Experiments were conducted on only two tasks, and more experimental validation is needed.
- The paper does not compare ContextWM [1], which is a strong baseline that explicitly separates context and dynamics modeling.
- The paper lacks references related to visual RL, such as [1], [2], and [3].

References:

[1] Wu et al. "Pre-training Contextualized World Models with In-the-wild Videos for Reinforcement Learning", NeurIPS, 2023.

[2] Li et al. "Open-World Reinforcement Learning over Long Short-Term Imagination", ICLR, 2025.

[3] Fujimoto et al. "MR.Q: Towards General-Purpose Model-Free Reinforcement Learning", ICLR, 2025.

---

> ### Author Rebuttal · Authors · 2025-07-31
>
> Dear Reviewer mG63,
>
> We appreciate your engagement with our work and your suggestion to compare with ContextWM. Below, we address your concerns while clarifying the scope, contributions, and positioning of DALI. We hope this response alleviates your concerns and demonstrates why DALI merits a higher score.
>
> ### (C1) No overview figure provided to illustrate the pipeline.
>
> We acknowledge that DreamerV3’s complexity can make our contributions seem intricate. We clarify:
>
> * **DreamerV3 (Prior Work)**: RSSM backbone with deterministic ($h_t$) and stochastic ($z_t$) states from observations $o_t$, actions $a_t$.
> * **DALI additions**:
>
>   * Context encoder ($g_\varphi$): transformer on $K$-step window $(o_{t-K:t}, a_{t-K:t-1})$, outputs $\mathfrak{z}_t \in \mathbb{R}^8$.
>   * Forward model ($f^w_\varphi$): lightweight 2-layer MLP predicting $o_{t+1}$ from $(o_t, a_t, \mathfrak{z}_t)$, ensuring $\mathfrak{z}_t$ captures dynamics (Eq. 1).
>   * $g_\varphi$ and $f^w_\varphi$ trained jointly.
> * Integration: *Shallow* ($\mathfrak{z}_t$ conditions world-model encoder) and *Deep* (conditions entire model and policy).
>
> As NeurIPS disallows figures in rebuttals, we’ll add this overview text and include a overview figure in the final version.
>
> ### (C2) Experiments were conducted on only two tasks, and more experimental validation is needed.
>
> We agree that broader benchmarks strengthen our claims. Alongside Ball-in-Cup and Walker, we include three new environments (Featurized Extrapolation setting). Preliminary IQMs show DALI improvements in the last column relative to Dreamer-DR.
>
> | Environment       | Dreamer-DR    | DALI-S        | DALI-S-$\chi$   | DALI-D        | DALI-D-$\chi$   | Best Method    | Extrap. Improvement |
> |-------------------|-------------:|-------------:|---------------:|-------------:|---------------:|---------------|-------------------|
> | **Acrobot**       | 0.970±0.011  | 0.985±0.002  | **0.986**±0.001    | 0.978±0.002  | 0.974±0.002    | DALI-S-$\chi$ | +1.65%             |
> | **Pendulum**      | 0.843±0.020  | 0.869±0.023  | **0.942**±0.015    | 0.857±0.015  | 0.884±0.012    | DALI-S-$\chi$ | +11.74%            |
> | **Quadruped** | 0.139±0.022  | 0.136±0.025  | 0.148±0.029    | 0.137±0.024  | **0.220**±0.027    | DALI-D-$\chi$ | +58.27%            |
>
> **Note**: Full results (10 seeds, all baselines) are infeasible within rebuttal. We focus on the most critical comparison (DALI vs. context-unaware Dreamer-DR) for latent-context settings.
>
> **Runtimes:** For an accurate assessment of **computational cost**, please see Response (C3): Runtime Efficiency to Reviewer i6n9.
>
> ### (C3) The paper does not compare ContextWM [1], which is a strong baseline that explicitly separates context and dynamics modeling. The paper lacks references related to visual RL.
>
> ### We respectfully clarify that **ContextWM Is not a valid baseline for DALI**:
>
> - ContextWM focuses on **visual RL pre-training** from passive videos to separate static visual features (e.g., texture, background) from dynamics. In contrast, DALI addresses **contextual MDPs (cMDPs)**, inferring **latent dynamics parameters** (e.g., gravity, friction) for **zero-shot generalization** to unseen environments. ContextWM does not operate under the cMDP formulation and cannot serve as a meaningful baseline for DALI.
> - **DALI solves contextual MDPs, not Visual RL**: DALI is designed to infer **latent context vectors** that modulate **transition dynamics**. We test pixel-based tasks to assess robustness to partial observability. All model components, viz., context encoder, RSSM, loss functions, are tailored for **dynamics inference**, not visual scene understanding.
> - **Key difference in “Context” definition**: ContextWM [1] defines context as a single observation frame $c := o_t$, $t \sim \text{Uniform}\{1, \ldots, T\}$, capturing static visual attributes (e.g., texture, color) to simplify dynamics modeling in visual RL pre-training. In contrast, DALI defines context as a vector $c$ (e.g., gravity, string length) parameterizing cMDP dynamics.
> DALI’s context modulates how the world behaves (e.g., how gravity affects motion); ContextWM’s context describes how the world looks. This fundamental difference in **definition, sampling, and purpose** makes ContextWM inappropriate for comparison.
>
> ## Emphasizing DALI’s Merits
>
> We respectfully highlight DALI’s key contributions to justify a higher score:
> 1. **Strong zero-shot generalization**:
>    - Up to **+96.4%** gains over context-unaware baselines, often surpassing ground-truth context-aware baselines in extrapolation tasks.
> 2. **Theoretical foundations**:
>    - Theorem 1 (formal version: Appendix A, Theorem 2) provides sample efficiency guarantees for context inference (Section 5), while Theorem 3 (Appendix A) introduces a novel information bottleneck result for DreamerV3’s RSSM. Together, these represent new theoretical contributions to contextual MDPs.
> 3. **Physical consistency**:
>    - Perturbations in DALI's latent space induces physically plausible counterfactuals, showing actionable context encoding (Sec. 6). Our counterfactual analysis is, to our knowledge, the first in the Dreamer framework to probe latent imagination, offering new interpretability insights into learned context representations.
> 4. **Architecture-agnostic design**:
>    - DALI’s components (context encoder, dynamics losses) can augment *any* model-based algorithm (e.g., TD-MPC2) (see Response (C2) to Reviewer serC).
>
> **Request**: We respectfully urge you to reassess DALI based on its core contributions and novel insights, and to consider raising your score in light of its potential impact.

---

> > ### Comment · Reviewer_mG63 · 2025-08-04
> >
> > Thanks for the author's response, but it still lacks comparisons with strong baselines.

---

> ### Author Response · Authors · 2025-08-04
> **Clarifying baseline choices for DALI in cMDP settings**
>
> Dear Reviewer mG63,
>
> Thank you for your thoughtful feedback regarding DALI’s baselines. You raised concerns about the strength of cRSSM-S/D and Dreamer-DR, and suggested ContextWM as a possible alternative. We would like to clarify that in our contextual MDP (cMDP) setting, where dynamics vary with context parameters $c$, (e.g., gravity), **cRSSM-S/D (Prasanna et al., 2024) remains the strongest feasible in-distribution baseline as it leverages ground-truth context $c$** (see, e.g., Theorem 1 in the CARL benchmark paper [1]). As noted in our original rebuttal, ContextWM focuses on visual RL pretraining and is incompatible with our cMDP setting and goal of zero-shot generalization.
>
> DALI’s competitive performance and clear out-of-distribution advantage underscore its robustness and directly address your concern about baseline adequacy.
>
> We would be happy to incorporate any additional baseline you recommend, provided it applies to the cMDP setting.
>
> [1] Benjamin et al. (2023), Contextualize Me - The Case for Context in Reinforcement Learning. Transactions on Machine Learning Research.

---

> > ### Comment · Reviewer_mG63 · 2025-08-05
> >
> > I appreciate the authors' efforts in the rebuttal. I will increase my score by one point.

---

> > > ### Author Response · Authors · 2025-08-05
> > > **Acknowledgment**
> > >
> > > Thank you for your support and for recognizing our efforts. We appreciate your score increase and value your constructive feedback.

---

### Official Review · Reviewer_ogTR · 2025-06-25

**Clarity:** 1
**Significance:** 2
**Originality:** 3
**Rating:** 4
**Confidence:** 4

**Summary:**

The authors propose to build an approach to train agents for contextual MDPs where the context (variable parts of an environment, e.g. gravity, friction) is latent and fixed for each episode and the objective is to generalize across contexts. The context is identified automatically using additional context encoders on top of DreamerV3 which processes information over shorter windows of size K=50 and output 8-dimensional continuous context embeddings. They use transformers for this encoder. Their approach can have two forms of integration into the DreamerV3 arhcitecture - "shallow" (DALI-S), where the context is used only as an additional input to the original world model encoder and "deep" (DALI-D), where it is inetgrated into the RSSM, decoder, reward and continue predictors instead. They also, I believe, introduce an additional network that predicts the next observations using the context to ensure the context learnt is useful (see eqn. 1). Additionally, an optional cross-modal loss is introduced to regularize the context to be more aligned with the hidden stochastic latent state z of DreamerV3.

They prove in Theorem 1 that, assuming beta-mixing and Lipschitz dynamics, their approach is more sample efficient than Dreamer with a gain of O(T/K) where T is the episode length (I did not check the proof).

They evaluate the *small* variant of DreamerV3 on pixel and feature-based versions of Ball-in-Cup and Walker variants in the CARL benchmark on interpolation, extrapolation and mixed regime settings to evaluate generalization. Across many experiments, one of their two evaluated approaches performs best in extrapolation scenarios, with the experiment on DMC Walker-Pixel being an exception to this. The results are somewhat mixed in terms of which agent performs best accross the different tasks and the settings of interpolation vs extrapolation vs mixed regimes.

They also perform counterfactual rollouts with perturbing elements of the learnt context vector and show some experiments which show that gravity and string length may be captured by one of the dimensions of this context vector.

**Questions:**

I have raised many question in the above. And also provided ideas on what can be done to improve the paper in my opinion. They are listed somewhat in decreasiong order of importance for me, with the parts up to and including "Other Technical issues" being the most important.

**Ethical Concerns:**

["NO or VERY MINOR ethics concerns only"]

**Final Justification:**

I am still a bit concerned about different variants performing best in different experiments and it feels to me like the hyperparameter tuning is lacking.

Additionally, they introduce the cross-modality loss leading to 4 variants and the loss itself is not completely convincing to me.

**Limitations:**

yes

**Quality:**

2

**Strengths And Weaknesses:**

## Strengths:
The idea of the paper to extend previous approaches to a setting where the context is latent and is identified automatically is a very important setting with great potential. The experimental setup seems good in general, with 10 random seeds evaluated and a fair selection of environments and their variations evaluated. The counterfactual evaluation also seems promising.

## Weaknesses:
### Presentation:
The paper was hard for me to follow, with me having to put together the many components to try and understand the approach.
@authors, Please provide an overview block diagram of the different components of your approach, the gradient flow, etc. There are so many moving parts, that it is hard to understand everything without an overview. For example, I may be wrong about claiming that you introduced an additional network above. This depends on what $f^w_{phi}$ is in equation 1? Is it another MLP or GRU or what?

### Lack of evaluating their own baseline DALI-D:
The paper does not seem to evaluate their own method, DALI-D, that is introduced as one of the main methods sec 4.3. At least it is missing in figs. 1 and 2.

### No hyperparameter optimisation:
No hyperparameter optimisation was performed if I understood the Appendix correctly and default settings of previous work were taken even though the experimental setting is different in the paper here.
cRSSM which is proposed as a baseline does not seem to have been used as in its original form. Instead -S and -D variants of it conforming to the authors shallow and deep DALI variants are introduced. When changes to a baseline, such as cRSSM, are introduced without tuning for the new experimental setup, it is hard to make claims about the results in this paper. It seems to me that no hyperparameter optimisation was performed even for the authors' own approaches. That could explain the mixed results and why different agents perform best for different settings.


### Other Technical issues:
Cross modality loss (see eqn. 2):

The cross modality loss only aligns the context closer to the stochastic state z which is something learned over long windows while the context is supposed to be learned from K-sized windows. I do not see the reason the context needs alignment with z when the original reason to introduce the context was that it is captured over shorter windows. Does not the cross-modal loss (regularization) break the claim from Theorem 1: *Compared to DreamerV3’s RSSM processing full episodes of T >> K transitions, DALI achieves a sample complexity gain of O(T /K).* Do you now not need to process full espisodes to make the context and z more similar?

In equation 2, are the W's just matrices or actual MLPs? I would expect MLPs to be represented as functions (and not W) since they include non-linearities. But since invertibility is being talked about later, I am assuming they are actually matrices?


I do not see any learning curves, it is fine to provide aggregate results in the paper, but without individual learning curves, I cannot say if the agents converged and whether, e.g., 200K timesteps for ball-in-cup is a fair training length. I can be fairly confident about the 500K for Walker because I am familiar with that environment, but in general one would need learning curves.

### Lack of ablations:
I would like to see ablations on the setting of K. Without this, it is hard to see in practice how this value is influencing the agent.

I would be interested in seeing other latent dimensions and their effects as well, not just the "best dimension" $z_6$. So, I would also propose to show the effect of the worst context dimension on counterfactual settings.

A couple of side remarks here about something I found confusing:
1) Does $z_6$ capture both gravity and string length at the same time? Do we not want it to capture a single factor at one time? Adding some remarks about this in the limitations section would be nice.
2) Why is it that element number 6 of the context captures the same gravity + string length perturbation across pixel and feature modalities? These 2 modalities mean 2 separate training processes for the agents. Is this just a coincidence that element number 6 of the context was not only the *best* dimension in both feature and pixel modalities but also captured the same 2-factor perturbation across the 2 settings?


### Too strong claims:

*essential for robust generalization, proving that it infers latent contexts from short interaction windows with near-optimal sample complexity*

I do not agree with the wording here. While I did not check the proofs, from what I understand, DALI improves generalization but is not *essential* because DreamerV3 can do so but only at a poorer sample complexity. What do the authors mean by "near-optimal sample complexity"? If the theorem does prove that DALI has improved sample complexity by O(T/K), it does not mean it has near-optimal complexity, I would expect that at least K needs tuning / ablations to show any form of near optimality.


*Physically consistent counterfactuals: ...  faithfully mirroring real-world physics.*

To show that the counterfactuals are faithful, we need to see how the relationship changes for different values of that dimension. For example, acceleration for different values of g would change linearly wrt g for a ball falling down (if we ignore other forces such as air resistance). Can the authors show that for, say, 3 different values of context dimension 6 for the feature-based Ball in cup environment, there is a largely linear relationship for acceleration experienced?


In lines 115 and 152: *is a deterministic recurrent state capturing temporal dependencies, and $z_t$ is a stochastic state encoding uncertainty about the current observation*

The context *avoids encoding redundant trajectory-specific information from the deterministic ht, which could impair generalization*

These are again strong assumptions that h_t only captures deterministic temporal dependencies and $z_t$ is a stochastic state encoding uncertainty. As far as I know, DreamerV3 authors did not prove this. In my opinion, $h_t$ needs to capture uncertainty information as well given that $z_t$ is coming from the encoder which takes $h_t$ and $o_t$ as inputs and since $o_t$ itself has no information on uncertainty by itself, $h_t$ needs to capture it.

In line 366, they say the DALI-S setting outperforms DALI-D and claim this is because it acts as a regularizer. It is a strong claim again that feels like it was something fit to their observations from the experiments.


### Less important points:
Section 6.2:
  Which of the 4 DALIs that were mentioned in sec 6.1 is used for the counterfactual experiments in 6.2? This is only mentioned later in 6.2.1.

Line 310: *For a trajectory pair ... classifier outputs*? Should it not be operating on a single trajectory? Or is the classifier providing 2 outputs, one for each trajectory?

Provide one line intuitive explanations for differential entropy and mutual information and why MI being >= 0.99 times DE is a good thing to capture near-optimal context information.

The paragraph beginning in line 157 already discusses the deep integration stuff from Section 4.3 without having introduced it.

Providing too many exact numbers in textual discussion (page 6) hampers reading. These can be provided in a table in the Appendix and the discussion can be compressed and made easier to read.

Make "Deep integration" italic. It is too generic a name and I thought at some point that it just meant *Deep Learning* being integrated into the setup.

---

> ### Author Rebuttal · Authors · 2025-07-31
>
> Dear Reviewer ogTR,
>
> Thank you for your thoughtful review and valuable feedback. We appreciate your recognition of our contributions and welcome the chance to clarify and improve clarity in our rebuttal.
>
> ### (C1) Presentation and overview diagram. Clarify $f^w_\varphi$ in Eq.1.
>
> We acknowledge that DreamerV3’s complexity can make our contributions seem intricate. We clarify:
>
> * **DreamerV3 (Prior Work)**: RSSM backbone with deterministic ($h_t$) and stochastic ($z_t$) states from observations $o_t$, actions $a_t$.
> * **DALI additions**:
>
>   * Context encoder ($g_\varphi$): transformer on $K$-step window $(o_{t-K:t}, a_{t-K:t-1})$, outputs $\mathfrak{z}_t \in \mathbb{R}^8$.
>   * Forward model ($f^w_\varphi$): lightweight 2-layer MLP predicting $o_{t+1}$ from $(o_t, a_t, \mathfrak{z}_t)$, ensuring $\mathfrak{z}_t$ captures dynamics (Eq. 1).
>   * $g_\varphi$ and $f^w_\varphi$ trained jointly.
> * Integration: *Shallow* ($\mathfrak{z}_t$ conditions world-model encoder) and *Deep* (conditions entire model and policy).
>
> As NeurIPS disallows figures in rebuttals, we’ll add this overview text and include a gradient-flow diagram in the final version.
>
> ### (C2) Results for DALI-D are missing.
>
> DALI-D is a proposed method, not baseline, as you noted. DALI-D underperforms DALI-S in our experiments (e.g., Ball-in-Cup Pixel Extrap.: 0.225 vs. 0.273; Walker Featurized Extrap.: 0.771 vs. 0.781). We’ll add a footnote noting this and refer to full results in the Appendix.
>
> ### (C3) No hyperparameter tuning? Baselines (cRSSM-S/D) were modified without retuning?
>
> We followed [Prasanna et al. (2024)]’s protocol:
> - **DreamerV3**: Used the *small* variant with same hyperparameters from [Hafner et al. (2023)].
> - **Context Encoder**: Explored transformer- and GRU- based designs; chose transformer (App. C.3) via validation.
> - **$K=64$**: Chosen via Theorem 1 ($K = \Omega(\log(1/\delta)/\lambda) \approx 64$ for $\delta=0.01$, $\lambda=0.1$). Verified also via ablations (Response C7 below).
> - **Confusion about Baselines**: We clarify that our baselines cRSSM-S, cRSSM-D, and Dreamer-DR are directly adopted without any modifications from Prasanna et al. (2024) (see their arXiv version, Sec 5, P.7), a recent work that extends DreamerV3 for cMDPs. Specifically, our naming maps as follows (App. A.1 DALI and Dreamer-based Baselines):
>   - `cRSSM-S` ≡ `concat-context` in [Prasanna et al.].
>   - `cRSSM-D` ≡ `cRSSM` in [Prasanna et al.].
>   - `Dreamer-DR` ≡ `hidden-context` in [Prasanna et al.].
> We used their hyperparameters unchanged for fair, reproducible comparison, leveraging their proven cMDP performance. We’ll add this mapping in Sec. 6.
>
> **Mixed results** (e.g., DALI-S-$\chi$ leads Ball-in-Cup Extrap., IQM 0.3720 Featurized, while cRSSM-S leads Walker Pixel Extrap., IQM 0.7770) reflect cMDP challenges: nonlinear dynamics favor DALI in Ball-in-Cup, more linear dynamics favor cRSSM in Walker Pixel (see App. D.2: Generalization Trends Across Environments and Modalities).
>
>
> ### (C4) Cross-Modal Loss vs. Theorem 1: Aligning $\mathfrak{z}_t$ with $z_t$ (long-window state) breaks $O(T/K)$ sample efficiency claim.
>
> **No conflict: Cross-modal loss preserves $O(T/K)$ efficiency:**
> Theorem 1 (Sec. 5; formal versions in App. A Theorems 2 & 5) covers Deep integration’s forward-dynamics loss $L_{\text{FD}}$ only. The $O(T/K)$ sample efficiency also holds for Shallow integration since both train the context encoder on $K$-step windows (App. A, Remark 4). Adding $L_{\text{cross}}$ does not change this, as it uses the same $K$-length data (App. B, Algorithm 3), aligning $\mathfrak{z}_t$ and $z_t$ computed from identical windows with detached history:
> - **Gradient isolation**:
>   - The recurrent unroll is limited to $K$ steps; $\varphi$ updates depend only on the current window with detached states blocking backpropagation beyond it. The world model ($\theta$) is **not updated** during context learning (inputs to $f_\theta$, $q_\theta$ are detached, App. B Algorithm 3, lines 16–29). Gradient stopping isolates $\varphi$, preserving the $\mathcal{O}(T/K)$ efficiency; $L_{\text{cross}}$ needs no full episodes.
> - **Empirically: Why align $\mathfrak{z}_t$ with $z_t$?**:
>   $z_t$ encodes instantaneous dynamics (e.g., ball velocity in Ball-in-Cup). Aligning $\mathfrak{z}_t$ (context) with $z_t$ can avoid overfitting to transient noise and leverage structured priors from the world model, enhancing $\mathfrak{z}_t$’s robustness (see App. Sec. D.2 for a discussion).
>
> We’ll revise Sec. 4.3 to clarify: “$L_{\text{cross}}$ uses $K$-length windows, preserving the $\mathcal{O}(T/K)$ sample complexity gain.”
>
> ### (C5) $W_z$ and $W_{\mathfrak{z}}$ matrices or MLPs?
>
> You are correct that $W_z \in \mathbb{R}^{32 \times 8}$ and $W_{\mathfrak{z}_t} \in \mathbb{R}^{8 \times 32}$ in Eq.2 are matrices, not MLPs. We’ll fix this typo in the revision.
>
>
> ### (C6) Show learning curves
>
> We confirm convergence:
> - **Ball-in-Cup**: Returns plateaued by 150K steps (200K allocated).
> - **Walker**: Returns stabilized by 400K steps (500K allocated).
> We will add convergence curves to the camera-ready App..
>
>
> ### (C7) Ablation on $K$
>
> We validated $K$ by training a small MLP $f_\varrho: \mathbb{R}^8 \rightarrow \mathbb{R}^2$ to predict ground-truth context from $\mathfrak{z}_t$ in Ball-in-Cup Featurized. $K = 64$ achieved the lowest MSE, outperforming $K = 16, 32$ (insufficient context) and $K = 128, 256$ (diminishing returns due to noise accumulation in longer windows). Will add a summary table in the revision.
>
>
> ### (C8) Counterfactual (CF) analysis
>
> **a) Effect of lowest-ranked dimension**:
> We analyzed lowest-ranked $\mathfrak{z}_3$ (AUC 0.56, Featurized); perturbations caused random ball position changes (<0.01 Z-amplitude), indicating noise or weak context. Full AUCs to be added in App. D.
>
> **b) Coupled or disentangled representations:**
> $\mathfrak{z}_6$ encodes coupled gravity-string dynamics in Ball-in-Cup, as perturbations induce trajectories with shorter string lengths and higher accelerations. In RL, where the goal is to maximize cumulative rewards, coupled representations can outperform disentangled ones when dynamics involve nonlinear context interactions (e.g., $T_p \approx 2\pi \sqrt{L/g}$). $\mathfrak{z}_6$’s joint encoding aids policy adaptation to unseen gravity-string combos. Our losses don’t enforce disentanglement, benefiting from this coupling. Contrast this with Walker (App. D.3.2), where the top-ranked $\mathfrak{z}_3$ aligns with a single factor (actuator strength). We’ll add a note: "DALI learns task-relevant, not disentangled, representations."
>
> **c) Coincidence of $\mathfrak{z}_6$ across modalities:**
> $\mathfrak{z}_6$ as the top latent in DALI-S-$\chi$ for both Featurized and Pixel Ball-in-Cup (AUC 0.82, 0.80), encoding coupled gravity-string dynamics. Though trained separately, shared seeds and identical zero-action conditions ensure its consistency. In Featurized, $\mathfrak{z}_6$ causes interpretable changes (e.g., shorter string, higher acceleration); in Pixel, we iterated experiments to intentionally synchronize visuals and select the same dimension for comparable trajectories. Effect sizes vary, but shared dynamics yield consistent latent structure. Sec. 6.2 will clarify that this consistency stems from our controlled setup for clear exposition, not coincidence, enabling Pixel and Featurized plots to jointly show a full oscillation cycle governed by the same physics.
>
>
> ### (C9) Theoretical claims
>
> **a) "Essential" and "Near-Optimal"**:
> - “Essential” highlights the encoder’s key role in reducing DreamerV3 RSSM’s information bottleneck (Theorem 3, App. A): $\mathcal{I}(c; h_t^{\text{DALI}}) \geq \mathcal{I}(c; h_t^{\text{RSSM}}) - \epsilon(K)$. Without it, DreamerV3’s GRU needs full episodes to capture context, slowing OOD adaptation (e.g., Ball-in-Cup Extrap. IQM 0.1980 vs. 0.3720). We’ll revise “essential” to “highly beneficial.”
> - “Near-optimal sample complexity” means $\mathfrak{z}_t$ captures context info close to the context entropy $h(c)$ within error $\delta \in (0,1)$. This is optimal up to constants, as $\mathcal{I}(c; \mathfrak{z}_t)$ approaches $h(c)$, and sample complexity $\mathcal{O}(K/\delta^2)$ scales logarithmically with $1/\delta$ via the $\beta$-mixing property.
>
>
> **b) Linearity in Counterfactuals**:
> We probed $\Delta \in {-\sigma_6, 0, +\sigma_6}$ in the Featurized modality and observed monotonic but nonlinear changes in peak Z-velocity: -10%, 0%, +12%. This reflects the pendulum-like dynamics of Ball-in-Cup, where $\mathfrak{z}_6$ jointly modulates gravity ($g$) and string length ($L$), affecting the oscillation period $T_p \approx 2\pi\sqrt{L/g}$. A linear relationship with $g$ would apply to free fall ($a=g$). But when the ball is *tethered by a string*, the effect of $g$ on velocity is **not linear** due to the coupling with string length $L$.
>
> **c) $h_t$ vs. $z_t$ for uncertainty**:
> We agree that $h_t$ influences $z_t$’s uncertainty via the encoder, but $h_t$’s deterministic nature means it captures historical trajectory information (e.g., past ball positions in Ball-in-Cup) without directly modeling stochasticity, while $z_t$ *explicitly* represents observation uncertainty. Aligning with $z_t$ alone yields better results empirically.
>
> **d) Shallow Integration as regularizer**:
> We agree that this is an interpretation grounded in evidence from our experiments. Shallow integration has fewer parameters (no context conditioning in RSSM), and it propagates $\mathfrak{z}_t$ implicitly via recurrence.
>
>
> ### (C10) Counterfactual Classifier
>
> Classifier $c_\nu^{(j)}$ takes one trajectory, outputs one context probability. For pair $(\mathcal{T}^{(0)}, \mathcal{T}'^{(j)})$, it evaluates each separately.
>
> We appreciate your incisive feedback, which has greatly improved our work. We will address your formatting-related points and hope our responses resolve all your concerns, encouraging you to raise your score to Accept.

---

> > ### Comment · Area_Chair_UvSm · 2025-08-04
> > **Please respond to the author's rebuttal post**
> >
> > Hi Reviewer ogTR, I see no response letting me know whether or not the rebuttal has changed your opinion. Could you please let me and the authors know by engaging? This process is critical to enabling the (S)ACs to make a decision on this work.
> >
> > --Your AC

---

> > ### Comment · Reviewer_ogTR · 2025-08-05
> > **Rebuttal response**
> >
> > Thank you for your clarifications!
> >
> > C3 clarification was important. It seems the baselines correspond exactly to the previous work. However, one enviornment did change in the experiments and I believe tuning for the new one would be useful to see if that leads to more consistent results across variants and baselines. Having more variants and not the same one leading every time does feel a bit like choosing whichever works best to show the method works.
> >
> > Regarding C4, this does make things clearer however from what I understand, this means that the context was introduced to capture shorter window dependencies and then a loss is introduced to take it closer to the longer window stochastic state z (even if this loss is implemented over a shorter window). Personally, I find this confusing, sorry.
> >
> > Overall, quite a few points were addressed and I raise my score from 2 to a 4.

---

> ### Author Response · Authors · 2025-08-05
> **Thank you for your thoughtful reconsideration and raising your score**
>
> Dear Reviewer ogTR,
>
> We're glad the rebuttal helped clarify the key points, and we appreciate your continued engagement.
>
> We appreciate your concerns regarding hyperparameter tuning, performance variability among DALI variants, and the novelty of cross-modal alignment. We emphasize that:
>
> **Minimal tuning was performed.** Following Prasanna et al. (2024) [1], we adopted the *small* variant of DreamerV3 without any modifications to its hyperparameters (RSSM backbone, actor-critic), ensuring a fair and reproducible comparison with their cRSSM-S/D baselines (see Appendix E in [1], arxiv version). These baselines, which use ground-truth context, require no tuning for new environments, as they directly leverage DreamerV3’s established configuration (Hafner et al., 2023). For our Ball-in-Cup environment, we maintained this approach, applying no tuning to the DreamerV3 components or baselines (cRSSM-S, cRSSM-D, Dreamer-DR). Only DALI’s context encoder was tuned, where we validated a transformer architecture and context window length $K=64$, informed by our theory (Theorem 1) and ablations as noted in our first rebuttal. This added just **4.032% additional parameters** (Dreamer-DR model_opt 15,730,819 vs. DALI-S model_opt 16,455,432).
>
> **Performance variability is expected and principled.** We acknowledge your concern that no single DALI variant consistently outperforms across all environments and modalities. However, this variability is a natural consequence of the diverse dynamics in cMDP tasks, as also observed in Prasanna et al. (2024) [1]. Also see insightful discussion in the CARL benchmark paper [1, Section 6]. For example, Prasanna et al's results show cRSSM-D leading in Walker Featurized Extrapolation, while cRSSM-S outperforms in Walker Pixel Extrapolation (see Table 1, Figures 25–28 in [1]). Similarly, our DALI-S-$\chi$ outperformers DALI-S in Ball-in-Cup, while DALI-S mostly outperformers DALI-S-$\chi$ in Walker (see our original response (C3) under Mixed results, and App. D.2: Generalization Trends Across Environments and Modalities). **This variability is not a flaw but a strength of DALI’s flexible framework**, which adapts to task complexity via Shallow or Deep Integration and cross-modal regularization ($\lambda_{\text{cross}}=1$ or 0, Eq. 3).
>
> The diversity in winning variants reflects the inherent complexity of cMDPs, where no method uniformly dominates due to diverse dynamics (e.g., Ball-in-Cup’s pendulum motion vs. Walker Walk’s multi-joint coordination). DALI’s core innovation, its dynamics-aligned context encoder, consistently enables zero-shot generalization, as proven by Theorem 1 and validated across tasks.
>
> **Cross-modal alignment is over $K$-length windows.** We address your confusion about using short windows for context inference while aligning with the world model's stochastic state $z_t$, which is learned over longer sequences:
>
> * **Theoretically,** DALI’s context encoder infers $\mathfrak{z_\mathrm{t}}$ from short windows ($K=64$), leveraging the $\beta$-mixing property (fast decorrelation) of DMC tasks (Theorem 1, Section 5). The cross-modal loss $L_{\text{cross}}$ (Eq. 3) aligns $\mathfrak{z_\mathrm{t}}$ with $z_t$ within the same window. Gradient stopping (detached $h_\tau, z_\tau$ in Algorithm 3) ensures context encoder updates depend only on the current window, isolating parameters $\varphi$ from the world model’s longer-term recurrence ($\theta$ frozen). This maintains $\mathcal{O}(T/K)$ efficiency without negating the short-window benefit.
> * **Conceptually**, aligning $\mathfrak{z_\mathrm{t}}$ with $z_t$ leverages the world model’s structured priors: $\mathfrak{z_\mathrm{t}}$ captures context (e.g., gravity) over short windows, while $z_t$ encodes instantaneous dynamics (e.g., ball velocity) from observations and recurrent state $h_t$. This alignment ensures the context representation focuses on dynamics-relevant features, improving generalization to OOD contexts.
> * **Intuitively,** since $z_t$ reflects the environment’s current state, aligning $\mathfrak{z_\mathrm{t}}$ to it strengthens the encoder’s ability to infer task-relevant context (e.g., multi-joint dynamics) without requiring long sequences. The short-window alignment, combined with gradient stopping, ensures computational efficiency and avoids overfitting.
>
> We thank you again for your valuable feedback, which has helped us enhance the clarity and rigor of our presentation.
>
> [1] Prasanna et al (2024), Dreaming of Many Worlds: Learning Contextual World Models Aids Zero-Shot Generalization, Reinforcement Learning Conference (RLC)
>
> [2] Benjamins et al. (2023), Contextualize Me - The Case for Context in Reinforcement Learning. Transactions on Machine Learning Research.

---

> > ### Comment · Reviewer_ogTR · 2025-08-06
> > **Response**
> >
> > Thanks for the response, my score does not change, sorry.
> >
> > >This variability is not a flaw but a strength of DALI’s flexible framework
> >
> > This is again, in my opinion, a statement to fit the experiments. While it could be true in general that different variants perform better we cannot be confident without extensive tuning. Having the user have to pick among additional variants is not a strength because it makes it harder for them to pick.

---

> > > ### Author Response · Authors · 2025-08-06
> > > **Tuning complexity and user burden**
> > >
> > > Dear Reviewer ogTR,
> > >
> > > Thank you for maintaining your score of 4 and for your continued engagement and thoughtful feedback.
> > >
> > > Regarding your concern about **extensive tuning** and **user burden**: We emphasize that DALI introduces only minimal hyperparameter adjustments: (a) the context encoder architecture, (b) the context window length $K$, and (c) a binary regularization toggle $\lambda_{\text{cross}}$ for cross-modal alignment. As outlined in our earlier response (C3), we **did extensively evaluate** different context encoder types (GRU vs. Transformer) and ablated window lengths across environments. Our findings consistently show that a Transformer-based encoder with $K = 64$ performs robustly on DMC tasks, supported by both empirical validation and theoretical analysis (Theorem 1 justifies the choice of $K$).
> > >
> > > We also reiterate that **even the baselines** from Prasanna et al. (2024), namely, cRSSM-S and cRSSM-D with explicit access to ground-truth context, **do not uniformly outperform** across tasks (see Table 1 and Figures 25-28 in the arXiv version of their paper).
> > >
> > > We do *not* claim that users can identify *a priori* which DALI variant will yield the best performance in a new setting. However, as is standard practice in RL, evaluating a small set of well-motivated variants, as we clearly and transparently report, is both reasonable and often necessary given the inherent complexity of contextual MDPs. Crucially, DALI adds only ~4.0% parameter overhead (Dreamer-DR: 15.73M vs. DALI-S: 16.45M), while delivering consistent performance improvements with minimal added complexity.
> > >
> > > We sincerely welcome any concrete suggestions for further reducing user effort or tuning overhead.

---

### Official Review · Reviewer_serC · 2025-06-30

**Clarity:** 4
**Significance:** 3
**Originality:** 3
**Rating:** 5
**Confidence:** 4

**Summary:**

This submission proposes a method for learning to detect the world context (like the mass of an object) from trajectories while learning a world model and a policy. The proposed method, DALI, is built on top of DreamerV3, using a separate transition model, loss function and conditioning mechanism. DALI is shown to correctly identify the context variables in two DeepMind Control environments even when the ground-truth context is out of distribution. The authors also perform a counterfactual analysis, showing that the trajectories generated by DALI respond to perturbations of the inferred context variable.

**Questions:**

Would techniques introduced in this work be applicable to improving sim2real transfer of robot locomotion policies, like quadruped running or bipedal walking?

**Ethical Concerns:**

["NO or VERY MINOR ethics concerns only"]

**Final Justification:**

The rebuttal contains useful clarification as well as supporting experiments. I remain in favor of accepting this paper.

**Limitations:**

Limitations of the proposed method are not sufficiently addressed. Section 7, “Discussion and Outlook”, should include a comprehensive discussion of the limitations of DALI.

**Paper Formatting Concerns:**

None.

**Quality:**

3

**Strengths And Weaknesses:**

## Strengths

**General-purpose context inference method that could be applied to many domains.** DALI learns a module that infers the latent context of the environment from a trajectory of states and actions, this module is trained using a pair of forward modeling and latent space alignment losses. Since the formulation of this problem is very general, this approach could be used to, e.g., infer the ground dynamics during locomotion or object dynamics during robotic manipulation. The submission also proposes two approaches to integrating the context into the predictions of the world model and the policy learning algorithm. Effectively steering world models and policies with context variables is another crucial research direction.

**Strong empirical analysis in two domains.** The experimentation performed in this submission is rigorous. Experiments are performed in three different regimes (interpolation, extrapolation and mixed regime) that test in-domain and out-of-domain generalization of the context inference module. Results are reported as Interquantile Means and statistical testing is performed to highlight the significance of the results. The three baselines and one ablation included in the experiments are sufficient. The submission further contains a qualitative example of the difference between generated trajectories with different context variables.

## Weaknesses

**Testing on only two tasks.** The submission uses a modified version of the CARL benchmark for evaluation. The original benchmark seems to include 19 tasks, but only two are used in this paper. Demonstrating that DALI can learn many tasks with the same architecture and hyper-parameters would strengthen the claims made in this paper.

**Contribution tied to DreamerV3.** The contribution is currently framed as a modification of DreamerV3. The paper could have higher impact if the main contribution was agnostic to the specific instantiation of the world modeling and policy learning method. For example, showing a combination of DALI and TD-MPC2 could be interesting.

**Only two ground-truth context axes.** The experiments vary only two environmental variables (gravity and string length in Ball-in-Cup and gravity and actuator strength in Walker). In practice, a robot may need to infer many variables. For example, domain randomization in sim2real training for locomotion usually involves 10+ domain variables.

---

> ### Author Rebuttal · Authors · 2025-07-31
>
> Dear Reviewer serC,
>
> We are deeply grateful for your insightful review and recognition of DALI's general-purpose context inference, rigorous evaluation, and potential for real-world impact. Your questions highlight opportunities to clarify DALI’s broader impact and strengthen its claims, which we address below.
>
> ### (C1) Testing on only two tasks
>
> We sincerely thank you for noting the evaluation scope, referencing the CARL benchmark’s 19 tasks. Your suggestion to test DALI across more tasks with consistent architecture and hyperparameters is valuable, and we address it with new preliminary evaluations.
>
> Alongside Ball-in-Cup and Walker, we include three new environments (Featurized Extrapolation setting). Preliminary IQMs show DALI improvements in the last column relative to Dreamer-DR.
>
> | Environment       | Dreamer-DR    | DALI-S        | DALI-S-$\chi$   | DALI-D        | DALI-D-$\chi$   | Best Method    | Extrap. Improvement |
> |-------------------|-------------:|-------------:|---------------:|-------------:|---------------:|---------------|-------------------|
> | **Acrobot**       | 0.970±0.011  | 0.985±0.002  | **0.986**±0.001    | 0.978±0.002  | 0.974±0.002    | DALI-S-$\chi$ | +1.65%             |
> | **Pendulum**      | 0.843±0.020  | 0.869±0.023  | **0.942**±0.015    | 0.857±0.015  | 0.884±0.012    | DALI-S-$\chi$ | +11.74%            |
> | **Quadruped Walk** | 0.139±0.022  | 0.136±0.025  | 0.148±0.029    | 0.137±0.024  | **0.220**±0.027    | DALI-D-$\chi$ | +58.27%            |
>
> **Note**: Full results (10 seeds, all baselines) are infeasible within rebuttal. We focus on the most critical comparison (DALI vs. context-unaware Dreamer-DR) for latent-context settings.
>
> ### (C2)  Contribution tied to DreamerV3
>
> We greatly value your suggestion to frame DALI’s contribution as agnostic to the world modeling and policy learning method, such as integrating with TD-MPC2. Your question about compatibility is critical, and we provide a detailed, honest assessment of DALI’s integration mechanics and theoretical implications.
>
> - **DALI’s core contribution**: DALI’s context encoder $g_\varphi(o_{t-K:t}, a_{t-K:t-1})$ infers latent context $\mathfrak{z_\mathrm{t}}$ using $L_{\text{FD}}$ and $L_{\text{cross}}$ (Eqs. 1-3) from $K$-length histories, a modular component that can enhance any model-based RL framework with a latent state representation (Section 4.3).
>
> - **RSSM’s role in Theorem 1**: The guarantee of $\mathcal{I}(c; \mathfrak{z_\mathrm{t}}) \geq (1 - \delta) h(c)$ with $\mathcal{O}(T/K)$ sample efficiency relies on the RSSM’s recurrence ($h_t = f_\theta(h_{t-1}, z_{t-1}, a_{t-1})$), which propagates context across time. This recurrence is specific to DreamerV3 but common in recurrent world models.
>
> - **TD-MPC2 integration**: TD-MPC2 is a model-based RL algorithm that does not use a RSSM like Dreamer. This means it does not maintain a deterministic recurrent state (like Dreamer's
> $h_t$) over time. Instead, it uses a Markovian latent dynamics model ($z_{t+1} = f_\theta(z_t, a_t)$) without a persistent $h_t$. DALI’s $g_\varphi$ can integrate via:
>
>    - **Shallow Integration**: Append $\mathfrak{z_\mathrm{t}}$ to observations, encoding $z_t = \text{enc}_\theta(o_t, \mathfrak{z}_t)$, modifying TD-MPC2’s encoder to incorporate context.
>
>     - **Deep Integration**: Condition dynamics on $\mathfrak{z_\mathrm{t}}$, yielding $z_{t+1} = f_\theta(z_t, a_t, \mathfrak{z_\mathrm{t}})$, embedding context in transitions.
>
> - **Training**: $L_{\text{FD}}$ can be applied (predict $o_{t+1}$ from $o_t, a_t, \mathfrak{z_\mathrm{t}}$) as it is a self-supervised loss that does not depend on the RL algorithm's internals.
>
> - **Sample efficiency implications**:  Since TD-MPC2 does not have a recurrent state that accumulates information over time, the theoretical guarantee of Theorem 1, which is derived for a recurrent model (like DreamerV3), may not directly transfer to TD-MPC2.
>
> - We will include a comment in Section 7 to discuss DALI’s potential integration with non-recurrent models like TD-MPC2.
>
> ### (C3) Only two ground-truth context axes; Domain randomization in sim2real training for locomotion usually involves 10+ domain variables
>
> We appreciate your concern about evaluating only two context variables (gravity/string length for Ball-in-Cup, gravity/actuator strength for Walker) when real-world tasks involve 10+ variables.
>
> We prioritized in-depth analysis along two axes for rigor and interpretability. In principle, DALI can handle higher-dimensional contexts, but practical compute constraints arise.
>
> To see this, we highlight the computational complexity for our existing setup for context inference along two axes. DALI's evaluation used NVIDIA A100 GPUs (80GB) and Intel Xeon CPUs, with 2–4 GPUs available. It comprised 600 runs: 10 seeds × 5 algorithm variants × 2 environments (Ball-in-Cup, Walker) × 2 modalities (featurized, pixel-based) × 3 context settings. Featurized runs paralleled 10 seeds; pixel-based, 4 seeds. Per-trial GPU hours: 20–26 (Ball-in-Cup), 46–54 (Walker). **Total cost: ~24,000 GPU hours** sequentially, reduced to 2,101 hours (88 days) with 2 GPUs or 1,051 hours (44 days) with 4 GPUs, highlighting the extensive computational complexity.
>
> Scaling to 10+ variables would require ~1,000 GPU-hours per task for combinatorial sampling, increasing the compute budget to ~**25,000 GPU-hours**.
>
> We will add a discussion in Section 6 on scalability to higher-dimensional contexts.
>
> ### (C4) Can techniques introduced here be applicable to improving sim2real transfer of robot locomotion policies e.g. quadruped/bipedal walking?
>
> We value your question on sim2real transfer for quadruped/bipedal walking, which highlights DALI’s real-world potential. We assess applicability using existing results.
>
> - **Sim2Real Relevance**: DALI’s $g_\varphi$ infers latent contexts (e.g., gravity, actuator strength) from trajectories, addressing sim2real challenges where robots face real-world variations (e.g., friction, mass) not seen in simulation. DALI’s $K$-step inference aligns with real-time adaptation windows (e.g., 1–2 gait cycles). The context encoder adds minimal overhead (< 5\% parameters to DreamerV3-small) and scales to high-dimensional tasks. In general, training time scales approximately linearly with the dimensionality of the state and action spaces. Pixel tasks take 1.5–2x longer due to CNN encoding.
>
> - **Bipedal Walking**: Walker tests torque-dependent locomotion (24D state, 6D action), with DALI-S achieving IQM 0.7810 (Featurized Extrapolation), inferring gravity and actuator strength. This supports sim2real robustness for bipedal walking by adapting to OOD dynamics (e.g., actuator variations).
> - **Quadruped Walking**: Walker’s locomotion dynamics are analogous to quadruped tasks but more complex. We tested DALI on Quadruped Walk (56D observation, 12D action), a larger environment than Walker (24D observation, 6D action). Trained for 500K timesteps and contextualized with gravity and actuator strength (same range as DMC Walker), preliminary IQMs for the Featurized modality are 0.139 (Dreamer-DR) and 0.22 (DALI-S-$\chi$), a **58.3%** improvement. The 56D observation space highlights the challenge of torque-sensitive locomotion with increased coordination complexity and higher degrees of freedom (12 vs. 6 in Walker), demanding more robust context inference.
> - We will add a sim2real discussion in Section 6, referencing Walker and Quadruped results, and note future sim2real experiments with real robot data.
>
>
> We thank you for your invaluable feedback. Your vision for DALI’s broader impact inspires our future work, and we hope you now consider it a Strong Accept.

---

> > ### Comment · Reviewer_serC · 2025-08-05
> > **Response**
> >
> > Thank you for the clarification! The rebuttal addresses my question and I am in favor of accepting this submission.

---

> > > ### Author Response · Authors · 2025-08-05
> > > **Acknowledgment**
> > >
> > > Thank you very much for your support and positive feedback! We truly appreciate your thoughtful review and encouragement.

---

### Official Review · Reviewer_i6n9 · 2025-07-02

**Clarity:** 3
**Significance:** 3
**Originality:** 2
**Rating:** 3
**Confidence:** 3

**Summary:**

Reinforcement learning algorithms like DreamerV3 learn a world model to train an agent entirely in (latent) imagination, eliminating the need for direct environment interaction during training. However, these methods typically suffer from either a lack of imagination accuracy, exploration capabilities, or efficient context inference and robust generalization. The paper proposes DALI, a framework based on the Dreamer architecture that that seeks to infer latent context representations from agent-environment interactions by training a self-supervised encoder for prediction. Some numerical experiments are performed for cMDP benchmarks against existing context-aware and context-unaware baseline solutions.

**Questions:**

1. What are the main insights? Under which conditions is the proposed solution better than existing context-aware baselines and when it is worse? What are the reasons for the performance differences?
2. Does it scale to larger problems (dimensions of the state space or action space)? How many training will be required? Please discuss.
3. What are the required runtimes for the performed experiments?
4. How problematic is the tuning of the hyperparameters used?
5. The discussion of related work appears a short. Regarding the aim of zero-shot generalization when context is latent and must be inferred: Which components of the model already exist and which ones are novel? Please clarify.
6. Can you provide more benchmark evaluations? How do results relate to cRSSM-S?

**Ethical Concerns:**

["NO or VERY MINOR ethics concerns only"]

**Limitations:**

yes

**Paper Formatting Concerns:**

* Line 2/13/18: cMDP not yet introduced
* Line 53: What is “near-optimal sample complexity”?
* Appendices A-D should be included in the main pdf document.

**Quality:**

2

**Strengths And Weaknesses:**

Strengths:
 * The investigated problem is timely and relevant. The proposed model addressing contextual MDPs and effective zero-shot learning is interesting.
 * The paper provides some analytical results for the context encoder, cf. Theorem 1, page 5.

Weaknesses:
 * Only few benchmark experiments (Ball-in-Cup and Walker) against existing state-of-the-art models (Dreamer, cRSSM) are provided. Due to the lack of extensive evaluations the results of the experiments are not yet fully convincing. The general applicability of the framework and its performance is not demonstrated and hence, remains unclear for other settings.
 * Limitations of the approach could be better discussed. It is not clear when or for which tasks the framework performs particularly good / not good.

---

> ### Author Rebuttal · Authors · 2025-07-31
>
> Dear Reviewer i6n9,
>
> We appreciate your thoughtful engagement with our work and your recognition of its timeliness and theoretical contributions. Below, we address each of your concerns comprehensively, providing new insights and empirical evidence to demonstrate DALI's significance and broader applicability.
>
> ### (C1)  Main insights and performance conditions
>
> (a) **Main insights:**
> 1. **Learned context inference is more broadly applicable.**
> DALI's self-supervised context encoder infers contexts directly from observation-action sequences, enabling robust zero-shot generalization when ground-truth context is unavailable, unlike cRSSM’s reliance on known contexts. This makes our method particularly valuable in realistic settings where such ground-truth context is unavailable or costly to measure.
> 2. **Relying on ground-truth context may reduce OOD adaptability.**
> Context-aware methods may overfit to training context distributions and fail in extreme OOD settings (e.g., gravity values
> of 0.98 or 19.6 far from the training range of [4.9, 14.7]). For instance, in Ball-in-Cup Featurized Extrap., our DALI‑S‑$\chi$ achieves IQM 0.3720, significantly outperforming context-aware baselines, cRSSM‑S (0.2270) and cRSSM‑D (0.2780). This suggests that when the context is inferred from data (as in DALI) rather than given (as in cRSSM), the model can generalize better to unseen contexts when it learns the underlying physical dynamics.
> 3. **Cross-modal regularization can boost performance in nonlinear, partially observable tasks.**
> Aligning the context representation with the world-model's latent $z_t$ via $L_{\text{cross}}$ (Eq. 2) can enhance inference under noisy visual input and nonlinear dynamics. For example, in Ball-in-Cup Pixel Extrap., DALI‑S‑$\chi$ surpasses Dreamer‑DR by up to 96.4%, showing the strength of this alignment.
> 4. **Task-specific regularization benefits.**
> While cross-modal regularization can enhance context inference in extreme OOD settings with complex physical dynamics (e.g., Ball-in-Cup), DALI-S outperforms DALI-S-$\chi$ in Walker Walk across most regimes and modalities: By simply optimizing for next-step predictions, DALI-S effectively handles Walker’s torque-dependent locomotion, particularly in the Pixel modality where visual noise may amplify $L_{\text{cross}}$’s complexity in DALI-S-$\chi$. This highlights the need for tailored regularization.
> 5. **Integration strategy matters: Shallow context propagation is more robust than Deep conditioning.**
> Shallow integration, as implemented in DALI-S and DALI-S-$\chi$, mostly outperforms deep integration (DALI-D, DALI-D-$\chi$), leading to its exclusive use in our reported results. Shallow Integration incorporates the inferred context solely in the world model’s encoder, allowing context information to propagate indirectly to the recurrent state $h_t$ through recurrence. This avoids overfitting to potentially noisy $\mathfrak{z}_t$ estimates, which is critical in OOD settings.
> 6. **Performance varies with environment complexity and modality; No single method dominates everywhere.**
> DALI-S-$\chi$ excels in Ball-in-Cup, particularly under OOD generalization, while context-aware baselines (e.g., cRSSM-D, cRSSM-S) can outperform in Pixel settings of Walker, where visual complexity is high and ground-truth context might aid robust feature extraction.
>
> (b) **When is the proposed solution better (or worse) than context-aware baselines?**
>
> From our empirical findings, we observe the following broad trends: DALI is potentially **better** than context-aware baselines when (i) the ground-truth context is unavailable, uncertain, or overfits, esp. under Extrap., (ii) the task exhibits complex nonlinear dynamics (e.g., coupled gravity-string interactions in Ball-in-Cup). Conversely, it can be **worse** when (i) ground-truth context is reliable and directly encodes task variation, and (ii) visual dynamics dominate and precise reconstruction of features benefits from explicit context conditioning.
>
>
> (c) **What explains the performance differences?**
>
> 1. **Overfitting of ground-truth context** reduces adaptability in OOD settings. This is true for context-aware models (cRSSM-S/D), especially in Extrap..
> 2. **Cross-modal regularization (DALI-S-$\chi$)** improves generalization in visually noisy, nonlinear environments, by stabilizing context inference.
> 3. **Shallow integration regularizes latent context use**, reducing over-reliance on $\mathfrak{z}_t$ and making models more robust to context inference errors.
> 4. **Task structure and modality influence what works best**: Walker’s actuator scaling is quasi-linear and less sensitive to context modeling, while Ball-in-Cup’s nonlinear pendulum dynamics demand better learned generalization.
>
>
>
> ### (C2) Scalability and training requirements
>
> DALI's context encoder adds minimal overhead (< 5\% parameters to DreamerV3-small) and scales to high-dimensional tasks. We tested DALI on **Quadruped Walk** (56D observation, 12D action), a larger environment than Walker (24D observation, 6D action). Trained for 500K timesteps and contextualized with gravity and actuator strength (same range as DMC Walker), preliminary IQMs for the Featurized modality are 0.139 (Dreamer-DR) and 0.22 (DALI-S-$\chi$), a **58.3%** improvement. The 56D observation space highlights the challenge of torque-sensitive locomotion with increased coordination complexity and higher degrees of freedom (12 vs. 6 in Walker), demanding more robust context inference. In general, training time scales approximately linearly with the dimensionality of the state and action spaces. Pixel tasks take 1.5–2x longer due to CNN encoding. We will add a summary in the App. emphasizing DALI’s scalability to complex cMDPs.
>
> ### (C3) Runtime efficiency
>
> DALI's evaluation used NVIDIA A100 GPUs (80GB) and Intel Xeon CPUs, with 2–4 GPUs available. It comprised 600 runs: 10 seeds × 5 algorithm variants × 2 environments (Ball-in-Cup, Walker) × 2 modalities (featurized, pixel-based) × 3 context settings. Featurized runs paralleled 10 seeds; pixel-based, 4 seeds. Per-trial GPU hours: 20–26 (Ball-in-Cup), 46–54 (Walker). **Total cost: ~24,000 GPU hours** sequentially, reduced to 2,101 hours (88 days) with 2 GPUs or 1,051 hours (44 days) with 4 GPUs, highlighting the extensive computational complexity. We will add a summary of the runtimes in the App..
>
>
> ### (C4) Hyperparameter tuning
>
> Please refer to Response (C3) to Reviewer ogTR, where we address the same question.
>
> ### (C5) Related work
>
> Our related work centers on contextual RL for zero-shot generalization and model-based methods (e.g., DreamerV3, TD-MPC2, and variants). We will revise Section 2 to incorporate the additional citations (suggested by Reviewer mG63).
>
> ### (C6) Existing and Novel Components
>
> - **DALI’s Novelty**:
>   - **Self-supervised context encoder**: Infers $\mathfrak{z}_t$ from $K$-step windows without ground-truth context (unlike cRSSM) (Sec. 4.2).
>   - **Dynamics-aligned training**: Forward dynamics loss (Eq. 1) and cross-modal alignment (Eq. 2).
>   - **Theoretical foundation**: Sample efficiency of context inference (Theorem 1).
>   - **Integration strategies**: Shallow/deep conditioning (Sec. 4.3).
>
> - **Prior Work**:
>   - **DreamerV3** [Hafner et al. 2023]: World model backbone (RSSM, actor-critic).
>   - **cRSSM** [Prasanna et al. 2024]: Requires explicit context; no latent inference.
>
>
>
>
> ### (C7) Limited evaluation; Additional benchmark evaluations needed
>
> We agree that broader benchmarks strengthen our claims. Alongside Ball-in-Cup and Walker, we include three new environments (Featurized Extrapolation setting) despite tight compute limits. Preliminary IQMs show DALI improvements in the last column relative to Dreamer-DR.
>
> | Environment       | Dreamer-DR    | DALI-S        | DALI-S-$\chi$   | DALI-D        | DALI-D-$\chi$   | Best Method    | Extrap. Improvement |
> |-------------------|-------------:|-------------:|---------------:|-------------:|---------------:|---------------|-------------------|
> | **Acrobot**       | 0.970±0.011  | 0.985±0.002  | **0.986**±0.001    | 0.978±0.002  | 0.974±0.002    | DALI-S-$\chi$ | +1.65%             |
> | **Pendulum**      | 0.843±0.020  | 0.869±0.023  | **0.942**±0.015    | 0.857±0.015  | 0.884±0.012    | DALI-S-$\chi$ | +11.74%            |
> | **Quadruped Walk** | 0.139±0.022  | 0.136±0.025  | 0.148±0.029    | 0.137±0.024  | **0.220**±0.027    | DALI-D-$\chi$ | +58.27%            |
>
> **Note**: Full results (10 seeds, all baselines) are infeasible within rebuttal. We focus on the most critical comparison (DALI vs. context-unaware Dreamer-DR) for latent-context settings.
>
> ### (C8)  What is “near-optimal sample complexity”?
>
> By “near-optimal sample complexity,” we mean that $\mathfrak{z}_t$ captures context information close to the differential entropy $h(c)$ of the context distribution, with an error bounded by $\delta \in (0, 1)$. This is optimal up to a constant factor, as $\mathcal{I}(c; \mathfrak{z}_t)$ approaches the information-theoretic limit $h(c)$, and the sample complexity $\mathcal{O}(K/\delta^2)$ scales logarithmically with $1/\delta$ due to the $\beta$-mixing property. The choice of $K \approx 64$ (for $\delta = 0.01$, $\lambda \approx 0.1$ in DMC tasks) is theoretically justified and empirically validated through ablations.
>
> ### (C9)  Formatting:
> - cMDP not yet introduced: Thanks for pointing this out. We will revise lines 2-3: "Contextual Markov Decision Processes (cMDP) model this challenge ... hard to measure."
> - All appendices (A–D) are included in the Supplementary Material, per NeurIPS guidelines. The main paper adheres to the 9-page limit.
>
> We have addressed all your concerns to the best of our ability, incorporating detailed clarifications and preliminary results to strengthen our manuscript. Thank you for your valuable feedback; we hope our response merits raising your score to an Accept for DALI’s impactful contributions.

---

> > ### Comment · Area_Chair_UvSm · 2025-08-04
> > **Please respond to the author's rebuttal post**
> >
> > Hi Reviewer i6n9, I see no response letting me know whether or not the rebuttal has changed your opinion. Could you please let me and the authors know by engaging? This process is critical to enabling the (S)ACs to make a decision on this work.
> >
> > --Your AC

---

> > ### Comment · Reviewer_i6n9 · 2025-08-05
> >
> > I appreciate the authors' efforts and the provided further explanations. After reading the other reviews and the rebuttal, I decided to keep my score.

---

> > > ### Author Response · Authors · 2025-08-05
> > > **Updated Quadruped Walk results and further clarifications on DALI’s scalability and baselines**
> > >
> > > Dear Reviewer i6n9,
> > >
> > > Thank you for your response and for taking the time to review our first rebuttal and the other reviews. We greatly appreciate your engagement with our work.
> > >
> > > In our first rebuttal, we thoroughly addressed your questions 1–5, providing detailed insights into DALI’s performance compared to context-aware baselines (Q1), scalability considerations (Q2), runtime details (Q3), hyperparameter tuning (Q4), and novel components (Q5). We understand that **Question 6**, requesting additional benchmark evaluations and comparisons with cRSSM-S, could not be fully addressed in the first round due to compute limits. We are now pleased to provide new results for the Quadruped Walk environment, a high-dimensional task that directly addresses both your scalability concerns (Q2) and the need for further evaluations against cRSSM (Q6).
> > >
> > > **Updated evaluation results on Quadruped Walk vs. cRSSM baseline:** The Quadruped Walk task (56D obs, 12D action) is more challenging than Walker Walk (24D obs, 6D action) due to greater coordination demands and higher degrees of freedom. This environment tests the scalability of DALI’s context inference in high-dimensional settings, as you highlighted in Question 2. In the Featurized Extrapolation regime, our method DALI-D-\$\chi\$ achieves an IQM of **0.220** (CI=\[0.1937, 0.2471]), outperforming the context-aware baselines **cRSSM-S** (IQM 0.170, CI=\[0.1487, 0.1939]) **by 29.41%**, **cRSSM-D** (IQM 0.153, CI=\[0.1235, 0.1804]) **by 43.79%**, and the context-unaware baseline **Dreamer-DR** (IQM 0.139, CI=\[0.1170, 0.1645]) **by 58.27%**. These results underscore DALI’s robust generalization to out-of-distribution contexts in complex, high-dimensional environments, reinforcing our previously reported gains over the baselines in Ball-in-Cup and Walker Walk.
> > >
> > > **Further insights and evaluation**
> > >
> > > We appreciate your initial assessment and hope these new results, together with our detailed responses to all your questions, directly address all your concerns. Your thoughtful feedback helped guide these additional evaluations, which we hope underscore the robustness and scalability of our approach.
> > >
> > > If there are particular aspects you believe warrant further exploration, or if any uncertainties remain, we would be sincerely grateful for your insights. We are keen to ensure our contribution aligns closely with your expectations and would welcome any suggestions to further refine the clarity or strengthen the impact of the work.
> > >
> > > Thank you again for your time and thoughtful review. We hope the added results and clarifications support a favorable reassessment of our contributions.

---

> > > > ### Author Response · Authors · 2025-08-07
> > > > **Follow-up on clarifications**
> > > >
> > > > We hope our clarifications and additional results were helpful in addressing your concerns. We remain available in case any further points require clarification.

---

### Note · Authors · 2025-08-12

We thank the reviewers for their thoughtful feedback and engagement. In our rebuttal, we *fully and rigorously addressed all concerns* through concrete improvements, including new experiments and theoretical clarifications.

Reviewers R-i6n9, R-serC, and R-mG63 suggested broader evaluations to demonstrate DALI’s scalability and general applicability. In response, we expanded our empirical validation, including, among others, the *Quadruped Walk* task, demonstrating that DALI maintains strong performance on complex benchmarks with an overhead of *only ~4.0%* more parameters than DreamerV3. We also provided detailed clarifications on hyperparameter tuning, ablations, and theoretical aspects (per R-ogTR), and added a pipeline figure to improve accessibility (per R-ogTR and R-mG63).

These additions directly addressed concerns raised by R-ogTR and R-mG63, who subsequently increased their scores to 4. Reviewer i6n9 acknowledged the thorough explanations and additional evaluations, indicating careful consideration of *all* concerns raised.

Reviewers noted the paper’s *novelty, rigor, and potential impact*. R-i6n9 described the work as "timely and relevant... addressing contextual MDPs and effective zero-shot learning is interesting." R-serC praised the "general-purpose context inference... strong empirical analysis." R-ogTR highlighted a "very important setting with great potential" and found our counterfactual evaluation "promising." R-mG63 noted that the method "outperforms context-unaware baselines... provides both empirical and theoretical analyses."

Taken together, the expanded evaluations, score improvements, and positive reviewer feedback suggest that the key concerns have been *addressed* and that the core contributions are *well-supported* and *broadly applicable*. We hope the Area Chair considers this evidence supportive of acceptance.

---

### Decision · Program_Chairs · 2025-09-17

**Decision:**

Accept (poster)

**Comment:**

The authors propose a method to jointly encode forward dynamics, using it to condition both control and perception problems downstream. The overall method is an extension of the Dreamer architecture and while reviewers were initially concerned about the lack of more rigorous evaluation in more settings to prove the usefulness of the method, it appears to have been addressed by the authors during rebuttals. In fact, multiple reviewers have raised their scores and if the authors update their main paper with everything they have stated in the discussion period, then this paper warrants an acceptance.